# DV-World: Benchmarking Data Visualization Agents in Real-World Scenarios

Jinxiang Meng [* 1 2]  Shaoping Huang [* 1 2]  Fangyu Lei [* 1 2]  Jingyu Guo [1]  Haoxiang Liu [1]  Jiahao Su [1]
Sihan Wang [1]  Yao Wang [1]  Enrui Wang [1]  Ye Yang [1]  Hongze Chai [1]  Jinming Lyu [1]  Anbang Yu [1]
Huangjing Zhang [1]  Yitong Zhang [3]  Yiming Huang [1]  Zeyao Ma [4]  Shizhu He [1 2]  Jun Zhao [1 2]  Kang Liu [1 2]

## Abstract

Real-world data visualization (DV) requires native environmental grounding, cross-platform evolution, and proactive intent alignment, yet existing benchmarks are often limited to code sandboxes, creation-only tasks, and fully specified intents. We introduce DV-World, a 260-task benchmark for evaluating DV agents across professional visualization lifecycles. DV-World covers three domains: DV-Sheet for native spreadsheet chart/dashboard creation and diagnostic repair; DV-Evolution for adapting reference visual artifacts to new data across programming paradigms; and DV-Interact for proactive intent alignment with a user simulator under ambiguous requirements. Its hybrid evaluation combines *Table-value Alignment* for numerical precision with rubric-based *MLLM-as-a-Judge* for semantic-visual assessment. Experiments show that state-of-the-art models achieve below 50% overall performance, exposing major gaps in real-world DV capabilities. DV-World provides a realistic testbed for developing DV agents suited to enterprise workflows. Data and code are available at `dv-world-project.github.io`.

## 1. Introduction

Data visualization (DV) is the critical interface bridging abstract data and decisions. The rapid evolution of Large Language Models (LLMs) and Multimodal LLMs (MLLMs) has driven a surge in DV agents (Goswami et al., 2025; Ouyang et al., 2025; Chen et al., 2025c; Ni et al., 2025; Seo et al., 2025; Chen et al., 2025a), which demonstrate impres-

sive capabilities in synthesizing executable scripts within standardized code sandboxes. Simultaneously, multimodal reasoning has matured to support visual deconstruction and complex analysis (Yang et al., 2024a; Luo et al., 2025). Similarly, evaluation paradigms in adjacent domains are pivoting toward authentic, ecosystem-grounded workflows (Ma et al., 2024). However, benchmarks remain confined to idealized, simplified settings, failing to capture the complexity and ambiguity in real-world scenarios.

Current agents remain ill-equipped for real-world visualization due to three gaps: *environmental decoupling*, *creation-only myopia*, and *perfect-intent assumptions*. First, *environmental decoupling* overlooks spreadsheet-centric visualization workflows: many benchmarks emphasize developer-style code generation and bypass the native chart object models, data-to-chart bindings, and GUI constraints that govern practical spreadsheet visual analytics (Ma et al., 2024; Xie et al., 2024). Second, the *creation-centric paradigm* evaluates one-shot chart construction but under-tests *evolutionary* work, where agents must revise visualizations under new data and requirements while preserving structure and aesthetics across diverse frameworks (Jimenez et al., 2023; Lei et al., 2025). Third, the *assumption of perfect intent* ignores ambiguity in user requests; benchmarks built on fully specified prompts miss the need for proactive clarification and dialogue-based alignment (Min et al., 2020; Wang et al.; Huo et al., 2025), rather than blindly executing commands.

To bridge these gaps, we introduce **DV-World**, a comprehensive benchmark evaluating data visualization agents on the full lifecycle of *native manipulation in real software environments* (**DV-Sheet**), *cross-modal logic evolution* (**DV-Evolution**), and *proactive iterative interaction* (**DV-Interact**). **(1) DV-Sheet** establishes native grounding by focusing on the manipulation of native spreadsheet object models. It encompasses three sub-tasks: *DVSheet-Create* requires generating native charts from diverse data using direct cell references; *DVSheet-Fix* requires diagnostic reasoning to identify and repair broken visualization bindings; and *DVSheet-Dashboards* evaluates holistic spatial planning by synthesizing multiple charts and tables into professional analytical views. **(2) DV-Evol** targets *cross-modal logic evolution* by requiring agents to convert reference images and datasets

---

[1] The Key Laboratory of Cognition and Decision Intelligence for Complex Systems, Institute of Automation, Chinese Academy of Sciences, Beijing, China [2] School of Artificial Intelligence, University of Chinese Academy of Sciences [3] NUS [4] Renmin University of China. Correspondence to: Kang Liu <kliu@nlpr.ia.ac.cn>.

*Proceedings of the 43$^{rd}$ International Conference on Machine Learning*, Seoul, South Korea. PMLR 306, 2026. Copyright 2026 by the author(s).

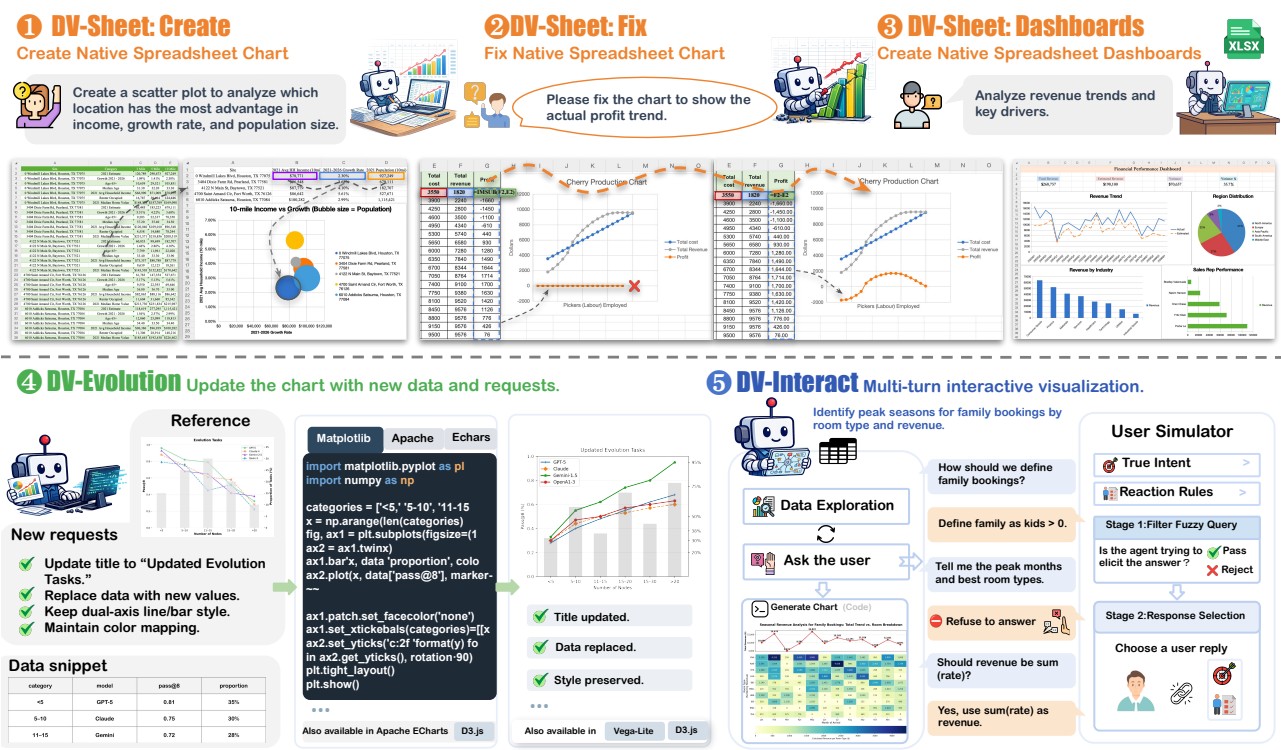

*Figure 1.* DV-World aims to evaluate data visualization agents across the full lifecycle of native manipulation in real software environments (**DV-Sheet**), cross-modal logic evolution (**DV-Evolution**), and proactive iterative interaction (**DV-Interact**) scenarios.

into executable code under evolving requirements. Unlike static reproduction (Yang et al., 2024a; Li et al., 2025b), it evaluates iterative updates and logic migration across five paradigms (e.g., Python, Vega-Lite, D3.js; see App. F.2), testing whether agents can generalize visualization principles beyond a single language or framework. **(3) DV-Inter** focuses on *proactive iterative interaction* under ambiguous user intent. Agents must identify underspecified requirements or data ambiguities, ask targeted clarification questions, and maintain state consistency across turns to ensure the final visualization matches the user's true analytical goal.

To assess these multifaceted capabilities, we establish a hybrid evaluation framework that transcends conventional code verification. We introduce a specialized *Table-value Alignment* protocol to enforce data fidelity, since numerical precision is a prerequisite for visual utility. Complementing this, we employ a hierarchical *MLLM-as-a-Judge* system (Li et al., 2024; Chen et al., 2024a) guided by fine-grained rubrics (Gou et al., 2025; Lei et al., 2025) annotated by experts to capture visual semantics and aesthetic compliance. Experiments demonstrate that this multi-dimensional system aligns closely with human judgment.

Our experiments on DV-World highlight a significant challenge: even state-of-the-art agents struggle with multi-tool visualization. In DV-Sheet, capabilities are pushed to their limits with a peak score of 40.48, revealing a critical short-

coming in managing native object models and dynamic bindings. Performance in DV-Evol and DV-Inter is similarly capped at 51.44 and 40.43, respectively, exposing significant semantic brittleness in logic migration and iterative alignment. Progress requires a shift from one-shot generation to comprehensive lifecycle management, integrating environmental mastery, semantic portability, and proactive intent alignment. By offering this rigorous testbed, DV-World guides DV agent development toward the integrated, robust capabilities needed for real-world visualization challenges.

## 2. Benchmark Construction

### 2.1. Task Definition

To bridge this gap, we design tasks covering three real-world visualization challenges: native spreadsheet charting, visualization evolution, and interactive visualization (Fig. 1).

**DV-Sheet**. An agent $\pi^{sheet}$ performs end-to-end native visualization editing in spreadsheet software. We model the process as $\mathcal{E}_\star = \pi^{sheet}(\mathcal{I}, \mathcal{E}_0)$, where $\mathcal{I}$ is the user instruction and $\mathcal{E}_0$ is the initial workbook. We evaluate three task types: (1) *DVSheet-Crea*: generate a native chart and place it in a new worksheet with dynamic range bindings $f$ (not hard-coded values); (2) *DVSheet-Fix*: diagnose and repair a defective chart $C_{err}$ into a corrected chart $C_{fix}$; (3) *DVSheet-Dash*: compose a professional dashboard in a new worksheet

*Table 1.* Comparison of DV-World with existing benchmarks. **Nat.** and **Prog.** denote Native and Programmatic environments, respectively; **Env.**, **NL**, **I**, **Synth.**, and **Fin.Tab** denote Environment, Natural Language, Image, Synthetic, and Final Table, respectively.

| Benchmark | # Tasks | Source | Env. | Input Format | Final Output | Interactive Agency | Open-ended | Evaluation Method |
|---|---|---|---|---|---|---|---|---|
| *Other Benchmarks* | | | | | | | | |
| SpreadsheetBench (Ma et al., 2024) | 912 | Real | Nat. | Sheet+NL | Sheet | ✗ | ✗ | Rule-based |
| Bird-Interact (Huo et al., 2025) | 600 | Manual | Prog. | DB+NL | 1 SQL | ✓ | ✗ | Execution-based |
| OSWorld (Xie et al., 2024) | 369 | Real+Manual | OS | Actions +NL | Actions | ✗ | ✗ | Execution-based |
| DAComp-DA (Lei et al., 2025) | 210 | Real+Manual | Prog. | Table+NL | Report+Chart | ✗ | ✓ | LLM-judge(rubrics) |
| *Data Visual Benchmarks* | | | | | | | | |
| Plot2Code (Wu et al., 2025) | 132 | Synthetic | Prog. | I+NL | Code | ✗ | ✗ | MLLM-judge |
| ChartMimic (Yang et al., 2024a) | 4,800 | Manual | Prog. | I+NL | Code | ✗ | ✗ | Multi-Level |
| DA-Code (Huang et al., 2024) | 500 | Manual | Prog. | Table+NL | 1 Chart | ✗ | ✗ | Rule-based |
| VisEval (Chen et al., 2024b) | 2,524 | Real+Synth. | Prog. | Table+NL | 1 Chart | ✗ | ✗ | Multi-Level |
| MatPlotBench (Yang et al., 2024b) | 100 | Manual | Prog. | Table+NL | 1 Chart | ✗ | ✗ | MLLM-judge |
| Text2Vis (Rahman et al., 2025) | 1,985 | Real+Synth. | Prog. | Table+NL | 1/N Chart | ✗ | ✗ | Multi-Level |
| nvBench 2.0 (Luo et al., 2025) | 7,878 | Synthetic | Prog. | Table+NL | N Chart | ✗ | ✓ | MLLM-judge |
| PlotCraft (Zhang et al., 2025b) | 982 | Real | Prog. | Table+NL | 1/N Chart | ✗ | ✗ | Multi-Level |
| **DV-World (Ours)** | **260** | **Real + Manual** | **Nat. & N Prog.** | **Sheet + NL & Table + I + NL** | **Fin.Table + 1/N Chart** | **✓** | **Both** | **Rule-based & MLLM-judge (rubrics)** |

by arranging multiple charts $\{C_i\}$ and tables $\{T_j\}$ into a coherent, insight-oriented layout spanning one or more sheets. Agents manipulate the workbook programmatically using libraries (e.g., `openpyxl/xlwings`) (details in App. F).

**DV-Evol.** An agent $\pi^{evol}$ evolves visual assets into executable code through logic synthesis, modeled as $\sigma = \pi^{evol}(\mathcal{I}, V, D, \mathcal{L})$. Given a reference image $V$, a new dataset $D$, and modification requirements $\mathcal{I}$, the agent must reverse-engineer the visual semantics of $V$ to produce an output $\sigma = \langle C_\star, T_\star \rangle$ in the target language $\mathcal{L}$. This artifact consists of the functional plotting code $C_\star$ and the resultant table $T_\star$ containing the data used for the visualization. DV-Evol covers multiple visualization frameworks (e.g., Python, Vega-Lite, and D3.js; see App. F.2).

**DV-Inter.** Given an ambiguous visualization task $q_0$ and a stateful environment $\mathcal{E} = \{D, \mathcal{L}\}$, the agent $\pi_{int}$ processes the data $D$ based on the task $\mathcal{I}$ and generates the corresponding visualization code $C_\star$. The agent interacts with the *ask_user* tool and a dual stage user simulator to clarify ambiguities in the task. This simulator first utilizes an interaction gatekeeper to detect and refuse cheating attempts such as requests for implementation code or internal schema details. If the inquiry is permitted, a response generator then provides feedback grounded in *true intents and reaction rules* (see App. B). This mechanism ensures that the generated code $C_\star$ produces the expected visualization $T_\star$ that aligns with the user latent objectives through authentic reasoning rather than information leakage.

## 2.2. Evaluation Metrics

To quantify the capabilities of data visualization agents, we adopt a hybrid evaluation framework that measures performance across dimensions. This approach integrates quali-

tative rubrics curated by experts with quantitative metrics to ensure a comprehensive assessment of visual quality and data fidelity. Specifically, *DVSheet-Crea* evaluates *Reliability, Appropriateness, and Aesthetics*, while *DV-Evol* assesses *Integrity, Consistency, and Aesthetics* (see App. C.1 and C.4). The MLLM judge applies these expert rubrics to calculate the score: $S_{rubric}(\mathcal{O}, \mathcal{R}) = \frac{\sum_{k=1}^{N} s_k}{\sum_{k=1}^{N} w_k}$, $s_k = \Lambda(c_k, \mathcal{O}) \in [0, w_k]$. Data integrity is measured by *Table Coverage (TC)* with tolerance-aware matching: $S_{TC} = \frac{1}{N_{valid}} \sum_{c \in \mathcal{C}} \mathbb{I}(\text{match}(v_{gen}, v_{gt}))$, where match uses exact equality for non-numeric cells and numeric tolerance $(\epsilon, \delta)$ for floats (App. C.1). These components are integrated into the final scores: $S_{crea/evol} = w \cdot S_{rubric} + (1 - w) \cdot S_{TC}$, as shown in Fig. 10. Details are provided in App. C.

For tasks emphasizing operational correctness and agentic reasoning, the framework transitions to logic based appraisal and gold standards established by human experts. *DVSheet-Fix* adopts a strict *Success Rate (SR)* $SR_{DVSheet-Fix} = \mathbb{I}[\forall f \in \mathcal{F}_{must} : \text{Sim}(C_f, G_f) \geq \tau]$, $\tau \geq 0.95$, where $\text{Sim}(\cdot)$ compares native chart specifications on *must-fix* attributes (see App. C.2). *DVSheet-Dash* assesses *Insightfulness, Accuracy, Professionalism, and Aesthetics* via the score $S_{dash} = S_{rubric}(\mathcal{O}_{dash}, \mathcal{R}_{dash})$, $\mathcal{O}_{dash} = \mathcal{S}(I_1, \ldots, I_n)$ represents the stitched analytical view (see App. C.3). Finally, *DV-Inter* measures *Interaction, Accuracy, and Aesthetics* by utilizing a rubric alongside an *Interaction Success Rate(ISR)*, $ISR = (1 - \lambda) + \lambda \cdot \frac{N_{success} - N_{ref}}{N_{req} + 1}$, $\lambda = 0.5$, $N_{success}$ and $N_{ref}$ represent successfully resolved turns and inappropriate refusals, respectively. The final score is calculated as $S_{final} = S_{rubric} \cdot ISR$, where $S_{rubric}$ is the rubric score for the evaluation, and App. D.5 shows that model rankings are stable across reasonable values of $\lambda$. Further details are provided in App. C.5.

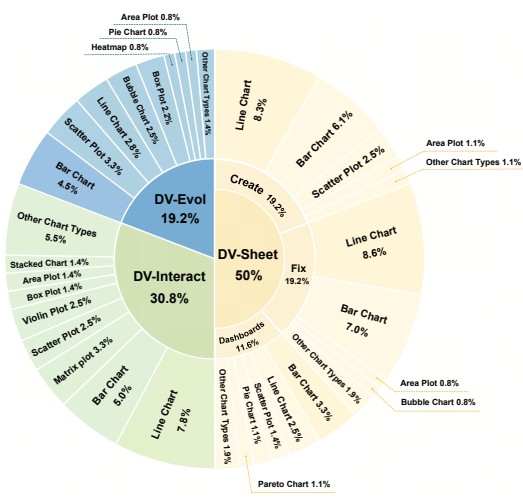

*Figure 2.* Chart-type distribution of DV-World.

## 2.3. Annotation Pipeline

**DV-World** is built through a rigorous pipeline executed by 18 visualization specialists, ensuring functional realism and semantic precision. Further details are provided in App. A.

**Data curation and adaptation.** Prioritizing authentic scenarios over synthetic datasets, we curate over 800 problem threads from communities such as ExcelForum and Kaggle (see App. A.1), collecting spreadsheets, tables, plotting code, and reference visualizations. To ensure robustness and privacy, we apply a three-step adaptation protocol: (i) structure retention that preserves merged cells and irregular layouts, (ii) value perturbation that renormalizes numbers while preserving distributions, and (iii) metadata anonymization that replaces identifiable entities with generic ones.

**Task design.** For generative tasks (*DVSheet-Crea* and *DVSheet-Dash*), we reduce annotator bias via a two-stage workflow: five experts standardize user questions, and seven others complete tasks based on the refined prompts. For *DVSheet-Fix*, experts inject common real-world errors (e.g., rigid axis scaling) into healthy workbooks to create diagnostic cases. For *DV-Evol*, experts verify semantic equivalence between paired implementations and author target code to preserve visual consistency. For *DV-Inter*, we construct interaction trajectories by starting from unambiguous tasks and systematically introducing controlled ambiguities. Experts then resolve these queries through clarification, and we record the resulting question–answer patterns and correction strategies. These logs define (i) hidden true intents that ground ambiguous requests and (ii) reaction rules that govern simulator feedback when agents make errors or fall into logic traps, enabling realistic multi-turn intent alignment.

**Evaluation construction.** We follow task-specific protocols to construct evaluations. For *DVSheet-Crea, DVSheet-*

*Table 2.* Key statistics for DV-World. Metrics represent per-example averages across different tasks. LOC, Prof., and Aesth. denote lines of code, Professionalism, and Aesthetics, respectively.

| Metric | Value |
|---|---|
| **Total** | 260 (100%) |
| - DV-Sheet tasks(Crea./Fix/Dash.) | 50 / 50 / 30 (50%) |
| - DV-Evol tasks | 80 (30.8%) |
| - DV-Inter tasks | 50 (19.2%) |
| - Chart Types | 51 |
| **DV-Sheet** | |
| # Question Tokens(Crea./Fix/Dash.) | 132.28 / 98.94 / 103.00 |
| # Columns / Rows / Sheet | 36.53 / 11583.36 / 3.59 |
| # Noise data (Fix) | 74.5% |
| # Chart Types (Crea./Fix/Dash.) | 24 / 21 / 26 |
| # Reliability / Appropriateness / Aesth. | 40% / 32% / 28% |
| # Insightfulness / Accuracy / Prof. / Aesth. | 40% / 25% / 20% / 15% |
| # Fix Types (Fix) | 12 |
| **DV-Evolution** | |
| # Question Tokens | 490.10 |
| # Columns / Rows | 58.98 / 52,584.58 |
| # Start LOC / Chart Types | 73.23 / 31 |
| # Integrity / Consistency / Aesth. | 25% / 37% / 38% |
| **DV-Interact** | |
| # Question Tokens / User Tokens | 74.62 / 734.54 |
| # Ambiguities per task / Ambiguity types | 3.17 / 15 |
| # Chart Types | 34 |
| # Columns / Rows | 12.04 / 14,030.04 |
| # Interaction / Accuracy / Aesth. | 24% / 41% / 35% |

*Dash, and DV-Inter*, experts design multidimensional rubrics (about 6 hours per rubric on average) and iterate through multiple verification rounds until consensus. To reduce false negatives, specialists additionally sample outputs from five models to ensure the rubrics cover diverse valid solution strategies. We also provide gold standards to improve precision: gold tables for *DVSheet-Crea* and *DV-Evol* to verify numerical accuracy, gold plots for *DV-Evol* as visual references, and the original healthy file (from reverse engineering) as ground truth for *DVSheet-Fix*.

### 2.4. Dataset Statistics

We present a statistical analysis of DV-World, highlighting its main features in comparison with prior datasets in Tab. 1, and providing characteristics in Tab. 2 and Fig. 2. DV-Sheet focuses on native grounding with workbooks averaging 36.53 columns and 11,583.36 rows, featuring 74.5% noisy data in Fix tasks and evaluation centered on Reliability (40%). DV-Evol covers cross-framework updates across 31 chart types and 52,584.58 rows, utilizing rubrics such as Integrity (25%) and Consistency (37%) to ensure design preservation. Finally, DV-Inter targets ambiguity resolution with 3.17 ambiguities per task and 734.54 user tokens, weighting Interaction (24%) and Accuracy (41%) to ensure intent alignment. Dataset statistics are provided in App. A.3.

## 3. Experiments and Analysis

### 3.1. Experimental Setup

We evaluate state-of-the-art LLMs, including open-source models like Qwen3 (Yang et al., 2025), Qwen3VL(Bai

*Table 3.* Detailed performance comparison on DV-Sheet. Scores include the mean and standard deviation ($\pm$) across evaluation trials.

| Method | Create | | | | | Fix | Dashboards | | | | | Score |
|---|---|---|---|---|---|---|---|---|---|---|---|---|
| | Reli. | Appro. | Aesth. | TC | Overall | SR | Insight. | Acc. | Prof. | Aesth. | Overall | |
| *SheetCopilot Baseline* | | | | | | | | | | | | |
| Gemini-3-Pro (Preview) | 33.76(±1.82) | 34.40(±3.15) | 24.31(±1.24) | 38.78 | 35.05(±2.11) | 44.00 | 27.73(±3.42) | 34.67(±1.95) | 37.33(±2.10) | 38.32(±3.21) | 32.97(±2.45) | 38.01(±1.85) |
| GPT-5.2 (2025-12-11) | 27.31(±1.54) | 35.33(±1.88) | 31.08(±3.45) | 31.45 | 31.19(±2.34) | 36.00 | 27.43(±1.35) | 36.31(±1.74) | 39.22(±3.26) | 36.43(±2.12) | 33.36(±2.18) | 33.54(±1.92) |
| DeepSeek-V3.2 | 31.56(±3.11) | 26.39(±1.42) | 27.37(±1.38) | 23.22 | 25.98(±1.86) | 36.00 | 33.37(±3.25) | 38.22(±2.41) | 36.11(±1.98) | 43.82(±3.54) | 36.70(±2.82) | 32.31(±2.15) |
| Qwen3-235B-A22B | 5.87(±0.64) | 15.55(±0.92) | 11.29(±0.75) | 13.49 | 11.99(±0.82) | 18.00 | 14.21(±0.84) | 22.64(±1.12) | 20.17(±0.95) | 16.04(±0.77) | 17.78(±1.05) | 15.64(±0.94) |
| *DV-World-Agent* | | | | | | | | | | | | |
| Gemini-3-Pro (Preview) | 34.71(±1.76) | 37.04(±3.32) | 27.26(±1.45) | 37.45 | 36.07(±2.18) | 48.00 | 31.79(±3.18) | 39.50(±2.04) | 36.22(±1.88) | 35.94(±3.10) | 35.29(±2.64) | 40.48(±2.12) |
| GPT-5.2 (2025-12-11) | 30.98(±1.62) | 37.64(±2.10) | 32.02(±3.55) | 34.99 | 34.43(±2.42) | 42.00 | 29.29(±1.54) | 37.48(±3.12) | 37.69(±1.95) | 38.39(±3.25) | 33.98(±2.31) | 37.24(±1.88) |
| DeepSeek-V3.2 | 30.10(±1.98) | 29.23(±1.56) | 26.49(±1.32) | 25.84 | 28.31(±1.74) | 36.00 | 33.94(±3.31) | 40.88(±3.55) | 33.19(±1.82) | 42.56(±3.48) | 36.35(±3.02) | 33.12(±2.45) |
| Kimi-K2-Thinking | 26.56(±3.05) | 29.99(±1.78) | 22.74(±1.44) | 26.11 | 26.94(±2.15) | 46.00 | 15.01(±1.12) | 14.92(±0.98) | 20.31(±3.24) | 28.17(±3.11) | 19.37(±2.08) | 32.52(±3.18) |
| GLM-4.7 | 4.37(±0.52) | 6.74(±0.68) | 7.39(±0.72) | 32.34 | 19.16(±1.22) | 40.00 | 34.35(±3.56) | 43.01(±3.68) | 29.47(±1.74) | 36.78(±3.42) | 38.03(±3.15) | 31.53(±2.95) |
| GPT-4.1 (2025-04-14) | 30.88(±1.68) | 39.00(±3.24) | 22.35(±1.44) | 29.01 | 29.98(±2.08) | 32.00 | 27.60(±1.48) | 28.09(±1.52) | 30.11(±3.16) | 40.50(±3.38) | 32.21(±2.44) | 31.27(±2.05) |
| Grok-4 | 18.25(±1.42) | 22.03(±3.55) | 16.68(±1.22) | 22.78 | 21.29(±1.65) | 42.00 | 25.13(±1.64) | 28.73(±1.72) | 25.92(±1.48) | 26.00(±2.85) | 27.78(±2.15) | 30.75(±2.24) |
| Gemini-2.5-Pro | 22.52(±3.58) | 30.35(±1.84) | 23.47(±1.42) | 25.10 | 25.75(±1.92) | 40.00 | 22.58(±1.35) | 27.44(±3.22) | 23.47(±1.28) | 24.22(±3.14) | 23.60(±2.05) | 30.73(±2.18) |
| Qwen3-Coder-Plus | 15.33(±1.12) | 16.21(±1.08) | 17.22(±3.15) | 17.93 | 17.04(±1.28) | 24.00 | 17.91(±0.98) | 16.00(±0.85) | 17.33(±0.92) | 20.67(±2.15) | 17.73(±1.44) | 19.87(±1.55) |
| Qwen3-235B-A22B | 7.29(±0.68) | 14.69(±3.18) | 13.41(±0.82) | 16.16 | 14.32(±1.05) | 22.00 | 15.33(±0.92) | 19.75(±1.10) | 22.29(±3.24) | 15.94(±1.85) | 17.01(±1.35) | 17.89(±1.42) |
| Qwen3-8B | 1.34(±0.22) | 2.26(±0.34) | 1.68(±0.28) | 1.32 | 1.52(±0.25) | 14.00 | 2.34(±0.38) | 4.44(±0.52) | 6.23(±0.64) | 5.12(±0.85) | 4.06(±0.72) | 6.91(±0.65) |
| Human | | | | | 80.81 | 88.00 | | | | | 87.34 | |

et al., 2025), GLM-4.7(Zeng et al., 2025), DeepSeek-V3.1 (DeepSeek-AI et al., 2025), and Kimi-K2 (Team et al., 2025), as well as proprietary ones such as the Gemini (Team et al., 2023), Grok(xAI, 2024) and GPT (OpenAI, 2023) families. For DV-Sheet, we compare against a spreadsheet agent, SheetCopilot (Li et al., 2023). For DV-Evol, we use OpenHands (Wang et al., 2024) as the primary baseline. We develop DV-World-Agent, a unified ReAct baseline (Yao et al., 2022) orchestrating `bash` and `load_image` tools, plus `render_chart` for multi-language rendering and `ask_user` for proactive alignment in DV-Evol (see details in App. D.2). We also conducted human evaluations, where a total of 10 evaluators completed 10 tasks for each domain. After completing the tasks, the evaluators scored each other's work based on the predefined rubrics (see details in App. D.4). For DV-Inter, we instantiate GPT-5-Mini as the user simulator. The performance of each agent is measured using the metrics detailed in §2.2 averaged over four independent runs. Unless otherwise specified, we set $w=0.5$ for DVSheet-Crea and DV-Evol to combine rubric scores with table-coverage signals, giving equal weight to visual quality and data fidelity as both are essential for usable visualizations, and use Gemini-2.5-Flash as the impartial MLLM-judge. A sensitivity analysis for this weighting is further detailed in App. D.5, confirming the stability of model rankings across various configurations. Further details are provided in App. D.

### 3.2. Main Results

**DV-Sheet results.** As shown in Tab. 3, native spreadsheet visualization presents a significant challenge for modern agents. Gemini-3-Pro leads the benchmark with a peak score of 40.48%, while even the most capable models such as GPT-5.2 and DeepSeek-V3.2 fail to exceed 38.00%. These results highlight a persistent performance gap compared to human

benchmarks and indicate a clear ceiling in native environment execution. The most prominent bottlenecks are found in the Fix and Dash tasks, where success rates and overall scores remain notably low. Such deficiencies underscore fundamental limitations in managing reliable data-to-chart binding, diagnostic error repair, and professional multi-view spatial planning within complex workbooks.

**DV-Evol results.** As shown in Tab. 4, chart evolution across diverse frameworks remains a significant challenge despite the use of specialized agentic orchestration. Gemini-3-Pro achieves the highest overall score of 51.44%, followed by Gemini-3-Flash at 49.46% and GPT-4.1 at 44.67%. Performance varies significantly by library, with top models demonstrating stronger proficiency in Python and Vega-Lite compared to the more complex requirements of D3.js and Plotly.js (shown in Fig. 4 (b)). Even the most capable systems remain far below the human baseline, highlighting a substantial gap in repository-level evolution. These results expose persistent limitations in cross-paradigm semantic transfer and design preservation, particularly when updates require coordinated modifications to data bindings, encodings, and underlying code structures.

**DV-Inter results.** Tab. 5 highlights the significant challenges of iterative intent alignment in DV-Inter. Grok-4 achieves a peak score of 40.43%, yet most top-tier models fail to reach 38.00%. This discrepancy underscores a fundamental weakness in proactive reasoning, as models frequently struggle to identify critical ambiguities or deliver the precise clarifications needed for professional results. These findings suggest that the primary bottleneck is the interactive process itself, which fails to bridge the gap between underspecified user intent and the complex logic required for accurate visualization.

*Table 4.* Main results on DV-Evol. Metrics include MLLM-Score (MS), Table Coverage (TC), and weighted Overall scores.

| Method | Python | | | Apache ECharts | | | Vega-Lite | | | D3.js | | | Plotly.js | | | Score |
|---|---|---|---|---|---|---|---|---|---|---|---|---|---|---|---|---|
| | MS | TC | Overall | MS | TC | Overall | MS | TC | Overall | MS | TC | Overall | MS | TC | Overall | |
| *Openhands Baseline* | | | | | | | | | | | | | | | | |
| Gemini-3-Pro (Preview) | 49.87(±2.15) | 64.32 | 57.10(±1.64) | 29.44(±3.12) | 65.47 | 47.46(±2.21) | 22.45(±1.85) | 57.11 | 39.78(±1.42) | 34.11(±3.24) | 67.45 | 50.78(±2.35) | 28.76(±2.10) | 64.21 | 46.49(±1.68) | 48.32(±2.12) |
| GPT-5.2 (2025-12-11) | 46.21(±1.98) | 66.44 | 56.33(±1.52) | 31.11(±3.25) | 54.98 | 43.05(±2.34) | 21.22(±2.64) | 56.33 | 38.78(±1.35) | 22.14(±1.95) | 57.22 | 39.68(±3.62) | 23.11(±1.82) | 54.22 | 38.67(±1.44) | 43.30(±1.95) |
| Gemini-2.5-Pro | 33.33(±2.45) | 40.67 | 37.00(±1.88) | 20.37(±3.18) | 55.32 | 37.85(±2.42) | 17.11(±1.55) | 33.49 | 25.30(±1.15) | 27.33(±3.41) | 52.41 | 39.87(±2.65) | 18.78(±1.65) | 54.56 | 36.67(±1.58) | 35.34(±2.24) |
| Qwen3-VL-Plus | 36.56(±3.12) | 44.54 | 40.55(±2.31) | 17.29(±1.88) | 34.23 | 25.76(±1.45) | 23.34(±2.10) | 30.45 | 26.90(±1.52) | 16.22(±1.68) | 33.39 | 24.81(±1.38) | 13.54(±1.45) | 38.32 | 25.93(±1.32) | 28.79(±1.88) |
| *DV-World-Agent* | | | | | | | | | | | | | | | | |
| Gemini-3-Pro (Preview) | 52.42(±2.15) | 68.29 | 60.36(±1.62) | 27.24(±3.45) | 61.67 | 44.45(±2.31) | 26.92(±1.92) | 65.69 | 46.30(±1.48) | 39.98(±3.58) | 72.71 | 56.34(±2.45) | 29.48(±2.12) | 70.04 | 49.76(±4.72) | 51.44(±2.18) |
| Gemini-3-Flash | 47.88(±1.82) | 69.54 | 58.54(±1.44) | 23.58(±3.12) | 68.43 | 46.01(±2.15) | 25.59(±1.75) | 65.19 | 45.39(±3.32) | 31.60(±3.34) | 68.05 | 49.83(±2.21) | 26.06(±1.95) | 69.03 | 47.54(±1.58) | 49.46(±2.05) |
| Grok-4 | 48.23(±2.45) | 59.44 | 53.84(±1.52) | 24.77(±3.21) | 65.21 | 44.99(±2.24) | 33.13(±3.56) | 68.23 | 50.68(±3.18) | 29.67(±2.12) | 69.21 | 49.44(±1.75) | 27.91(±1.88) | 62.31 | 45.11(±1.52) | 48.81(±2.14) |
| GPT-4.1 (2025-04-14) | 41.16(±1.74) | 33.90 | 37.53(±1.42) | 17.81(±1.68) | 70.16 | 43.98(±1.58) | 21.46(±1.62) | 70.88 | 46.17(±3.48) | 20.10(±2.84) | 71.70 | 45.90(±1.55) | 25.39(±2.15) | 74.11 | 49.75(±1.78) | 44.67(±1.92) |
| GPT-5.2 (2025-12-11) | 42.92(±1.88) | 68.69 | 55.81(±1.45) | 21.58(±3.12) | 58.06 | 39.82(±2.15) | 25.00(±1.78) | 59.59 | 42.29(±1.42) | 15.49(±1.54) | 61.52 | 38.51(±1.48) | 16.51(±1.62) | 61.04 | 38.78(±1.42) | 43.04(±1.85) |
| Gemini-2.5-Pro | 35.97(±2.31) | 42.44 | 39.21(±1.75) | 18.16(±1.95) | 57.53 | 37.85(±1.68) | 16.42(±1.54) | 37.65 | 27.04(±1.22) | 28.66(±3.48) | 56.65 | 42.65(±2.55) | 16.75(±1.68) | 56.01 | 36.38(±1.44) | 36.63(±2.15) |
| Qwen3-VL-Plus | 37.99(±3.15) | 40.23 | 39.11(±2.24) | 19.81(±1.82) | 30.23 | 25.02(±3.45) | 20.43(±1.78) | 35.62 | 28.02(±1.35) | 13.23(±2.55) | 39.12 | 26.17(±2.38) | 14.32(±1.42) | 42.33 | 28.32(±1.32) | 29.33(±1.92) |
| Qwen3-VL-32B | 39.12(±3.24) | 33.12 | 36.12(±2.18) | 13.21(±1.62) | 22.34 | 17.77(±2.28) | 23.12(±3.85) | 29.31 | 26.21(±1.35) | 13.23(±1.48) | 28.23 | 20.73(±3.28) | 9.03(±1.15) | 31.03 | 20.03(±1.08) | 24.17(±1.75) |
| Qwen3-VL-8B | 33.14(±3.12) | 32.86 | 33.00(±2.10) | 9.32(±1.15) | 21.09 | 15.21(±1.02) | 14.86(±1.42) | 31.67 | 23.27(±1.18) | 6.13(±0.95) | 26.77 | 16.45(±0.98) | 5.40(±0.82) | 29.25 | 17.32(±0.88) | 21.05(±1.52) |
| Human | | 85.23 | | | 82.11 | | | 88.46 | | | 85.21 | | | 84.44 | | |

*Table 5.* Main results on DV-Inter. We report MLLM-Scores across interaction quality and visual metrics, alongside Interaction Success Rate (ISR) and average User Cost per task.

| Method | MLLM-Score | | | | ISR | Score | User Cost |
|---|---|---|---|---|---|---|---|
| | Interaction | Accuracy | Aesthetics | Overall | | | |
| *DV-World-Agent* | | | | | | | |
| Grok-4 | 69.07(±2.12) | 61.29(±1.88) | 27.19(±3.15) | 51.10(±2.24) | 79.57 | 40.43(±1.95) | $0.051 |
| DeepSeek-V3.2 | 69.33(±2.34) | 61.31(±1.95) | 29.03(±3.24) | 51.30(±2.15) | 74.05 | 37.94(±2.05) | $0.032 |
| GPT-5.2 (2025-12-11) | 59.43(±1.82) | 60.98(±2.10) | 32.99(±3.41) | 50.58(±1.92) | 69.25 | 35.09(±1.88) | $0.021 |
| Gemini-3-Pro (Preview) | 66.43(±2.05) | 58.16(±1.76) | 31.62(±3.18) | 50.08(±2.15) | 66.83 | 34.43(±2.12) | $0.017 |
| Gemini-2.5-Pro | 65.33(±3.12) | 52.09(±1.68) | 24.93(±1.42) | 45.78(±2.31) | 69.42 | 31.34(±2.24) | $0.018 |
| GLM-4.7 | 57.37(±1.95) | 54.40(±2.15) | 31.83(±3.32) | 46.36(±1.85) | 62.33 | 29.57(±2.08) | $0.013 |
| Kimi-K2-Thinking | 55.93(±3.21) | 52.93(±1.54) | 25.12(±1.35) | 43.19(±2.42) | 63.67 | 27.39(±2.55) | $0.024 |
| GPT-4.1 (2025-04-14) | 55.13(±1.64) | 47.94(±1.42) | 21.29(±1.28) | 39.56(±1.76) | 64.58 | 25.68(±1.92) | $0.013 |
| Qwen3-235B-A22B | 41.23(±1.15) | 35.80(±1.05) | 14.23(±0.92) | 29.06(±1.22) | 74.62 | 20.90(±1.15) | $0.032 |
| Qwen3-8B | 48.03(±1.08) | 30.04(±0.92) | 14.92(±0.85) | 22.36(±1.15) | 62.39 | 18.09(±1.24) | $0.024 |
| Human | | | | | 79.60 | | |

### 3.3. Analysis of Native Grounding in Spreadsheet

**Challenges in native object model mastery.** Analysis of model performance on DV-Sheet highlights key challenges across task categories. Figure 3(a) shows a positive correlation between Table Coverage and Visual Aesthetics for the *DVSheet-Crea* task, with models like Gemini-3-Pro and GPT-5.2 performing better aesthetically as their table coverage improves, emphasizing the importance of data grounding for effective visualization. In Fig. 4(b), success rates across different fix categories show that while agents excel at resolving filtering logic, they struggle more with axis scaling and encoding errors that require precise geometric mapping. Finally, Fig. 4 illustrates that as table size increases in the *DVSheet-Dash* task, performance declines, confirming that large datasets present significant reasoning challenges and affect the consistency of dashboard layouts.

**Error analysis.** As shown in Fig. 5, errors in the DV-Sheet task are categorized into four areas: Data Accuracy, Layout Readability, Chart Design, and Format Compliance. In *DVSheet-Crea*, Data Accuracy accounts for over 50% of errors, followed by Layout Readability at around 19%. In *DVSheet-Fix*, data accuracy is the dominant issue (69%), with smaller contributions from Format Compliance. In *DVSheet-Dash*, Data Accuracy makes up nearly 46%, with Visual Design and Rigor & Completeness contributing to the remaining errors. These results highlight the need to

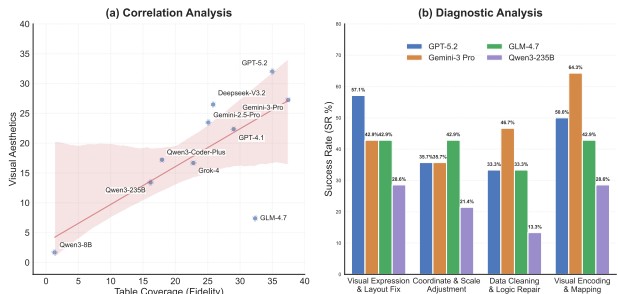

*Figure 3.* (a) shows the correlation between Table Coverage and Visual Aesthetics for the *DVSheet-Crea* task. (b) shows the Success Rate (SR%) for fix categories in the *DVSheet-Fix* task.

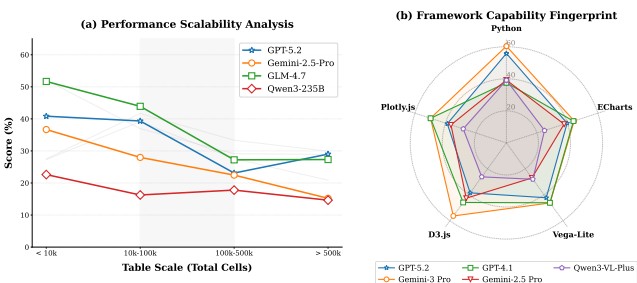

*Figure 4.* (a) Performance scores (%) for the *DVSheet-Dash* task across varying table scales (cells). (b) The performance of the five visualization frameworks in the *DV-Evol* task.

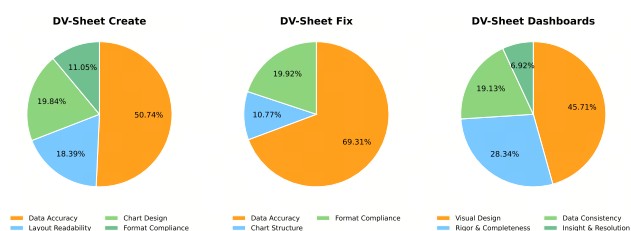

*Figure 5.* Error analysis of the *DV-Sheet* task.

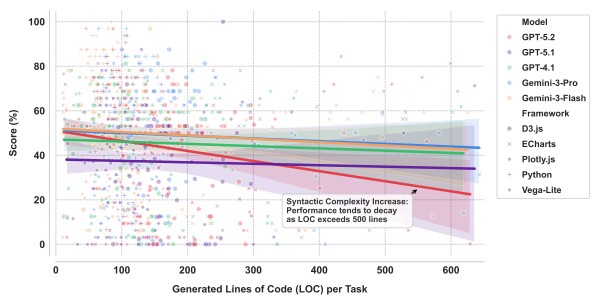

*Figure 6.* Performance decay with respect to Lines of Code (LOC).

improve data handling and layout design in autonomous visualization. Further error analysis are provided in App. H.1.

### 3.4. Analysis of cross-paradigm evolutionary adaptation

*Table 6.* Ablation study of visual feedback across different models. Values represent the success rate (%), and Δ denotes the performance drop compared to the full model.

| Model Settings | Python | ECharts | Vega-Lite | D3.js | Plotly.js |
|---|---|---|---|---|---|
| **Gemini-3-Pro** | **60.36** | **44.45** | **46.31** | **56.34** | **49.76** |
| − w/o load_image tool | 58.11 (↓2.25) | 41.32 (↓3.13) | 46.28 (↓0.03) | 48.65 (↓7.69) | 48.42 (↓1.34) |
| **GPT-5.2** | **55.81** | **39.82** | **42.29** | **38.51** | **38.78** |
| − w/o load_image tool | 54.64 (↓1.17) | 39.21 (↓0.61) | 40.12 (↓2.17) | 37.21 (↓1.30) | 35.31 (↓3.47) |
| **Qwen3-VL-Plus** | **39.11** | **25.02** | **28.03** | **26.18** | **28.33** |
| − w/o load_image tool | 35.22 (↓3.89) | 25.45 (↑0.43) | 26.65 (↓1.38) | 22.21 (↓3.97) | 23.91 (↓4.42) |

**Performance decay with loc and visual feedback.** As shown in Fig. 6, performance decays consistently as target Lines of Code increase, suggesting that the high syntactic density of frameworks like D3.js creates a substantial reasoning burden. This verbosity tax is further explored in Tab. 6, which quantifies the role of the `load_image` tool as a critical compensatory mechanism. Ablation results demonstrate that removing this tool leads to a universal decline in success rates, with Gemini-3-Pro suffering a peak attrition of 7.69% in D3.js tasks. These findings indicate that while API verbosity imposes a tax on reasoning stability, the `load_image` tool is essential for maintaining semantic fidelity in high-complexity visualization evolution.

**Error analysis.** Error profiles in DV-Evol are highly dependent on framework abstraction. Low-level libraries like D3.js challenge agents with complex rendering logic, exemplified by Gemini-3-Flash reaching a 40.96% visual style error rate. Conversely, declarative frameworks such as Apache ECharts and Vega-Lite expose weaknesses in data mapping and organization. Specifically, Gemini 3 Pro records 45.20% for data consistency issues in ECharts and Gemini-3-Flash reaches 51.80% for layout errors in Vega-Lite. These framework-specific failures indicate that verbose APIs primarily degrade styling accuracy, whereas high-level tools test the ability of an agent to preserve logical data-to-visual structures. Further error analysis are provided in App. H.2.

### 3.5. Analysis of interactive agency and alignment

**Strategic proactivity and interaction efficiency.** As shown in Fig. 7, transitioning to interactive alignment offers significant benefits, but the gains depend on proactive reasoning quality. Gemini-3-Pro shows the highest improvement with a 23.0% performance gain and the strongest correlation between clarification and task quality. In contrast, models like Grok-4 and DeepSeek-v3.2 have high interaction frequencies that don't always lead to effective results, highlighting that the quality of questions matters more than the number of interactions. Weaker models show performance loss, suggesting that poorly targeted interactions add noise, with identifying critical ambiguities as the key challenge for achieving professional-grade visualizations.

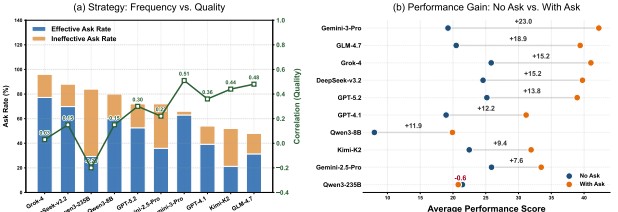

*Figure 7.* Comparison of ask strategies and performance: (a) effective vs. ineffective ask rates and task quality; (b) performance with vs. without asking.

**Error analysis.** As summarized in Tab. 29, error distribution in DV-Inter identifies Cognitive Execution Gap and Interactive Avoidance as primary bottlenecks. The Execution Gap is prominent in GPT-5.2 at 60.07%, where successful intent alignment fails to produce grounded results. Interactive Avoidance in GLM-4.7 at 59.47% stems from an overconfidence bias that leads to simplified task downgrades. These failures highlight a persistent struggle to bridge the gap between clarified user intent and technical execution. Further details are provided in App. H.2.

## 4. Meta-Evaluation of the Framework

### 4.1. User Simulator Analysis

**Impact of simulator intelligence on agent performance and cost.** Fig. 8 shows that the performance of agents is primarily driven by the intelligence of the simulator. The GPT-5 series leads, with GPT-5-mini being the optimal user simulator. It offers high instructional fidelity while reducing operational costs, setting the efficiency frontier. While other simulators are useful for stress tests or average complexity, GPT-5-mini provides the most stable and consistent environment to measure the interactive agent's capability ceiling. Further analysis are provided in App. E.1.1.

**Alignment with human user.** We manually audit 150 interaction trajectories to assess the alignment between our

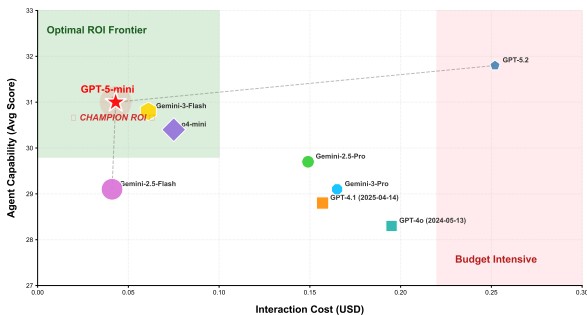

*Figure 8.* Comparison of agent capability and interaction cost.

simulators and human behavior, focusing on response reliability and sensitive information prevention. We quantify this by computing the Pearson correlation between human ratings and simulator performance. As shown in Table 7, the GPT-5-mini simulator achieves 88.67% Faithfulness and a 0.86 Pearson correlation ($p < 0.04$), indicating strong alignment with human judgment. Ablation results show that removing Reaction Rules or Stage 1 Filtering significantly reduces this alignment, leading to less reliable and potentially insecure interactions. Further details in App. E.1.1.

*Table 7.* Ablation study of simulator backbones and components. Fidelity measures consistency with $\mathcal{I}_{gt}$, while Pearson $\rho$ reflects alignment with human trajectories.

| Simulator Model | Faithfulness ↑ | Pearson $\rho$ ↑ | *p*-value |
|---|---|---|---|
| GPT-5-mini (Ours) | 88.67 | 0.86 | < 0.04 |
| – w/o Reaction Rules | 84.00 | 0.78 | < 0.05 |
| – w/o Stage 1 Filtering | 85.33 | 0.80 | < 0.05 |
| Gemini-3-Flash | 86.00 | 0.81 | < 0.02 |
| – w/o Reaction Rules | 82.66 | 0.76 | < 0.03 |
| – w/o Stage 1 Filtering | 82.00 | 0.74 | < 0.06 |
| O4-mini | 85.33 | 0.83 | < 0.03 |
| – w/o Reaction Rules | 80.67 | 0.73 | < 0.01 |
| – w/o Stage 1 Filtering | 79.33 | 0.69 | < 0.05 |

### 4.2. Validation of MLLM-Judge Method

**Human-model alignment.** We further assess the reliability of the MLLM-as-a-judge protocol on a pooled set of 210 tasks spanning the four MLLM-judged settings (Tab. 8). Human annotators achieve strong agreement, with ICC(A,1)=0.932 and weighted $\kappa = 0.903$, indicating that the rubric is sufficiently well specified to support consistent expert judgment and providing a practical upper bound for automated judges. Among the evaluated MLLM judges, Gemini-2.5-Flash shows the strongest alignment with human ratings ($\kappa = 0.821$, ICC(A,1)=0.850), while the remaining judges also maintain relatively high item- and case-level agreement. Overall, these results suggest that our rubric-based evaluation is stable, human-aligned, and not overly dependent on a single judge model.

**Cross-judge consistency.** To mitigate family-specific bias and ensure leaderboard reproducibility, we evaluate multiple proprietary judges on the same 210 tasks. In Tab. 8 and Tab. 9, relative rankings are largely preserved across judge families, though scores may shift. We standardize on Gemini-2.5-Flash as the primary judge due to its strongest item- and case-level alignment with human experts.

*Table 8.* Validation of the hybrid evaluation framework. We report agreement metrics across the four MLLM-judged tasks (Crea, Dash, Evol, Inter). (Details in App. E.1.2)

| Judge Model | Pooled Tasks (*N*=210 tasks, 3k+ items) | | |
|---|---|---|---|
| | Item-level (Weighted $\kappa$) | Case-level (ICC(A,1)) | Model-level (Kendall's $\tau_b$) |
| Human-Human | 0.903 | 0.932 | 1.000 |
| Gemini-2.5-Flash | 0.821 | 0.850 | 1.000 |
| Gemini-3-Flash | 0.815 | 0.842 | 1.000 |
| Gemini-2.5-Pro | 0.801 | 0.844 | 1.000 |
| GPT-4.1 | 0.791 | 0.831 | 1.000 |
| GPT-4o | 0.778 | 0.799 | 0.800 |

*Table 9.* Ranking stability across MLLM judges. **Overall DV Score** excludes DVSheet-Fix (rule-based).

| Agent Model | Primary | Alternative Judges | | | |
|---|---|---|---|---|---|
| | Flash | Pro | GPT-4.1 | Gemini-3-Flash | GPT-4o |
| GPT-5.2 | 39.12 | 38.69 | 42.01 | 38.82 | 40.82 |
| Gemini-3-Pro | 43.74 | 42.27 | 45.44 | 41.58 | 44.54 |
| Gemini-2.5-Pro | 33.13 | 31.92 | 33.88 | 32.17 | 32.02 |
| GPT-4.1 | 32.82 | 30.76 | 32.99 | 31.03 | 33.10 |
| Qwen3(-VL)-8B | 13.41 | 14.57 | 16.16 | 14.51 | 15.75 |

## 5. Related Work

**Agentic benchmarks.** Agentic benchmarks increasingly evaluate LLM-based agents in execution-grounded and interactive settings, emphasizing long-horizon tool use and adaptation to feedback (Zhou et al., 2024; Koh et al., 2024; Pan et al., 2024; Xu et al., 2024; Davydova et al., 2025; Deng et al., 2024; Wang et al., 2025; Bytedance Seed, 2026). Representative suites cover repository-level software engineering and requirement-driven code generation (Jimenez et al., 2023; Ding et al., 2025; Si et al., 2025; Zhu et al., 2025; Lu et al., 2025; Zhang et al., 2025a), realistic SQL-/data workflows (Lei et al., 2024; Li et al., 2025a; Lei et al., 2025), and interactive protocols that stress intent clarification and recovery (Huo et al., 2025; Yao et al., 2024; Barres et al., 2025; Mu et al., 2024; He et al., 2025; Lù et al., 2025). However, these benchmarks remain largely domain-general and do not explicitly model professional data-visualization workflows, where underspecified intent and design choices are central.

**Benchmarks for data visualization.** Data-visualization benchmarks mainly study single-shot translation from natural language to visualization specifications or plotting code (Chen et al., 2024b; Luo et al., 2025; Rahman et al., 2025), complemented by chart understanding/editing and chart-

to-code generation (Zhang et al., 2025b; Wu et al., 2025; Tang et al., 2025; Yang et al., 2024a). Spreadsheet benchmarks evaluate long-horizon table manipulation and control (Payan et al., 2023; Ma et al., 2024; Li et al., 2023; Chen et al., 2025b; Lei et al., 2026). Yet few benchmarks evaluate spreadsheet-native chart/dashboard creation with interactive intent alignment and diagnostic repair, motivating our benchmark.

## 6. Conclusion

The **DV-World** benchmark provides an evaluation suite for data visualization (DV) agents, targeting real-world workflows beyond traditional benchmarks. It assesses performance across DV-Sheet, DV-Evol, and DV-Inter, covering spreadsheet-centric tasks (e.g., chart creation and debugging), cross-paradigm visualization evolution, and multi-turn interaction under ambiguous user intent. Results show that even strong models (e.g., Gemini-3-Pro and GPT-5.2) still struggle with error correction, faithful data binding, and consistent evolution. By combining rubric-based judgments with quantitative checks and interaction success metrics, DV-World enables diagnosis and progress tracking for end-to-end DV workflows. More broadly, DV-World establishes a standardized yardstick for the community to quantify and accelerate progress toward reliable DV agents.

## Acknowledgements

This work was supported by Beijing Natural Science Foundation (L243006), the National Natural Science Foundation of China (No.62376270) and the independent research project of the Key Laboratory of Cognition and Decision Intelligence for Complex Systems.

## Impact Statement

This paper presents work aimed at advancing the field of data visualization agents through the development of the DV-World benchmark. The benchmark is designed to improve the performance of autonomous visualization systems, particularly in complex, real-world tasks across various domains. By enabling more robust and adaptive agents, this work has the potential to significantly enhance data-driven decision-making in professional environments, such as healthcare, finance, and research. The ability to create reliable, interactive, and visually insightful tools can drive innovation, improve accessibility, and support better-informed decisions at scale.

While this research focuses on enhancing system capabilities, it is important to consider the broader societal implications of autonomous visualization agents, including ensuring their fairness, transparency, and ethical use. As

these systems are adopted across industries, responsible development will be key to maximizing their benefits and addressing challenges related to data privacy and security. This work aims to contribute positively to the future of AI-assisted visualization, and we are committed to furthering its potential in a responsible and impactful manner.

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

# A. Task Construction Details

## A.1. Data Sources

Our data sources cover both *community Q&A* (forums/blogs) and *open datasets*, capturing real-world spreadsheet questions, answers, and data tables.

**Kaggle Datasets**[1] is an open repository where users publish and share machine-learning-ready tables. At the time of access, Kaggle reports *643K* high-quality public datasets and *29M+* community users spanning *190+* countries, making it a large-scale source of diverse tabular data for analysis and visualization tasks.

**Chandoo.org**[2] is a long-running Excel/BI learning site that mixes tutorials with an active Q&A-style community. The site reports substantial audience scale (e.g., *250K* monthly views) and community reach (e.g., *100K+* newsletter readers), and also notes a large video footprint (e.g., *870K+* YouTube subscribers as of June 2025). These characteristics make it a practical source of realistic Excel charting, reporting, and dashboarding scenarios.

**Excelguru**[3] provides professional resources on Excel and the Power platform (Power Query / Power BI), along with an open help forum. Forum category statistics indicate non-trivial interaction volume (e.g., the *Formulas* board lists about *4.5K* threads and *21.1K* messages), which is useful for collecting naturally occurring question patterns involving formulas, pivots, programming, and dashboards.

**ExcelForum**[4] is a large Excel help forum with fine-grained sub-forum organization and high activity. Its live forum statistics report approximately *1,032,088* threads, *5,116,544* posts, and *1,405,242* registered members, providing a broad coverage of real spreadsheet issues (formulas, VBA, charting, Office integrations) with extensive peer-produced answers.

**MrExcel**[5] is a widely recognized Excel community centered around its message board and educational content. Public third-party summaries of the site describe the forum as exceeding *one million* threads and around *five million* posts, reflecting a comparable scale of organically generated Excel questions and solutions. This makes it a complementary source to ExcelForum, especially for capturing real user language and troubleshooting behaviors at scale.

## A.2. Construction Process

### A.2.1. DATA ANNOTATION: DVSHEET-CREA

For the *DVSheet-Crea* task, we implement a systematic three-stage annotation pipeline: (1) Categorized Data Selection, (2) Instruction & Solution Generation, and (3) Quantitative Rubric Design. We aim to construct a diverse benchmark of 80 distinct test cases that evaluate an agent's ability to generate auditable, native Excel charts.

**(1) Categorized data selection.** We aim to select tabular data that reflects real-world spreadsheet complexity. Sourced from expert forums (e.g., *ExcelForum*, *MrExcel*) and open repositories (e.g., *Kaggle*), the dataset is organized into three complexity levels to test different agent capabilities: *1) Direct Visualization*, where the target chart can be plotted directly from the input range without modification. This subset is further divided into *Specified Requests* where chart types are explicit, and *Open-Ended Requests* where the agent must infer the optimal visualization; 2) Extraction-Required, involving large-scale tables where the agent must first aggregate or filter data to produce an intermediate summary table before visualization; and 3) Multi-Sheet Reasoning, which requires synthesizing data distributed across multiple worksheets, addressing a common challenge in complex spreadsheet workflows.

**(2) Instruction & solution generation.** We aim to verify that agents can handle both explicit constraints (e.g., "Create a scatter plot with Income on X...") and abstract user intents (e.g., "Help me understand the energy composition..."). For the ground truth, we do not rely on static rasterized images. Instead, we utilize Python scripts (via libraries such as `openpyxl` or `xlsxwriter`) to generate **native Excel chart objects**. This ensures the solution is a live file with correct data bindings and series definitions, serving as the gold standard for auditability and reproducibility.

**(3) Quantitative rubric design.** We aim to ensure objective evaluation through a strict, evidence-based rubric. Instead

---

[1] https://www.kaggle.com/datasets
[2] https://chandoo.org/
[3] https://excelguru.ca/
[4] https://www.excelforum.com/
[5] https://www.mrexcel.com/

of subjective visual scoring, we establish a "Quantitative Constraints" framework that evaluates three dimensions using binary checks: *1) Fidelity(Data Fidelity)*, verifying if the chart accurately reflects the source data without hallucination (e.g., "Does the chart strictly exclude data from year $< 2020$?" or "Is the max value for 'East' visually $\approx 1.2M$?"); *2) Logic(Visualization Logic)*, checking if the chosen chart type and data organization align with the instruction (e.g., "Is the data sorted in descending order of revenue?" or "Is the X-axis mapped to the correct date column?"); and *3) Aesthetics(Presentation Quality)*, assessing professional layout standards (e.g., "Are axis labels non-occluded?" and "Is the title descriptive and keyword-rich?"). Each case is assigned a specific rubric derived from its unique data constraints, ensuring a granular and fair assessment.

### A.2.2. DATA ANNOTATION: DVSHEET-FIX

Complementing the creation task, the *DV-Sheet Fix* (50 cases) evaluates an agent's diagnostic and remedial capabilities. We simulate a human-in-the-loop debugging scenario where the agent is presented with a defective spreadsheet and a user complaint, and must restore the visualization to a functional and professional state.

**(1) Error taxonomy & sourcing.** To ensure real-world relevance, we first derived a taxonomy of common visualization failures by analyzing user queries from communities like *ExcelForum* and *MrExcel*. We categorized these failures into three distinct classes: 1) Data Binding Errors, the most prevalent failures where the chart object references incorrect ranges, including *Series/Category Inversion* (e.g., plotting years as data series instead of axes), *Header Inference Failures*, and *Phantom Data* (e.g., flat lines caused by including empty cells or hidden zeros); 2) Axis & Scaling Issues, covering structural issues that render charts unreadable, such as *Date vs. Text Axis Mismatches* (causing date gaps), *Fixed Bound Deadlocks* (manual min/max limits obscuring data), and *Log-Scale Errors* with negative values; and 3) Type Mismatch & Visual Clutter, involving semantic errors where the chosen chart type conflicts with the data nature (e.g., using stacked bars for non-additive growth rates) or aesthetic failures involving excessive cognitive load.

**(2) Dataset construction: inverse injection strategy.** While we gather real-world samples, relying solely on raw forum files makes obtaining a perfect Ground Truth difficult. Therefore, we adopt a Controlled Inverse Injection approach. We start with high-quality, verified charts (from the *Create* task or expert curation) serving as the *Gold Standard*. We then programmatically or manually inject specific defects based on our taxonomy (e.g., forcing a *Row/Column Switch* or hard-coding an invalid *Axis Limit*) to generate the *Input State*. This method ensures that every problem has a deterministic, mathematically proven repair path. The accompanying user queries are standardized into natural language complaints (e.g., "Why is my line chart flat?" or "Fix the crowded axis").

**(3) Evaluation: state-based property verification.** Unlike generation tasks that allow for creative variance, repair tasks often have binary success criteria. We employ a Hybrid Rule-Based Evaluation system: 1) Property Inspection (Primary), utilizing Python (via `openpyxl`/`xlwings`) to inspect the internal state of the fixed chart object. For instance, to verify a Flat Line fix, the rubric checks if the `chart.plot_by` attribute has been correctly toggled or if the `axis.scaling.max` is reset to 'Auto'. This direct XML-level inspection is more reliable than visual similarity metrics; and 2) Side-Effect Checks (Secondary), performing sanity checks (e.g., ensuring data variance $> 0$) to confirm the chart remains visually meaningful and preventing over-correction.

### A.2.3. DATA ANNOTATION: DVSHEET-DASH

The *DVSheet-Dash* benchmark (30 tasks) represents an advanced stage of spreadsheet automation, requiring agents to combine multiple visualizations into cohesive, interactive interfaces. Unlike single-chart tasks, dashboards demand high-level planning, layout logic, and the implementation of interactive controls such as Slicers. The annotation process follows a three-stage pipeline.

**(1) Hybrid reference construction.** Constructing a high-quality Gold Standard dashboard is difficult. We employ a hybrid workflow to ensure quality: 1) Data Selection, selecting multi-dimensional datasets (e.g., Sales, Inventory) capable of supporting hierarchical analysis; 2) Baseline Prototyping, using coding agents to generate a Python script (via `openpyxl`) that constructs the initial layout and binds Slicers; and 3) Expert Refinement, where humans polish the baseline to ensure a clear business story. This results in a final file complete with calculated KPI cards and interactive elements, serving as the ground truth.

**(2) Instruction generation via user archetypes.** To verify agent robustness across different communication styles, we categorize instructions into three archetypes: 1) Executive Directives (Formal Email), prioritizing business value where agents must focus on storytelling (e.g., trend analysis) rather than aesthetic details; 2) Technical Specifications (Project Requirements), focusing on visual and functional constraints akin to formal requirement documents (e.g., "No gridlines") to test strict adherence; and 3) Ad-hoc Queries (Instant Messaging), centering on intent inference where agents must deduce necessary metrics from vague requests without clear guidance.

**(3) Dynamic rubric formulation.** To standardize evaluation, we design a task-specific Rubric that maps requests to binary criteria across four mandatory dimensions: *1) Insight*, checking if the dashboard answers the core business questions (e.g., "Are KPIs visible?"); *2) Consistency*, ensuring calculation integrity by matching the aggregation logic in the code with the ground truth; *3) Visual*, evaluating structural logic such as appropriate chart selection and clean grid alignment; and *4) Professionalism*, verifying the output is client-ready, including correct currency formatting and no overlapping elements.

### A.2.4. DATA ANNOTATION: DV-EVOL

The *DV-Evol* benchmark (80 tasks) targets the critical capability of Visual Refinement. Unlike generation from scratch, real-world workflows often involve modifying existing charts to accommodate new data or changing requirements. We construct these tasks via a rigorous pipeline designed to ensure robustness and recoverability.

**Atomic subplot isolation.** The process begins by sourcing high-complexity composite figures (e.g., $3 \times 3$ trellis plots or faceted grids) from open benchmarks. To create focused editing tasks, we programmatically decompose these composites into atomic visualization units. We extract the plotting logic for a single subplot (e.g., the top-left panel) into a standalone script, ensuring the starting point is a high-quality, executable chart with complex styling, rather than a trivial default plot.

**Adversarial data perturbation.** To prevent agents from bypassing logic via memorization, we employ a data mocking strategy with two distinct perturbation modes. First, we apply *Schema-Preserving Shifts*, where we retain column names but statistically alter the distribution (e.g., introducing outliers, shifting means, or injecting long-tail noise) to enforce reliance on actual values. Second, we introduce *Schema Mutation*, where we rename key semantic columns (e.g., changing "Revenue" to "Total_Sales") and simultaneously update the ground truth code to test the agent's ability to adapt scripts to evolving data structures.

**Taxonomy-based task design.** We simulate realistic user feedback by sampling edit operations from a structured taxonomy. Each case combines commands from four categories: *DataOps* for modifications requiring re-aggregation or filtering (e.g., "Change Mean to Median" or "Show only Top-5"); *EncodeOps* for changes to visual mappings (e.g., "Swap X/Y axes" or "Map 'Size' to Profit"); *LayerOps* for adding annotation layers (e.g., "Add a threshold line at $y = 100$"); and *StyleOps* for aesthetic adjustments (e.g., "Change color to `#FF5733`" or "Format axis ticks as percentages").

**Executable verification.** Finally, for each task, we generate a Ground Truth tuple containing the refined script, the final rendered image, and metadata. Crucially, the verification pipeline enforces a strict "No-Hardcode" policy: the solution code must derive all visual elements (such as bar heights or text labels) dynamically from the input data rather than using static constants. This guarantees that the agent's solution is robust, reusable, and mathematically sound.

### A.2.5. DATA ANNOTATION:DV-INTER

The *DV-Inter* benchmark evaluates an agent's ability to navigate vague user requests through multi-turn dialogue. Unlike single-turn tasks, success requires identifying ambiguity, soliciting clarification, and refining visualizations based on feedback. We construct these samples via a rigorous five-stage pipeline.

**Ambiguity injection and complexity scaling.** Starting with a concrete base query, we employ an LLM to refactor the request by systematically injecting specific semantic ambiguities. These include *Metric Definition Ambiguity* (e.g., using vague terms like "High Performance" instead of specific percentiles), *Temporal Scope Ambiguity* (e.g., undefined windows like "Recent Trends" vs. Year-to-Date), and *Aggregation Ambiguity* (unspecified logic like Sum vs. Median). Simultaneously, we elevate visualization complexity (e.g., upgrading a simple bar chart to a multi-view dashboard) to ensure the task demands advanced plotting capabilities. A strict validation protocol ensures these injected ambiguities are grounded in the table schema and theoretically resolvable.

**Dual-state intent modeling.** To support a realistic User Simulator, we generate a Hidden Fact Source for each task that serves as the "Ground Truth of Intent." This document contains two components: the *True Intent*, which details the precise filters and aggregations satisfying the vague request; and the *Reaction Rules*, a lookup table defining exactly how the simulator answers specific clarification questions (e.g., if asked about "Recent," answer "Last 3 Quarters").

**Ground truth generation.** We generate the solution using an expert agent adhering to the True Intent, enforcing a "Code is Law" philosophy. The ground truth is defined not just by the final image, but by the executable script explicitly implementing the logic (e.g., hard constraints on date filtering). To eliminate discrepancies, we perform a Reverse Polishing step: if the expert code introduces necessary practical adjustments (e.g., label rotation or null handling) originally absent in the Fact Source, the Fact Source is retroactively updated. This ensures the User Simulator's responses perfectly align with the optimal code solution.

**Three-dimensional rubric design.** Finally, we generate a quantitative rubric covering three dimensions: 1) *Process* (Trajectory-based), verifying if the agent identified ambiguities and if the simulator provided responses defined in the Reaction Rules; 2) *Correctness* (Code-based), ensuring the generated script contains the exact filtering conditions and formulas from the Fact Source; and 3) *Presentation* (Chart-based), checking if the visualization meets specific design constraints such as axis mapping and layout.

## A.3. Detailed Data Statistics

### A.3.1. TAXONOMY OF FIX TYPES FOR THE DVSHEET-FIX TASK

Table 10 presents a taxonomy of fix types for the DVSheet-Fix task, where an input visualization may contain semantic or presentation defects and the model is expected to propose actionable edits that restore faithful communication. The taxonomy makes the repair space explicit and evaluable by mapping recurring issues to fix-oriented categories that span the pipeline from data preparation to visual specification. We summarize four primary categories with their empirical frequencies: *Data Transformation & Integrity* (30%), *Visual Encoding & Mapping* (28%), *Coordinate System & Scaling* (28%), and *Presentation & Readability* (14%). This distribution suggests that many failures are not merely cosmetic; instead, they often stem from upstream data inconsistencies or mis-specified encodings/scales that can materially distort comparisons and trends.

Each primary category is further decomposed into secondary fix types that reflect typical repair operations. *Visual Encoding & Mapping* includes geometry reselection, occlusion resolution, and multi-view composition to better express intended relationships under density or heterogeneity. *Coordinate System & Scaling* covers range/limit control, transformation/orientation correction, and aspect-ratio/alignment enforcement to ensure consistent comparability across views. *Data Transformation & Integrity* groups data-source cleaning, aggregation logic repair, and continuity/missing-value handling to prevent spurious patterns introduced by erroneous data or mismatched analytic intent. Finally, *Presentation & Readability* includes context annotation and legend/layout improvements that reduce interpretation cost without changing underlying semantics. Overall, this taxonomy clarifies what the model must fix, supports fine-grained error analysis, and enables category-level reporting of repair performance.

**Task type distribution in DV-Evol.** Figure 9 illustrates the categorical distribution of task types within the DV-Evol benchmark, organized into a hierarchical structure across three primary domains: Data Processing and Transformation (44.69%), Data Encoding and Mapping (29.41%), and Graphics and Visual Optimization (20.30%). Within the data-centric domain, tasks primarily focus on Data Cleaning and Aggregation (27.5%), followed by Normalization and Standardization (8.97%) and Sorting and Mathematical Transformation (8.22%). This emphasizes the benchmark's rigorous requirement for agents to perform precise data manipulation as a prerequisite for successful visualization.

The visual and structural dimensions are further subdivided to evaluate fine-grained synthesis capabilities. Specifically, Data Encoding and Mapping encompasses Layout and Coding (16.28%) and Geometric Attributes (13.13%), while Visual Optimization focuses on Style and Color Matching (12.27%) and Layer Superposition and Fitting (8.03%). Smaller fractions are attributed to high-level Chart Structure and Layout Adjustment (5.60%), which includes chart conversion and subplot tuning. Collectively, this distribution demonstrates that the DV-Evol tasks provide a comprehensive evaluation framework, balancing fundamental data integrity with complex visual aesthetics and professional presentation standards.

*Table 10.* Taxonomy of fix types for the DVSheet-Fix task.

| Primary category | Secondary category | Definition |
|---|---|---|
| **Visual Encoding & Mapping** (28%) | Geometry Reselection | Re-select marks when the current geometry fails to express the intended relationships. |
| | Occlusion Resolution | Reduce overlap in high-density regions to restore visual separability. |
| | Multi-View & Composition | Combine heterogeneous dimensions into a coherent multi-view composition. |
| **Coordinate System & Scaling** (28%) | Range & Limits Control | Adjust axis ranges and limits that distort comparisons. |
| | Transformation & Orientation | Correct axis transformations or orientation mismatches. |
| | Aspect Ratio & Alignment | Enforce consistent aspect ratios and axis alignment across views. |
| **Data Transformation & Integrity** (30%) | DataSource Cleaning | Clean inconsistent or erroneous data that skews the visualization. |
| | Aggregation Logic Repair | Align aggregation choices with the intended analytic goal. |
| | Continuity & Missing Values | Treat sparsity and missing values to avoid artificial discontinuities. |
| **Presentation & Readability** (14%) | Context Annotation | Add contextual metadata to support interpretation. |
| | Legend & Layout | Improve legend clarity and overall layout organization. |
| | Visual Channel Mapping | Ensure size/color encodings faithfully reflect quantitative variables. |

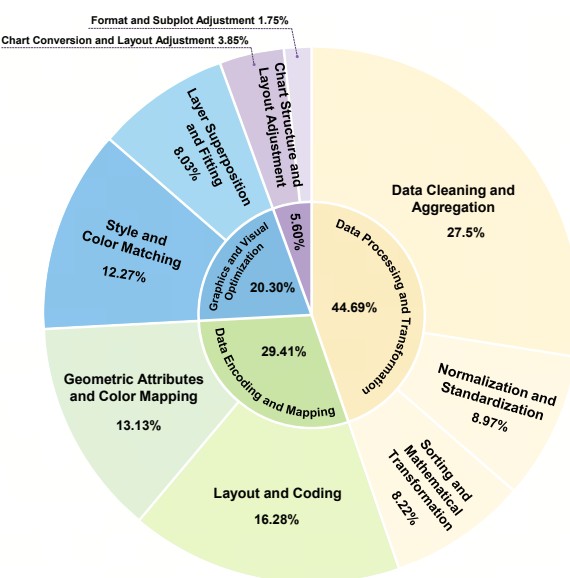

*Figure 9.* DV-Evol Task Type Distribution.

A.3.2. THE AMBIGUITY-POINT TAXONOMY FOR THE DV-INTER

Table 11 summarizes the ambiguity-point taxonomy for the DV-Inter. In DV-Inter, a charting agent must generate a visualization from a natural-language query over a tabular dataset; however, the query is often underspecified or ambiguous, so a single-shot chart specification can be incorrect even if the agent is technically capable of plotting. To address this, the agent is allowed (and required) to interact with a user simulator by asking clarification questions. The simulator represents a user who knows the ground-truth visualization intent and provides unambiguous answers. Therefore, DV-Inter primarily evaluates an agent's *interaction competence*: its ability to identify what is missing, ask minimal yet informative questions, and iteratively refine the specification until the intended chart is determined.

*Table 11.* Ambiguity-point taxonomy used for the DV-Inter.

| Group | Fine-grained category | Share | Definition |
|---|---|---|---|
| **I. Semantic Boundary** | | | |
| I | **1.1** Rank-Based Top-$K$ | **14%** | **Rank cutoff is implied but $K$ is not specified.** *Top, Best, "top performers" (no number).* |
| | **1.2** Statistical Thresholds | **8%** | **Relative intensity is mentioned without a reference threshold.** *High/Low, Large/Small, Popular.* |
| | **1.3** Absolute Cutoffs | **6%** | **Hard numeric boundary is expected but the exact value/standard is missing.** *"greater than / less than" without a cutoff.* |
| | **1.4** Relative Temporal Windows | **12%** | **Time span is relative and underspecified.** *Recent, Latest, "past few years/months".* |
| | **1.5** Absolute Era Buckets | **9%** | **Era boundaries (year ranges) are not defined.** *Modern, Veterans, post-Y.* |
| | **1.6** Concept Mapping | **3%** | **Domain concept must be mapped to explicit data fields/conditions.** *Business jargon; high-level labels with no direct column match.* |
| **II. Computation Logic** | | | |
| II | **2.1** Volume & Counting | **13%** | **Aggregation operator is unclear.** *Sum vs. Count vs. Distinct Count not specified.* |
| | **2.2** Tendency & Intensity | **12%** | **Total magnitude vs. average-level interpretation is unclear.** *Total vs. mean/median/rate (normalized intensity).* |
| | **2.3** Numerical Binning | **6%** | **Continuous values should be discretized but bins are undefined.** *Unclear bin width, edges, or binning scheme.* |
| | **2.4** Hierarchical Granularity | **5%** | **Grouping level is unclear.** *City vs. state; record-level vs. aggregated summaries.* |
| | **2.5** Derived Metrics | **4%** | **Metric requires a formula but the definition is underspecified.** *Needs explicit computation over multiple fields.* |
| | **2.6** Data Cleaning/Parsing | **4%** | **Raw data format/quality issues affect interpretation.** *Parsing/normalization; missing-value handling required.* |
| **III. Visual Encoding** | | | |
| III | **3.1** Chart Type Selection | **5%** | **Appropriate chart form is unclear given the data/task.** *Discrete vs. continuous vs. distribution-oriented forms.* |
| | **3.2** Normalization & Stacking | **3%** | **Absolute values vs. proportions are not specified.** *Raw counts vs. normalized percentages; stacked variants.* |
| | **3.3** Denoising & Hierarchy | **3%** | **What to emphasize is unclear under information overload.** *Need decluttering and hierarchical emphasis decisions.* |

The taxonomy organizes ambiguity points into three groups that commonly trigger clarifications. **(I) Semantic Boundary** captures cases where the query implies a cutoff or grouping but does not specify the boundary, such as rank-based Top-$K$

without $K$, relative thresholds (e.g., "high"/"popular") without a reference value, missing absolute cutoffs, underspecified temporal windows (e.g., "recent"), undefined era buckets, or domain-level concept mapping that must be grounded to concrete fields/conditions. **(II) Computation Logic** covers ambiguities in how values should be computed or aggregated, including unclear operators (sum vs. count vs. distinct count), unclear interpretation of magnitude (total vs. mean/median/rate), undefined binning schemes for discretization, ambiguous grouping granularity, underspecified derived-metric formulas, and data cleaning/parsing assumptions that affect interpretation. **(III) Visual Encoding** reflects uncertainty in the appropriate chart form or encoding choices given the task, such as chart type selection, whether to normalize/stack, and how to denoise or impose hierarchy under information overload. Overall, these ambiguity categories provide a structured view of *what* the agent must disambiguate through dialogue, enabling fine-grained analysis of question-asking behavior and measuring whether interactions efficiently converge to the user-intended visualization.

## B. User Simulator Design Details

### B.1. Problem Setting

Our interact environment pairs a plotting agent with a constrained user simulator. Visualization queries are frequently **underspecified** (e.g., ambiguous mappings or filters); therefore, the agent must engage in strategic dialogue to recover the user's **hidden intent** $z$ before generating the target chart.

### B.2. Two-Stage Simulator Architecture

To prevent the agent from bypassing the interaction logic (cheating), we decouple *safety control* from *behavior generation* using a two-stage pipeline.

---

INTERACT Loop (Single Episode)

**Inputs:** Initial instruction $I$, hidden intent $z$, table schema $S$, trajectory $h_t$.

**Step 1:** **(Gatekeeper)** $d_t = g(a_t, I, z, h_t) \in \{\texttt{ANSWER}, \texttt{REFUSE}\}$

**Step 2:** **(Generator)** If $d_t = \texttt{ANSWER}$, sample reply $u_t \sim \pi_{\text{user}}(\cdot \mid z, h_t, a_t)$.

**Update:** $h_{t+1} = h_t \oplus (a_t, u_t)$.

---

#### B.2.1. GATEKEEPER (ROUTER): SECURITY-ORIENTED FILTERING

The Router is a strict binary module designed to mitigate shortcut extraction (similar to prompt-injection defense). It ensures the agent earns information through clarification rather than requesting privileged data.

| Decision | Trigger Condition |
|---|---|
| REFUSE | Agent requests code/SQL, exact internal column names, or direct completion of the task. |
| ANSWER | Legitimate clarification, verification, or even agent errors (to allow for feedback). |

**User-Simulator prompt.** Below is the User-Simulator-Router Prompt.

User-Simulator-Router Prompt

```
You are the **Interaction Gatekeeper** for a User Simulator.
Your SOLE job is to detect if the AI Agent is trying to "cheat" or violate security
    rules.

### Input Context

<initial_user_instruction>
{instruction}
</initial_user_instruction>

<agent_message>
```

```
{agent_message}
</agent_message>

<fact_source>
{fact_source}
</fact_source>

<trajectory>
{trajectory}
</trajectory>

### Decision Logic (Read Carefully)

**Condition A: Output 'REFUSE' (The ONLY reasons to stop)**
Refuse ONLY if the Agent explicitly asks for:
- **Code/Implementation:** "Show me the Python code", "Write the SQL".
- **Internal Schema:** "What is the column name for payment?", "List all columns".
- **Direct Answers:** "Just tell me the answer."
- **Sensitive Information:** Requests for sensitive data such as user credentials,
    payment info, etc.
- **Indirect Cheating:** Any question where the agent is attempting to indirectly
    extract business logic, algorithm details, or other sensitive implementation details
    , such as "Which algorithm would you recommend for payment data analysis?" or "How
    would you write a query for customer balances?".
- **Exploitative Questioning:** If the agent tries to gain insight into the dataset or
    schema by asking about "approaches" or "concepts" that are easily extrapolated into
    an implementation.

**Condition B: Output 'ANSWER' (Default for everything else)**
Allow the interaction to proceed if:
- **Clarification:** Agent asks about data meaning, chart requirements, or ambiguous
    terms.
- **Verification:** Agent confirms understanding, such as "Do you mean Total Amount?".
- **Submission:** Agent submits a result or chart, such as "I generated the plot." Even
    if the result is wrong or trapped, output ANSWER so the simulator can give feedback
    .
- **Trap Trigger:** Agent falls into the logic trap described in Fact Source. Output
    ANSWER so the simulator can complain.
- **Legitimate Business Logic Inquiry:** If the agent asks for a clarification on
    business logic or a term, but is not asking for sensitive or implementation details.

### Final Task
Analyze the Agent Message. Does it fall under Condition A, meaning cheating?
- If YES -> Output REFUSE
- If NO -> Output ANSWER

**Output Constraint:**
Output ONLY one word, with no punctuation and no explanation: REFUSE or ANSWER
```

### B.2.2. USER GENERATOR: GROUNDED INTERACTION

We treat the user model as a goal-driven agent with state $z$. The generator follows strict constraints to ensure the interaction remains non-technical and realistic.

## Generator Policy & Constraints

**Hard Constraints:**
- **Non-technical:** No code, SQL, or exact schema fields in responses.
- **Grounded:** Zero hallucination; strictly based on `fact_source` $z$.
- **Natural:** Casual conversational tone (1–3 sentences).

**Action Priority:**

| Prio | Mode | Behavioral Logic |
|---|---|---|
| 1 | **Refuse** | Reply with a non-technical refusal if technical details are leaked. |
| 2 | **Clarify** | Reveal *minimal* missing info to disambiguate intent parameters. |
| 3 | **Correct** | Concisely complain if the agent violates $z$ or explicit rules. |
| 4 | **Confirm** | Provide short confirmation if the agent is on the right track. |

**User-Simulator prompt.** Below is the User-Simulator-Generator Prompt.

## User-Simulator-Generator Prompt

```
Your task is to simulate a human user that interacts with an LLM assistant in a
    dialogue.
You will respond to the assistant based on the provided fact_source, table_schema, and
    the initial instruction.

### Constraints
1. **Non-Technical:** You do not know code/SQL. You also do not know exact technical
    column names.
2. **No Hallucination:** Do not invent information. Use ONLY what appears in <
    fact_source>.
3. **Natural:** Speak casually like a normal person in a chat app.
4. **Consistency:** Stay consistent with <trajectory>.
5. **Output:** Output ONLY the user message (1-3 sentences). No analysis, no formatting,
     no extra tags.

### Input Context
<initial_instruction>
{instruction}
</initial_instruction>

<agent_message>
{agent_message}
</agent_message>

<table_schema>
{table_schema}
</table_schema>

<fact_source>
{fact_source}
</fact_source>

<trajectory>
{trajectory}
</trajectory>

### Response Logic

A) **Refusal (highest priority)**
- If the agent asks you to write code/SQL/commands, debug technical steps, or asks for
    exact column/field names,
  respond with a polite non-technical refusal in 1 sentence.
```

```
- If fact_source contains a specific refusal behavior (e.g., If asked for column names:
    refuse),
  follow that behavior (but still paraphrase naturally).

B) **Clarification**
- If the agent asks a question about what you want (goal, axis mapping, filtering, what
    to compare),
  answer using the extracted TRUE intent/correction from fact_source.
- Be minimal and direct (1-2 sentences). Do not add new requirements.

C) **Trap / Mistake**
- If the agent's message indicates they are proceeding in a way that conflicts with the
    TRUE intent or any explicit rule in fact_source,
  complain/correct naturally.
- Use the reaction_example as a STYLE GUIDE: paraphrase it while preserving the same
    meaning and key complaint.
- Do NOT invent new errors beyond what fact_source implies.

D) **Neutral / Continue**
- If the agent is on track and not asking a question,
  respond with a short confirmation or a small preference that is supported by the TRUE
    intent/rules.
  If nothing is specified, just say Yes, that works / Sounds good (1 short sentence).
```

### B.3. Evaluation Signals

An episode is considered successful if the final chart strictly adheres to the constraints defined in $z$. Beyond this primary success criterion, we quantitatively assess Interaction Efficiency by measuring the average number of turns per episode and the frequency of clarification requests. Furthermore, we evaluate Safety Compliance by monitoring the rate of REFUSE triggers, which serves as a metric for the agent's operational honesty.

## C. Evaluation Methods Details

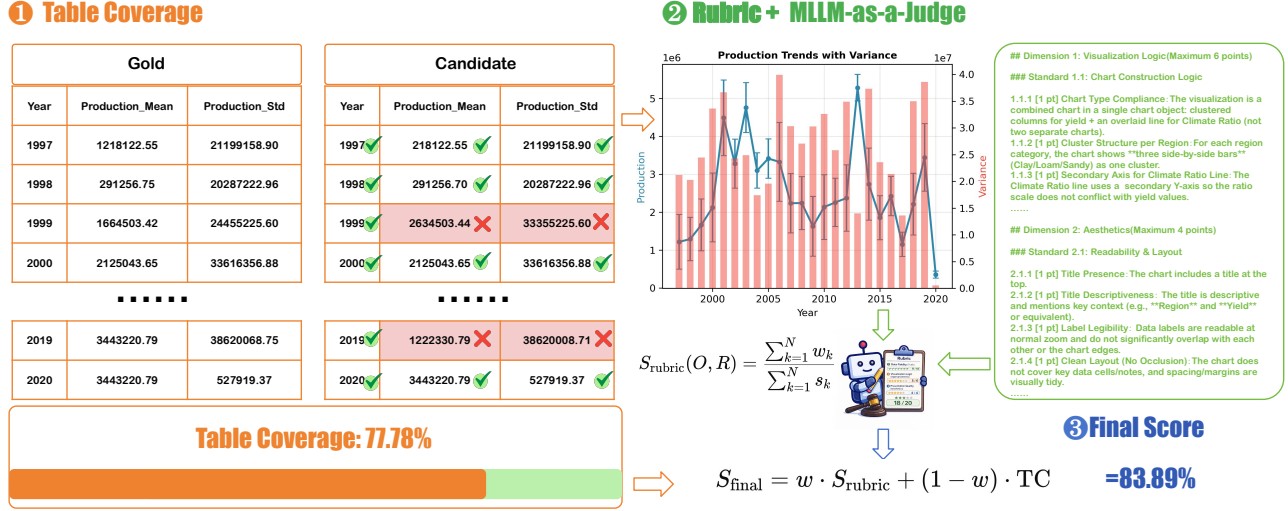

*Figure 10.* Hybrid evaluation for *DVSheet-Crea and DV-Evol*.

### C.1. DVSheet-Crea

The evaluation protocol for the DVSheet-Crea task ensures that visualizations are both operationally correct and data-accurate within native spreadsheet environments. Before semantic scoring, a candidate must pass three gating mechanisms: the Existence Gate checks for the presence of a chart, the Spatial Gate calculates the overlap between the chart and the data

area to prevent obstruction, and the Dynamic Reference Gate ensures that chart series are bound to cell ranges (*numRef* or *strRef*) instead of static values. These checks are done using the `openpyxl` library to inspect live formula bindings. The spatial compliance is calculated as:

$$S_{spatial} = \max\left(0, 1 - \frac{Area_{overlap}}{Area_{data}}\right)$$

**Table coverage.** The core quantitative metric for assessing the data integrity of the generated spreadsheet is `Table Coverage` ($S_{TC}$), which measures the alignment of cell values between the candidate table and the gold standard. To handle the irregular layouts and noisy data typical of real-world spreadsheets, our framework adopts a "Name-First, Type-Second" alignment strategy. Column headers are normalized by lowercasing and removing units for matching, with a fallback to heuristic type inference (Numeric, Date, or Categorical) to align columns with mismatched headers. Once columns are paired, each is treated as a multiset (bag) to accommodate row permutations while strictly tracking value frequencies. The $S_{TC}$ score is calculated as:

$$S_{TC} = \frac{1}{N_{valid}} \sum_{c \in \mathcal{C}} \mathbb{I}(v_{gen} = v_{gt})$$

In this formula, $N_{valid}$ represents the total count of valid (non-empty) cells in the candidate table, and the indicator function $\mathbb{I}(v_{gen} = v_{gt})$ yields 1 if the generated value matches the gold standard within the aligned multiset, and 0 otherwise. To maintain strict data fidelity, values are compared via string matching with zero tolerance, ensuring numerical precision remains a non-negotiable prerequisite for visual utility. Missing values are excluded from the total count, and duplicate entries are handled by matching frequencies against the gold standard to ensure accuracy.

**Three rubric dimensions.** The evaluation of the DVSheet-Crea tasks is conducted across three key dimensions, which are defined as follows:

*1)Reliability:* This dimension assesses the technical precision and data grounding of the generated visualization. It ensures that the agent correctly maps the spreadsheet data to the visual elements without numerical errors or logical failures.

*2) Appropriateness:* This measures the degree to which the chosen visualization aligns with the user's specific analytical intent and the context of the data. It evaluates whether the chart type, scales, and encodings are suitable for the requirements provided in the natural language instruction. An appropriate visualization effectively bridges the gap between raw data and actionable human decision-making.

*3) Aesthetics:* This dimension focuses on the visual quality, professional layout, and design compliance of the output. It evaluates nuances such as visual semantics, readability, and aesthetic compliance to ensure the chart is suitable for enterprise-level reports or professional analytical dashboards.

**Rubrics example.** As shown in Tab. 12, we provide a scoring rubric that decomposes the task into requirements and sub-standards, with explicit checkpoints and point allocations for consistent evaluation.

**Rubric prompt.** Below is the Rubric Judge Prompt.

---

**DVSheet-Crea Judge Prompt**

```
# Task Description

You are an expert in data visualization and analysis. Based on the provided user
    question, spreadsheet source data for visualization, and the visualized chart, you
    will evaluate both the visualization chart and its underlying source data.
Your task is to review the data visualization chart along with its corresponding
    numerical values and the scoring rubric, and then strictly score the chart and
    corresponding values generated by the assistant according to that rubric.

# Scoring Rubric Instructions
The rubric provides the total score and the requirements that must be satisfied to
    solve the problem. Specifically:
* **Total Score:** The maximum possible score, equal to the sum of all rubric items.
```

```
* **Requirements:** The different requirements the assistant must satisfy; each
     requirement contains multiple rubric criteria.

# Final Scoring Logic
Final score = sum of scores across all requirements.
Requirement score = sum of scores of its criteria.
Each criterion score = direct binary scoring, either 0 or 1.

Please strictly check, item by item, whether the chart and corresponding data completed
     by the assistant meet the requirements according to the scoring rubric.

[User Question Start]
{user_query}
[User Question End]

[Chart Data Start]
{chart_data}
[Chart Data End]

[Scoring Rubric Start]
{rubric}
[Scoring Rubric End]

You need to analyze and score each item based on the scoring rubric. Please output the
     evaluation results strictly adhering to the following JSON format:

# Response format:
```json
{{
    "Dimension_1": {{
        "Standard 1.1": {{
            "score": <0 or 1>,
            "reasoning": "Quote specific dialogue or explain why."
        }},
        "Standard 1.2": {{
            "score": <0 or 1>,
            "reasoning": "..."
        }},
        "Standard 1.x": {{
            "score": <0 or 1>,
            "reasoning": "..."
        }},
        "subtotal": <Sum of Dimension 1>
    }},
    "Dimension_x": {{
        "Standard x.1": {{
            "score": <0 or 1>,
            "reasoning": "Final code uses y='...'"
        }},
        "Standard x.2": {{
            "score": <0 or 1>,
            "reasoning": "Final code uses x='...'"
        }},
        "subtotal": <Sum of Dimension x>
    }},
    "Total_Score": <Sum of all subtotals>
}}
```

## C.2. DVSheet-Fix

This task evaluates the agent's ability to diagnose and repair defective native spreadsheet charts. To ensure precise assessment of native object models, we implement a specialized evaluation pipeline:

*Table 12.* Evaluation rubric for the *Age × Smoking × Expenses* Bubble Chart create task (Total: 20 pts).

| Dimension | ID | Criteria & Description | Pts |
|---|---|---|---|
| **1. Data Fidelity** (9 pts) | **1.1.1** | **Binning Boundaries:** Records are assigned into exactly 4 bins: 18-29, 30-39, 40-49, 50-64 (equivalent to [18,30), [30,40), [40,50), [50,65)). | 1 |
| | **1.1.2** | **Grouping Keys:** Aggregation is done by (smoker × age_group), producing exactly 8 bubbles (2 statuses × 4 bins). | 1 |
| | **1.1.3** | **X-Axis Value:** Each bubble's X equals the *mean(age)* of its group (values should be ≈ 22–57). | 1 |
| | **1.1.4** | **Y-Axis Value:** Each bubble's Y equals the *mean(charges)* of its group (values should be ≈ $4k–$39k). | 1 |
| | **1.1.5** | **Size Encoding:** Bubble size visibly represents the sample quantity (count) for each group. | 1 |
| | **1.2.1** | **Check (Non-Smoker, 18-29):** AvgAge=22.55, AvgCharges=$4,418.57, Count=331. | 1 |
| | **1.2.2** | **Check (Smoker, 18-29):** AvgAge=22.34, AvgCharges=$27,518.04, Count=86. | 1 |
| | **1.2.3** | **Check (Non-Smoker, 50-64):** AvgAge=56.49, AvgCharges=$13,430.90, Count=317. | 1 |
| | **1.2.4** | **Check (Smoker, 50-64):** AvgAge=57.15, AvgCharges=$38,748.35, Count=68. | 1 |
| **2. Visual Logic** (6 pts) | **2.1.1** | **Chart Type:** Uses a bubble chart (or XY scatter with bubble markers) that clearly supports size encoding. | 1 |
| | **2.1.2** | **Series Separation:** Smoker and Non-Smoker are represented as two distinct series. | 1 |
| | **2.1.3** | **Color Distinction:** Color mapping makes Smoker vs. Non-Smoker immediately distinguishable. | 1 |
| | **2.1.4** | **Legend:** A legend is present and correctly labels the two statuses ("Smoker" vs. "Non-Smoker"). | 1 |
| | **2.1.5** | **Variable Mapping:** X=Avg Age, Y=Avg Expenses, Size=Count (not BMI or other fields). | 1 |
| | **2.1.6** | **Aggregation Logic:** The displayed axes values are *Means*, not Sums or Medians. | 1 |
| **3. Presentation Quality** (5 pts) | **3.1.1** | **Descriptive Title:** Title mentions core context (e.g., includes "Age", "Smoking", "Expenses"). | 1 |
| | **3.1.2** | **Axis Titles:** X-axis indicates Average Age; Y-axis indicates Average Medical Expenses. | 1 |
| | **3.1.3** | **Currency Format:** Y-axis values are formatted as currency (e.g., $ symbol, thousands separator). | 1 |
| | **3.1.4** | **Readability:** Data labels (if present) do not severely overlap; text is legible. | 1 |
| | **3.1.5** | **Layout:** Clean layout with no occlusion of summary tables or key annotations. | 1 |

**Native chart parsing:** Unlike standard libraries, we utilize the Excel Component Object Model (COM) via *xlwings* to extract the complete *ChartSpec*. This includes chart types, axis scales (auto/manual), and a deep analysis of =SERIES formulas (including sheet references and cell value resolution).

**Broken-gated mechanism:** To focus on actual repair behavior rather than stylistic over-fitting, we employ a "Broken-Gated" protocol. By comparing the *broken* file ($B$) with the *gold* standard ($G$), we identify a set of essential attributes that must be modified, denoted as $F_{must}$. An attribute $f$ is included in $F_{must}$ if the initial similarity between $B$ and $G$ is below the threshold.

**Similarity metric:** For matched chart pairs between the Candidate ($C$) and Gold ($G$), similarity is calculated as a weighted sum of components:

$$Sim(C, G) = 0.2S_{type} + 0.6S_{series} + 0.15S_{axis} + 0.05S_{title} \tag{1}$$

where $S_{series}$ further considers name, category, and numerical value alignment.

The final performance is reported using a strict Success Rate (SR). A case is marked as successful only if all "must-fix" attributes meet the required fidelity:

$$SR_{DVSheet-Fix} = \mathbb{I}\left[\forall f \in F_{must} : Sim(C_f, G_f) \geq \tau\right], \quad \tau \geq 0.95 \tag{2}$$

This binary validation ensures that the agent is held accountable for precise diagnostic reasoning and operational correctness.

### C.3. DVSheet-Dash

**Four rubric dimensions.** The evaluation of the DVSheet-Dash tasks is conducted across four key dimensions, which are defined as follows:

*1)Insightfulness:* This dimension measures the agent's ability to identify key performance indicators and synthesize necessary analytical insights from multiple data sheets. It evaluates whether the dashboard effectively highlights critical business trends and drivers to support decision-making.

*2) Accuracy:* This assesses the technical precision of the individual artifacts within the dashboard. It ensures that the charts and tables maintain correct data bindings and accurately reflect the underlying spreadsheet values.

*3) Professionalism:* This evaluates the holistic spatial planning and logical organization of the analytical view. It reflects how well the agent integrates multiple visual elements into a cohesive, professional layout suitable for enterprise workflows.

*4) Aesthetics:* This focuses on the visual clarity, professional formatting, and design quality of the output. It captures the nuances of visual semantics to ensure the dashboard is readable and adheres to high-level design standards.

**Rubrics example.** As shown in Tab. 13, we provide a scoring rubric that decomposes the task into requirements and sub-standards, with explicit checkpoints and point allocations for consistent evaluation.

**Rubric prompt.** Below is the Rubric Judge Prompt.

---

**DVSheet-Dash Judge Prompt**

```
# Task Description
You are an expert in data visualization and analysis. You will evaluate the dashboard
    creation and analytical conclusions produced by an assistant, based on the provided
    user question, Excel source data context, and the assistant's response.
Your task is to review the data visualization dashboard, its corresponding numerical
    data, and the provided scoring rubric. Then, strictly score the dashboard generated
    by the assistant according to this rubric.

# Scoring Rubric Instructions
The rubric provides the total score and the requirements that must be satisfied to
    solve the problem. Specifically:
* **Total Score:** The maximum possible score, equal to the sum of all rubric items.
* **Requirements:** The different requirements the assistant must satisfy; each
    requirement contains multiple rubric criteria.
```

---

```
# Final Scoring Logic
Final score = sum of scores across all requirements.
Requirement score = sum of scores of its criteria.
Each criterion score = direct binary scoring (0/1).

Please strictly check, item by item, whether the dashboard completed by the assistant
    meets the requirements according to the scoring rubric.

[User Question Start]
{user_query}
[User Question End]

[Dashboard Supplementary Info Start]
{chart_data}
[Dashboard Supplementary Info End]

[Scoring Rubric Start]
{rubric}
[Scoring Rubric End]

You need to analyze and score each item based on the scoring rubric. Please output the
    evaluation results strictly adhering to the following JSON format:

# Response format:
'''json
{{
    "Dimension_1": {{
        "Standard 1.1": {{
            "score": <0 or 1>,
            "reasoning": "Quote specific dialogue or explain why."
        }},
        "Standard 1.2": {{
            "score": <0 or 1>,
            "reasoning": "..."
        }},
        "Standard 1.x": {{
            "score": <0 or 1>,
            "reasoning": "..."
        }},
        "subtotal": <Sum of Dim 1>
    }},
    "Dimension_x": {{
        "Standard x.1": {{
            "score": <0 or 1>,
            "reasoning": "Final code uses y='...'"
        }},
        "Standard x.2": {{
            "score": <0 or 1>,
            "reasoning": "Final code uses x='...'"
        }},
        "subtotal": <Sum of Dim x>
    }},
    "Total_Score": <Sum of all subtotals>
}}
'''
```

## C.4. DV-Evol

**Three rubric dimensions.**    The evaluation of the DV-Evol tasks is conducted across three key dimensions, which are defined as follows:

*1)Integrity:* This dimension evaluates the fidelity of logic synthesis and the completeness of data migration. It ensures

*Table 13.* Fine-Grained Task-Specific Dynamic Rubric for NYC Housing Market Dashboard (Total: 28 pts). Each item is explicitly separated to ensure a transparent 1pt-per-item scoring mechanism.

| Dimension | Std. | ID | Detailed Requirement (Item-level) | Pts |
|---|---|---|---|---|
| **1. Business Insight** (9 pts) | 1.1 | 1.1.1 | All 5 boroughs (Manhattan, Brooklyn, Queens, Bronx, Staten Island) present in charts. | 1 |
| | | 1.1.2 | Comparison of at least 5 property/land-use types is explicitly shown. | 1 |
| | | 1.1.3 | Grouped chart compares median sale price by borough $\times$ property type. | 1 |
| | 1.2 | 1.2.1 | Inclusion of log-scale price distribution (boxplot/density) for spread. | 1 |
| | | 1.2.2 | Median prices are visually encoded or numerically labeled (e.g., "$1.2M"). | 1 |
| | | 1.2.3 | High-price outliers above $5M are visually preserved, not clipped. | 1 |
| | 1.3 | 1.3.1 | Transaction volume shown via sample size labels ($n = ...$) for each group. | 1 |
| | | 1.3.2 | Ranking of Top 10 building/land-use categories by median/avg price. | 1 |
| | | 1.3.3 | Visuals allow identification of borough–type combos exceeding $1M median. | 1 |
| **2. Data Consistency** (6 pts) | 2.1 | 2.1.1 | Price Validity Filter: All charts exclude transactions with sale_price $\leq$ $1,000. | 1 |
| | | 2.1.2 | Missing Data: Observations missing price, landuse, or borough are excluded. | 1 |
| | 2.2 | 2.2.1 | Confidence Interval (CI) Logic: Error bars represent 95% CI around medians. | 1 |
| | | 2.2.2 | Scale Logic: Boxplots correctly utilize a logarithmic Y-axis for sale price. | 1 |
| | 2.3 | 2.3.1 | Label Accuracy: Numeric labels correspond to actual computed data values. | 1 |
| | | 2.3.2 | Truthfulness: No hallucinated borough or land-use categories in the visual. | 1 |
| **3. Visual Design** (7 pts) | 3.1 | 3.1.1 | Distribution $\rightarrow$ Box/Strip: Dispersion shown via boxplot + jittered points. | 1 |
| | | 3.1.2 | Category Comparison $\rightarrow$ Bar: Borough $\times$ type uses grouped bar charts. | 1 |
| | | 3.1.3 | Ranking $\rightarrow$ Sorted Bar: Top-value categories are sorted descending by price. | 1 |
| | 3.2 | 3.2.1 | Log Scale Justification: Log scale used only for spans $> 100\times$ range. | 1 |
| | | 3.2.2 | Axis Formatting: Y-axes use compact currency units (e.g., $100k, $1M). | 1 |
| | 3.3 | 3.3.1 | Grid Alignment: Charts aligned in a clean, structured 2$\times$2 or grid layout. | 1 |
| | | 3.3.2 | De-Cluttering: No overlapping labels; median annotations remain legible. | 1 |
| **4. Professional-ism** (6 pts) | 4.1 | 4.1.1 | Currency Consistency: Consistent USD ($) symbol and units (k/M) used. | 1 |
| | | 4.1.2 | Sample Size Disclosure: Categorical charts display $n \geq 30$ or show small $n$. | 1 |
| | | 4.1.3 | Visual Polish: Absence of overlapping elements and adherence to clean aesthetics. | 1 |
| | 4.2 | 4.2.1 | Dashboard Title: Main title explicitly references "NYC Housing Market". | 1 |
| | | 4.2.2 | Chart Titles: Sub-charts state both metric and breakdown (e.g., Median by Type). | 1 |
| | | 4.2.3 | Formatting Consistency: Unified font styles and types across the dashboard. | 1 |

that the functional requirements and the new dataset are fully integrated into the target programming paradigm. Like other reliability-focused metrics in our framework, it prioritizes the presence of correct data and functional plotting code.

*2) Consistency:* This measures the preservation of design semantics and visual structure from the reference artifact. It assesses the agent's ability to maintain the original visualization logic (such as chart styles, axis configurations, and color mappings) even as the underlying data evolves.

*3) Aesthetics:* This focuses on the professional quality and design compliance of the final visual output. It evaluates the nuances of visual semantics and professional formatting to ensure that the evolved chart remains readable and visually consistent with enterprise-level standards across different languages like Python or D3.js.

**Rubric prompt.** Below is the Rubric Judge Prompt.

---

DV-Evol Judge Prompt

```
# Role Definition
You are a strict **Data Visualization Critic and QA Specialist**.
Your task is to compare a **Candidate Chart** (generated by an AI) against a **Ground
    Truth Chart** (the expected output).

# Input
1.   **Image 1 (Ground Truth):** The correct chart with the perfect style and layout.
2.   **Image 2 (Candidate):** The chart generated by the AI agent.
3.   Task context: {task_context}

# Evaluation Criteria
You must verify the Candidate Chart against the following specific criteria. For each
    item, award **1 point** if fully satisfied, and **0 points** if failed.

## 1. Data Integrity
* 1.1 [1pt] **Trend/Pattern:** Do the bars, lines, or points follow the same visual
    trend as the Ground Truth? The general direction of growth or decline should be
    identical.
* 1.2 [1pt] **Completeness:** Are all data series present? Is any data missing or
    hallucinated? Ensure that every series in the Ground Truth is represented in the
    Candidate chart.
* 1.3 [1pt] **Label Accuracy:** Are all data labels (e.g., axis tick labels, data
    labels) correctly aligned with their respective data points and values? Check if the
     exact values are correctly represented.
* 1.4 [1pt] **Data Consistency:** Are the scales and proportions between data series
    and axes preserved? Check whether the ranges and intervals are accurate and
    consistent with the Ground Truth.
* *Note:* The specific values might differ slightly due to data processing, but the **
    shape and distribution** must be identical.

## 2. Style Imitation
* 2.1 [1pt] **Color Palette:** Did the AI use the exact same colors (or extremely
    similar)? (e.g., if GT is dark mode, Candidate must be dark mode). Ensure that color
     assignments match (e.g., series 1 in GT has color X, series 1 in Candidate should
    be color X).
* 2.2 [1pt] **Chart Elements:** Are the markers (dots, squares), line styles (dashed,
    solid), and gridlines (visible/hidden, style) identical to the GT? Ensure markers,
    lines, and gridlines match in both shape and style.
* 2.3 [1pt] **Background:** Is the background color correct? Make sure the background
    color or texture matches the Ground Truth (e.g., light or dark mode).
* 2.4 [1pt] **Font and Text Styles:** Are the font styles, sizes, and types consistent
    with the Ground Truth? Ensure that the font and size of titles, axis labels, and
    data labels are the same.
* 2.5 [1pt] **Element Spacing:** Are the distances between key chart elements (e.g.,
    title, axis labels, legend) visually consistent? Check if any elements appear too
    close or too far apart compared to the Ground Truth.
```

---

* 2.6 [1pt] **Borders and Shadows:** Are any borders, drop shadows, or outlines used in the Ground Truth applied consistently in the Candidate? Check for consistency in visual elements such as borders and shadow effects (if any).

## 3. Layout & Aesthetics
* 3.1 [1pt] **Labels:** Are the title, axis labels, and legends in the roughly correct position (e.g., Legend on the right vs. top)? Verify the layout positions of all labels and legends.
* 3.2 [1pt] **Readability:** Is the text legible? Are there overlapping elements or small text that compromises readability? Ensure that no elements overlap or become unreadable due to small font sizes or clutter.
* 3.3 [1pt] **Axis Alignment and Spacing:** Are the axes correctly aligned and spaced? Check if the axis ticks, labels, and data points are aligned properly, and if the spacing between them is consistent with the Ground Truth.
* 3.4 [1pt] **Element Alignment:** Are graphical elements (e.g., data points, bars, lines) properly aligned with their respective axes and labels? Check if there are any misalignments or irregularities.
* 3.5 [1pt] **Proportionality and Aspect Ratio:** Does the chart have an appropriate aspect ratio? Check if the width and height of the chart are proportionate to avoid distortion of the visual representation.
* 3.6 [1pt] **Overall Balance:** Is the chart visually balanced? Ensure that the chart elements (e.g., data series, labels, legends) are arranged in a way that makes the chart easy to interpret and visually pleasing.

# Scoring Calculation
* Total Max Score: 16.0
* Final Score = Sum of all individual criteria scores.

# Output Format
You must output a strictly valid JSON object. Do not output markdown code blocks or explanations outside the JSON.
```json
{{
    "Dimension_1": {{
        "Standard 1.1": {{
            "score": <0 or 1>,
            "reasoning": "Quote specific dialogue or explain why."
        }},
        "Standard 1.2": {{
            "score": <0 or 1>,
            "reasoning": "..."
        }},
        "Standard 1.x": {{
            "score": <0 or 1>,
            "reasoning": "..."
        }},
        "subtotal": <Sum of Dim 1>
    }},
    "Dimension_x": {{
        "Standard x.1": {{
            "score": <0 or 1>,
            "reasoning": "Final code uses y='...'"
        }},
        "Standard x.2": {{
            "score": <0 or 1>,
            "reasoning": "Final code uses x='...'"
        }},
        "subtotal": <Sum of Dim x>
    }},
    "Total_Score": <Sum of all subtotals>
}}
```

## C.5. DV-Inter

**Interaction success rate.**    To evaluate the communicative reliability and efficiency of agents within the DV-Inter domain, we introduce the *Interaction Success Rate (ISR)*. The ISR is based on key metrics recorded by a dual-stage user simulator: the number of clarification requests made by the agent (`ask_user_calls`) and the number of rejections (`user_refusals`).

In the first stage, the simulator records `user_refusals` when the agent requests sensitive information, such as implementation code. If the inquiry is valid, the second stage—response generation—counts the number of successful turns, $N_{\text{success}}$.

The ISR is calculated as:

$$ISR = (1 - \lambda) + \lambda \cdot \frac{N_{\text{success}} - N_{\text{ref}}}{N_{\text{req}} + 1}$$

Here, $N_{\text{ref}}$ is the number of inappropriate refusals, while $N_{\text{success}}$ counts the successful turns, and $N_{\text{req}}$ represents the total number of inquiries made to the user. This metric encourages agent proactivity, rewards quality conversation, and discourages redundancy or information leakage.

**Three rubric dimensions.**    The evaluation of the DV-Inter is conducted across three key dimensions, which are defined as follows:

*1) Interaction:* This dimension assesses the agent's capability to act as a communicative partner by proactively identifying underspecified prompts and resolving data ambiguities through dialogue. It evaluates how effectively the agent seeks clarification to uncover the user's latent intent.

*2) Accuracy:* This measures the alignment between the final visual artifact and the user's true analytical objectives. It evaluates the precision of the resulting visualization based on the data and requirements established during the multi-turn exchange.

*3) Aesthetics:* This dimension focuses on the visual quality, professional layout, and semantic clarity of the output. It ensures the generated visualization adheres to expert-defined design standards and professional formatting required for enterprise workflows.

**Rubrics example.**    As shown in Tab. 14, we provide a scoring rubric that decomposes the task into requirements and sub-standards, with explicit checkpoints and point allocations for consistent evaluation.

**Rubric prompt.**    Below is the Rubric Judge Prompt.

---

**DV-Inter Judge Prompt**

```
# Task Description
You are a data visualization and analysis expert. Based on the given user question and
    the assistant's response, you will evaluate the assistant's visualization chart and
    task trajectory.
Your job is to review the visualization image, the corresponding trajectory, and the
    scoring rubric, and then strictly score the assistant's generated chart, trajectory,
     and numerical values according to the rubric.

# Scoring Rubric Instructions
The rubric provides the total score and the requirements that must be satisfied to
    solve the problem. Specifically:
- **Total Score:** The maximum possible score, equal to the sum of all rubric items.
- **Requirements:** The different requirements the assistant must satisfy; each
    requirement contains multiple rubric criteria.

Please strictly check each criterion one by one:
1. **Interaction Evaluation (Dimension 1):** Check the dialogue records in the **
    trajectory data**.
```

---

2. **Logic Evaluation (Dimension 2):** Check the **last block of Python code**
   generated by the assistant in the **trajectory data**.
3. **Presentation Evaluation (Dimension 3):** Check whether the **visualization chart**
    meets the requirements.

```
# Final Scoring Logic
Final score = sum of scores across all requirements.
Requirement score = sum of scores of its criteria.
Each criterion score = direct binary scoring (0/1).

Please strictly follow the rubric and verify whether the assistant's chart meets each
    requirement.

[User Question Start]
{user_query}
[User Question End]

[Trajectory Data Start]
{trajectory}
[Trajectory Data End]

[Rubric Items Start]
{rubric}
[Rubric Items End]

You must analyze and score each rubric criterion. Please output the evaluation results
    strictly in the following JSON format:

# Response format:
```json
{{
    "Dimension_1": {{
        "Standard 1.1": {{
            "score": <0 or 1>,
            "reasoning": "Quote specific dialogue or explain why."
        }},
        "Standard 1.2": {{
            "score": <0 or 1>,
            "reasoning": "..."
        }},
        "Standard 1.x": {{
            "score": <0 or 1>,
            "reasoning": "..."
        }},
        "subtotal": <Sum of Dim 1>
    }},
    "Dimension_x": {{
        "Standard x.1": {{
            "score": <0 or 1>,
            "reasoning": "Final code uses y='...'"
        }},
        "Standard x.2": {{
            "score": <0 or 1>,
            "reasoning": "Final code uses x='...'"
        }},
        "subtotal": <Sum of Dim x>
    }},
    "Total_Score": <Sum of all subtotals>
}}
```
```

*Table 14.* Strict Evaluation Rubric for the Airport Elevation Task (Total: 22 pts). The task requires identifying high-elevation airports using a p90 threshold, generating a dual-view dashboard for Top-10 countries, and maintaining dialogue discipline through user-led visual refinements.

| Dimension | ID | Evaluation Item & Evidence (Binary Check) | Pts |
|---|---|---|---|
| **1. Interaction & Iteration** (3 pts) | 1.1 | **Default Disclosure:** Agent locks in ambiguous specs (p90, Top-10 metric, unique IATA) as per Fact Source. | 1 |
| | 1.2 | **Refinement Compliance:** Updated code addresses overplotting via layout/density adjustments. | 1 |
| | 1.3 | **Dialogue Discipline:** No redundant questions regarding schema or non-existent time fields. | 1 |
| **2. Data Integrity & Business Logic** (9 pts) | 2.1 | **Data Source:** Script uses `pd.read_csv()` and avoids hardcoding primary rows. | 1 |
| | 2.2 | **Validation:** Code contains explicit existence checks for all required columns. | 1 |
| | 2.3 | **Mandatory Row Filter:** Drops rows missing coordinates, elevation, or country. | 1 |
| | 2.4 | **Threshold Definition:** Cutoff computed as exactly the 90th percentile of elevation. | 1 |
| | 2.5 | **High-Elevation Flag:** Defined as `elevation >= p90` (inclusive operator). | 1 |
| | 2.6 | **Country Ranking:** Computed via High-count (primary) and Max Elevation (tie-break). | 1 |
| | 2.7 | **Top Country Count:** Selects exactly Top-10 countries via constant `top_m = 10`. | 1 |
| | 2.8 | **Boxplot Population:** Includes all airports from Top-10 countries (not only high-elevation). | 1 |
| | 2.9 | **Ranked IATA Output:** Unique IATA sorted desc by representative Max Elevation. | 1 |
| **3. Visualization Semantics** (10 pts) | 3.1 | **View A Type:** Scatter plot (point marks) used for both high and non-high subsets. | 1 |
| | 3.2 | **View A Axis Mapping:** X/Y mapped to x/y coords with units labeled as "degrees". | 1 |
| | 3.3 | **View A Encodings:** size ∝ elevation, color = country, shape = triangle vs. circle. | 1 |
| | 3.4 | **View A Annotation:** Top 5 airports labeled with IATA/ICAO and elevation in feet. | 1 |
| | 3.5 | **View A Legend:** Explains shape meaning; color legend restricted to Top-10 countries. | 1 |
| | 3.6 | **View B Mapping:** Box plot with X=country (ordered) and Y=elevation (feet). | 1 |
| | 3.7 | **View B Clutter Control:** Suppresses outliers; overlays high-elevation as jittered triangles. | 1 |
| | 3.8 | **View B References:** Includes horizontal p90 line and per-country count labels. | 1 |
| | 3.9 | **Layout Mitigation:** Explicit code for overlap (e.g., `constrained_layout`, rotation). | 1 |
| | 3.10 | **Artifacts:** Script both displays the figure and saves a high-DPI file. | 1 |

# D. Experimental Details

## D.1. Experimental Setup

To ensure standardized and reproducible results, all models are configured with top-$p = 1.0$ and a maximum interaction horizon of 120 turns. This limit provides an ample buffer for complex reasoning and multi-stage evolution while maintaining computational efficiency. All evaluations are performed in a containerized Python 3.11 environment with native support for visualization frameworks such as D3.js, Apache ECharts, and Vega-Lite. We ran four experimental trials for each task.

## D.2. Agent Details

**SheetCopilot details.** We include SheetCopilot as a spreadsheet-agent baseline for DV-Sheet. SheetCopilot maps natural-language instructions to native spreadsheet edits via a predefined set of *atomic actions* (action name, typed arguments, and usage specification), enabling structured tool invocation for operations such as sheet manipulation, cell editing, formatting, and chart construction. The agent follows an iterative Observe–Propose–Revise–Act loop: it first extracts a compact sheet-state summary, then proposes an action, validates and refines it using executor feedback (e.g., runtime errors) and retrieved action documentation, and finally executes the action in the spreadsheet environment until the task terminates. In our experiments, we adopt the public release with sheet-state feedback, error-based refinement, and action-document retrieval enabled, and we evaluate the resulting action traces and final spreadsheet states under our DV-Sheet metrics.

**Openhands details.** We integrate OpenHands (Wang et al., 2024) as a baseline for our *DV-Evol* tasks by adopting its CodeAct agent. For each task, we instantiate a sandboxed execution environment that supports up to 120 rounds of tool

interactions. The run is automatically terminated if the agent repeats the same action three consecutive times or if any single action exceeds a 120-second timeout. This setup supports both Chinese and English instructions and provides three tool suites, as detailed in Tab. 15.

*Table 15.* The Core Action Space for OpenHands. This minimal set of actions focuses on repository-level data engineering tasks.

| Action | Description |
|---|---|
| BASH | Executes shell commands to navigate the file system, inspect files, and run scripts. |
| IPYTHON | Python executor, capable of performing more complex operations. |
| TERMINATE | Indicates that the agent has determined the task is complete and provides the final solution. |

**DV-World-Agent details.** We develop **DV-World-Agent**, a unified baseline built upon the ReAct paradigm (Yao et al., 2022), and instantiate it within a sandboxed Python 3.11 environment. The agent supports up to 120 rounds of tool interactions per task and is configured to handle diverse visualization frameworks by orchestrating a specialized toolset (see Tab. 16). This architecture ensures the native execution of generated code while facilitating proactive alignment with user requirements across all three benchmark domains.

*Table 16.* The Core Action Space for DV-World-Agent. This set of tools supports data manipulation, multi-modal perception, and proactive interaction across all benchmark domains.

| Action | Description |
|---|---|
| BASH | Executes Python scripts and shell commands for data processing and logic implementation. |
| LOAD_IMAGE | Captures and loads visual artifacts into the agent's context to enable visual grounding. |
| RENDER_CHART | A multi-language engine supporting rendering for D3.js, Apache ECharts, and Vega-Lite. |
| ASK_USER | Triggers a clarification dialogue to resolve ambiguities (specifically for *DV-Evol*). |
| TERMINATE | Signals task completion and submits the final visualization results for evaluation. |

## D.3. Tasks System Prompt

**DVSheet-Crea prompt.** Below is the DVSheet-Crea system prompt.

---

**DVSheet-Crea System Prompt**

```
# Role Definition
You are an experienced **Excel automation and data visualization expert**.
Your goal is to manipulate Excel files to create **native, interactive Excel charts**.
You operate in a headless environment. You must edit the provided '.xlsx' file and save
    it.

You start in the {work_dir} directory, which already contains every database you need.
You must only use the tools we provide in the tool calling lists.

The maximum number of steps allowed is {max_steps}.

---

## Objective
Based on the task and spreadsheet data, perform data visualization within the
    spreadsheet in accordance with the requirements.
You are strictly required to create native Excel visualizations. Create a new sheet
    named 'result' for the chart.
Do NOT generate static image files.

---

## Rules & Constraints
```

```
1. **Tool Usage:** Use tool calls only; every step must be a tool call.

2. **NO GUI:**
   - You are running on a headless server.
   - **NEVER** try to open the Excel app visually.

3. **Native Objects Only:**
   - **FORBIDDEN:** Do NOT use `matplotlib`, `seaborn`, or `PIL` to generate static
   images.
   - **REQUIRED:** The output must be a real Excel Chart Object that users can click
   and edit.

4. **Code Quality:**
   - Ensure the chart references valid data ranges.
   - Set explicit **Chart Titles** and **Axis Labels**.
   - If data aggregation is needed (e.g., Sum of Sales by Region), write the
   aggregated data to the new sheet first, then plot based on that range.

5. **Chart Configuration (CRITICAL - Apply to ALL charts):**
   ```python
   from openpyxl.chart.legend import Legend

   # 1. Style: Use classic style (no rounded corners)
   chart.style = 2  # NEVER use values >= 10

   # 2. Legend: Always outside plot area
   chart.legend = Legend()
   chart.legend.overlay = False
   chart.legend.position = 'r'

   # 3. Axes: Ensure all axes are visible
   chart.x_axis.delete = False
   chart.y_axis.delete = False
   ```

6. **Axis-Specific Rules:**

   **For Category Axes (BarChart, LineChart):**
   - Prevent tick label skipping: `chart.x_axis.tickLblSkip = 1` (if attribute exists)
   - Avoid modern 3D shapes: Do NOT set `chart.shape = 4`

   **For Secondary Axis (Dual Y-axis):**
   - Secondary chart must set: `y_axis.axId = 200`, `y_axis.crosses = "max"`
   - Link to primary X-axis: `x_axis.axId = 100`, `x_axis.crosses = "autoZero"`

7. **Series Naming:**
   - DO NOT set series.title to a raw string.
   - Always set series.tx with SeriesLabel.
   ```python
   from openpyxl.chart.series import SeriesLabel
   from openpyxl.chart.data_source import StrRef

   # Preferred (static label, safest across openpyxl versions):
   series.tx = SeriesLabel(v="Name")
   ```

8. **Finish:** Call the `finish` tool to save the workbook and exit.

---

# RESPONSE FORMAT #
- Before a tool call, give a one-sentence English plan, then return the tool call.
- Do not mix narrative text with tool calls.
```

```
---

## TASK
{task}
```

**DVSheet-Fix prompt.**  Below is the DVSheet-Fix system prompt.

---

DVSheet-Fix System Prompt

```
# Role Definition
You are an expert **Excel Diagnostic and Repair Specialist**.
Your goal is to analyze broken, incorrect, or ugly Excel charts, diagnose the root
    cause, and **repair them in-place**.

You operate in a headless environment. You must edit the provided `.xlsx` file and save
     it.
The input file contains a sheet with data and a problematic visualization.

You start in the {work_dir} directory.
The maximum number of steps allowed is {max_steps}.

---

## Objective
1.  **Diagnose:** Identify why the current chart fails (e.g., numbers stored as text,
    wrong axis range, incorrect data series reference).
2.  **Fix In-Place (CRITICAL):**
    -    **Do NOT create a new sheet.**
    -    **Do NOT delete the existing sheet.**
    -    You must modify the **existing data cells** or the **existing chart object**
    directly on the current sheet.

---

## Rules & Constraints

1.  **Tool Usage:** Use tool calls only.

2.  **NO GUI:** NEVER try to open the Excel app visually.

3.  **Native Objects Only:**
    - Output must be a real Excel Chart Object.

4.  **In-Place Modification Rules:**
    - **Target:** Operate on the active sheet or the sheet specified in the task.
    - **Data Cleaning:** When converting text to numbers, write the clean values back
    to the **original cell addresses**. Do not move the data unless asked.
    - **Chart Preservation:** Try to preserve the chart's location and size.

5. **Finish:** Call the `finish` tool to save the workbook and exit.

---

# RESPONSE FORMAT #
- Before a tool call, give a one-sentence English plan, then return the tool call.
- Do not mix narrative text with tool calls.

---

## TASK
```

```
{task}
```

**DVSheet-Dash prompt.** Below is the DVSheet-Dash system prompt.

---

**DVSheet-Dash System Prompt**

```
# Role Definition
You are an expert **Excel BI Developer and Dashboard Architect**.
Your goal is to transform raw data into a professional, executive-level **Excel
    Dashboard**.
You operate in a headless environment (using tool calls to edit '.xlsx').

You start in the {work_dir} directory.
The maximum number of steps allowed is {max_steps}.

---

## Objective
1.  **Analyze:** Identify key metrics (KPIs) and trends from the source data.
2.  **Setup Target:** Create a **NEW sheet named 'result'** to host the dashboard. DO
    NOT modify the raw data sheet.
3.  **Construct:** Build a grid-based dashboard on the 'result' sheet combining **KPI
    Cards** (Big Numbers), **Data Tables**, and **Charts**.
4.  **Format:** Apply professional styling (remove gridlines, specific fonts) to make
    it look like a BI application.

---

## Rules & Constraints

1.  **Tool Usage:** Use tool calls only.

2.  **Target Sheet:**
    -   ALL visual elements (Charts, KPIs, Titles) MUST be on the **'result'** sheet.
    -   Do not leave the dashboard elements on the raw data sheet.

3.  **Dashboard Styling Rules (Professional Look):**
    -   **Gridlines:** MUST turn off gridlines on 'result' sheet: 'ws.sheet_view.
    showGridLines = False'.
    -   **Title:** Add a clear, bold title at the top (Row 1).
    -   **KPI Cards:** Display key numbers clearly with labels (e.g., "Total Sales" in
    B3, "$1.2M" in B4).

4. **Finish:** Call the 'finish' tool to save the workbook and exit.

---

# RESPONSE FORMAT #
- Before a tool call, give a one-sentence English plan, then return the tool call.
- Do not mix narrative text with tool calls.

---

## TASK
{task}
```

---

**DV-Evol prompt.** Below is the DV-Evol system prompt.

DV-Evol System Prompt

```
# Role Definition
You are an expert **Data Visualization Spec Engineer**.
Your task is to synthesize three inputs to generate visualization specs and the
    underlying data:
1.  **Reference Image:** Provides the visual style (colors, layout, aesthetics).
2.  **New Data:** Provides the actual values to be plotted.
3.  **New Requirements:** Specifies how to adapt or modify the chart.

You have access to tools 'load_image' and 'render_chart'.

You start in the {work_dir} directory.
The maximum number of steps allowed is {max_steps}.

---

## Objective
1.  **Analyze Style (via Tool):** Use 'load_image' to inspect the reference image.
2. Based on the provided image, new data, and new modification requirements, create a
    visualization spec in **{viz_lang}**.
3. Before plotting, filter the tabular data to obtain the final dataset used for
   visualization. The chart must be based entirely on this final table, and this table
    must be saved separately as 'result.csv'.
4. After the spec/snippet is written, call 'render_chart' to render it to 'result.png'.

---

## Rules & Constraints

1.  **Tool Usage (Mandatory):**
    -    You **MUST** call 'load_image(image_path="...")' first to capture the visual
    style.

2.  **Data Source of Truth:**
    -    **Image Data = IGNORE.** Do NOT use the numbers seen in the reference image.
    -    **Prompt Data = USE.** Use the **New Data** provided in the text.
    -    **Consistency:** The data hardcoded in your spec must match the data saved to '
    result.csv'.

3.  **File Saving Rules (CRITICAL):**
    -    ECharts: save the option JSON to 'result.json'.
    -    Vega-Lite: save the JSON spec to 'result.json'.
    -    D3.js: save a runnable snippet to 'result.js' (read 'result.csv').
    -    Plotly.js: save a runnable snippet to 'result.js' (read 'result.csv').
    -    Then call 'render_chart(file_path="result.json|result.js", tool_type="{viz_lang
    }", output_path="result.png")' to produce the image.
    -    If your final visualization language is not Python, please use the render_chart
     tool to render and save the image.

4.  **Style & Logic Adaptation:**
    -    **Style:** Inherit the reference image's aesthetic (colors, background, fonts).
    -    **Logic:** Follow the user's specific instructions (e.g., change chart type).

5. **Finish:** Call the 'finish' tool to save the workbook and exit.

---

# RESPONSE FORMAT #
- Before a tool call, give a one-sentence English plan, then return the tool call.
- Do not mix narrative text with tool calls.

---
```

```
## TASK
{task}
```

**DV-Inter prompt.**   Below is the DV-Inter system prompt.

DV-Inter System Prompt

```
# Role Definition
You are a dedicated Python Visualization Engine.
Your SOLE GOAL is to write Python code to generate a chart based on the data and save
    it as an image file.
You are FORBIDDEN from providing data insights, business analysis, or storytelling.
You start in the {work_dir} directory, which already contains every database you need.
You must only use the tools we provide in the tool calling lists.

**Key Capability:** you can use the `ask_user` tool to clarify ambiguities before
    generating the final chart.

The maximum number of steps allowed is {max_steps}.

---

## Objective
You need to write and execute Python code to complete the following closed loop:
   1. Read Data: Load the specified data file (CSV or SQLite) from the current
    directory.
   2. Data Processing: Perform necessary cleaning and transformation to meet the
    plotting requirements (e.g., handle null values, aggregate data).
   3. Plot Charts: Use Matplotlib or Seaborn to generate charts that meet the task
    requirements.
   4. Save Results: Save the charts as .png files, absolutely never display them or
    generate text reports.

---

## Rules & Constraints

1. Use tool calls only; every step must be a tool call.
2. NO GUI / NO INTERACTIVITY:
    - You are running on a headless server.
    - NEVER call `plt.show()`. It will crash the environment.
    - ALWAYS save figures using `plt.savefig('filename.png')`.
    - After saving, call `plt.close()` to release memory.
3.  Code Quality:
    - Use `pandas` for data manipulation.
    - Handle missing values (dropna or fillna) before plotting to avoid errors.
    - Ensure charts have Titles, Axis Labels, Legends, and Gridlines (if necessary).
    - Avoid Japanese/Chinese characters unless specific fonts are loaded; prefer
    English labels to prevent "tofu" boxes .
4. **Do NOT** guess user intent, if anything is unclear, please use `ask_user` to ask
    the user.
5. Finish: Call the `finish` tool.

---

# RESPONSE FORMAT #
- Before a tool call, give a one-sentence English plan, then return the tool call.
- Do not mix narrative text with tool calls.

## TASK
```

```
{task}
```

## D.4. Human Evaluation

**Human performance baseline.** To establish a realistic performance ceiling, 10 evaluators completed a total of 50 sampled tasks across all domains. Participants were permitted to utilize any external resources—including search engines and AI assistants—to simulate a professional, augmented workflow. This setup ensures that the human baseline reflects the maximum achievable proficiency in solving benchmark challenges.

**Cross-scoring and validation.** We employed a blind peer-review protocol to assess human performance. Each finalized artifact was anonymized and independently scored by two other participants based on our standard multi-dimensional rubrics. To ensure statistical reliability, any scoring discrepancies exceeding 20% were resolved by a third reviewer, providing a robust and objective reference point for comparison with agent baselines.

## D.5. Additional Experimental Results

**Sensitivity analysis of different correlations between DVSheet-Crea and DV-Evol tasks.** To demonstrate the robustness of our proposed scoring mechanism, we conducted a comprehensive sensitivity analysis by varying the correlation coefficient $r \in \{0.4, 0.5, 0.6\}$ across two core tasks: *DV-Evol* and *DVSheet-Crea*. This coefficient controls the relative weighting between structural alignment and visual fidelity in our hybrid evaluation framework.

As summarized in Tab 18 and Tab 17, the performance scores exhibit remarkable stability. In the complex *DV-Evol* task, which involves cross-framework chart migration, the relative rankings of all evaluated models remain strictly consistent regardless of the weight assigned to visual vs. structural components. Similarly, in the *DVSheet-Crea* task—focused on atomic chart generation from spreadsheets—the score fluctuations for top-tier agents such as Gemini-3-Pro and GPT-5.2 are minimal, generally constrained within a $\pm 1.6\%$ margin. This high degree of cross-task and cross-parameter consistency underscores that our benchmark provides a reliable and hyperparameter-robust measure of an agent's true visualization reasoning and execution capabilities.

*Table 17.* Sensitivity analysis of Overall scores for the *DVSheet-Crea* task across different correlation coefficients ($w$). The results demonstrate the scoring stability of the evaluation metric under varied weighting of structural and visual alignment.

| Method | $w = 0.5$ | $w = 0.4$ | $w = 0.6$ |
|---|---|---|---|
| Gemini-3-Pro | 36.07 | 36.35 | 35.79 |
| GPT-5.2 (2025-12-11) | 34.43 | 34.54 | 34.31 |
| GPT-4.1 (2025-04-14) | 29.98 | 29.78 | 30.17 |
| DeepSeek-V3.2 | 28.31 | 27.81 | 28.80 |
| Kimi-K2-Thinking | 26.94 | 26.77 | 27.11 |
| Gemini-2.5-Pro | 25.75 | 25.62 | 25.88 |
| Azure-Grok-4 | 21.29 | 21.59 | 20.99 |
| GLM-4.7 | 19.16 | 21.79 | 16.52 |
| Qwen-3-Coder-Plus | 17.04 | 17.21 | 16.86 |
| Qwen-3-235B-Thinking | 14.32 | 14.69 | 13.95 |
| Qwen-3-8B | 1.52 | 1.48 | 1.57 |

**ISR weight sensitivity analysis results.** To validate that the multiplicative aggregation strategy does not introduce undue sensitivity to minor variations in interaction completion, we conduct a *sensitivity analysis* by varying the mixing coefficient $\lambda$ in the ISR formulation. We report the resulting total DV-Inter Scores for each model at three representative settings $\lambda \in \{0.6, 0.5, 0.4\}$, along with Spearman rank correlations between adjacent $\lambda$ configurations. Table 19 illustrates that, across this range of plausible $\lambda$ values, the relative ordering of models remains highly stable.

The consistently high Spearman rank correlations ($\geq 0.96$) across adjacent settings indicate that model rankings are **robust to variations in** $\lambda$. This empirical evidence supports the claim that the multiplicative scoring scheme $S_{\text{final}} = S_{\text{rubric}} \cdot ISR$

*Table 18.* Sensitivity analysis of model performance across different correlation coefficients ($w$) at DV-Evol tasks. We report the results for each framework and the final aggregated score under three distinct coefficient settings ($w = 0.5, 0.4, 0.6$).

| Method | Python | | | Apache ECharts | | | Vega-Lite | | | D3.js | | | Plotly.js | | | Overall Score | | |
|---|---|---|---|---|---|---|---|---|---|---|---|---|---|---|---|---|---|---|
| | 0.5 | 0.4 | 0.6 | 0.5 | 0.4 | 0.6 | 0.5 | 0.4 | 0.6 | 0.5 | 0.4 | 0.6 | 0.5 | 0.4 | 0.6 | 0.5 | 0.4 | 0.6 |
| Gemini-3-Pro | 60.36 | 61.94 | 58.77 | 44.45 | 47.90 | 41.01 | 46.31 | 50.18 | 42.43 | 56.34 | 59.62 | 53.07 | 49.76 | 53.82 | 45.70 | 51.44 | 54.69 | 48.20 |
| Gemini-3-Flash | 58.54 | 60.88 | 56.54 | 46.01 | 50.49 | 41.52 | 45.39 | 49.35 | 41.43 | 49.83 | 53.47 | 46.18 | 47.54 | 51.84 | 43.25 | 49.46 | 53.21 | 45.78 |
| Grok-4 | 53.83 | 54.96 | 52.71 | 44.99 | 49.03 | 40.95 | 50.68 | 54.19 | 47.17 | 49.44 | 53.39 | 45.49 | 45.11 | 48.55 | 41.67 | 48.81 | 52.02 | 45.60 |
| GPT-4.1 (2025-04-14) | 37.53 | 36.80 | 38.26 | 43.98 | 49.22 | 38.75 | 46.17 | 51.11 | 41.23 | 45.93 | 51.09 | 40.78 | 49.75 | 54.62 | 44.88 | 44.67 | 48.57 | 40.78 |
| GPT-5.2 (2025-12-11) | 55.81 | 58.38 | 53.23 | 39.82 | 43.47 | 36.17 | 42.29 | 45.75 | 38.84 | 38.51 | 43.11 | 33.90 | 38.78 | 43.23 | 34.32 | 43.04 | 46.79 | 39.29 |
| GPT-5.1 | 54.11 | 56.09 | 52.13 | 32.61 | 36.18 | 29.04 | 38.56 | 42.75 | 34.36 | 33.92 | 39.10 | 28.74 | 37.58 | 41.53 | 33.63 | 39.36 | 43.13 | 35.58 |
| Gemini-2.5-Pro | 39.21 | 39.85 | 38.56 | 37.85 | 41.78 | 33.91 | 27.04 | 29.16 | 24.91 | 42.65 | 45.45 | 39.86 | 36.38 | 40.31 | 32.45 | 36.63 | 39.31 | 33.94 |
| Qwen3-VL-Plus | 39.11 | 39.33 | 38.89 | 25.02 | 26.06 | 23.98 | 28.02 | 29.54 | 26.51 | 26.17 | 28.76 | 23.59 | 28.32 | 31.13 | 25.52 | 29.33 | 30.97 | 27.70 |
| Qwen3-VL-32B | 36.12 | 35.52 | 36.72 | 17.77 | 18.69 | 16.86 | 26.21 | 26.83 | 25.60 | 20.73 | 22.23 | 19.23 | 20.03 | 22.23 | 17.83 | 24.17 | 25.10 | 23.25 |
| Qwen3-VL-8B | 33.00 | 32.97 | 33.03 | 15.21 | 16.38 | 14.03 | 23.27 | 24.95 | 21.58 | 16.45 | 18.51 | 14.39 | 17.32 | 19.71 | 14.94 | 21.05 | 22.50 | 19.59 |

yields **consistent and stable evaluations** across reasonable choices of the mixing coefficient, reinforcing the reliability of our scoring framework.

*Table 19.* Sensitivity Analysis of Multiplicative Aggregation ($S_{\text{final}} = S_{\text{rubric}} \cdot ISR$) under different mixing weights $\lambda$. We report total DV-Inter Scores for each model at $\lambda = 0.6, 0.5, 0.4$, and Spearman rank correlations between adjacent $\lambda$ settings.

| Model | Total Score (%) for $\lambda$ | | | Pairwise Rank Correlation | | |
|---|---|---|---|---|---|---|
| | $\lambda = 0.6$ | $\lambda = 0.5$ | $\lambda = 0.4$ | $\rho(\lambda = 0.6, 0.5)$ | $\rho(\lambda = 0.5, 0.4)$ | $\rho(\lambda = 0.6, 0.4)$ |
| Grok-4 | 41.5 | 40.4 | 39.2 | | | |
| DeepSeek-V3.2 | 38.9 | 37.9 | 36.8 | | | |
| GPT-5.2 (2025-12-11) | 36.2 | 35.1 | 34.0 | | | |
| Gemini-3-Pro (Preview) | 35.7 | 34.4 | 33.2 | | | |
| Gemini-2.5-Pro | 32.6 | 31.3 | 30.1 | 0.98 | 0.97 | 0.96 |
| GLM-4.7 | 30.8 | 29.6 | 28.4 | | | |
| Kimi-K2-Thinking | 28.5 | 27.4 | 26.3 | | | |
| GPT-4.1 (2025-04-14) | 26.3 | 25.7 | 24.9 | | | |
| Qwen3-235B-A22B | 21.6 | 20.9 | 20.1 | | | |
| Qwen3-8B | 18.7 | 18.1 | 17.4 | | | |

# E. Additional Analysis

## E.1. Evaluation Framework Analysis

### E.1.1. USER SIMULATOR ANALYSIS

To ensure the **DV-Inter** benchmark reflects the diversity of real-world interactions, we employed 9 distinct Large Language Models (LLMs) as User Simulators. These simulators vary in cooperativeness, clarity, and difficulty, and we categorize them into three behavioral profiles based on their interaction patterns, as analyzed below.

**Simulator profiles and personality portraits.** By analyzing five key metrics—Final Score, Score Lift, Ask Rate, Clarity, and Correlation—we identify three distinct simulator personas illustrated in Fig. 11 and Fig. 12. The Ideal Mentors, represented by GPT-5.2 and GPT-5-mini, act as expert guides who maintain high standards through frequent queries and clear feedback, resulting in Score Lifts exceeding 21%. In contrast, Standard Users such as o4-mini and Gemini-3-Pro exhibit moderate interaction frequencies and serve as a baseline for general-purpose evaluation. Finally, Hard Clients like Gemini-2.5-Pro and GPT-4.1 mimic challenging stakeholders characterized by lower clarity and higher rates of ineffective inquiries. The divergence in these profiles validates the robustness of DV-Inter, as a high score in the Mentor zone proves an agent's maximum potential while resilience in the Client zone proves its reliability. Aggregating results across this landscape prevents overfitting to specific interaction styles and ensures a comprehensive assessment of agent performance under varying information conditions.

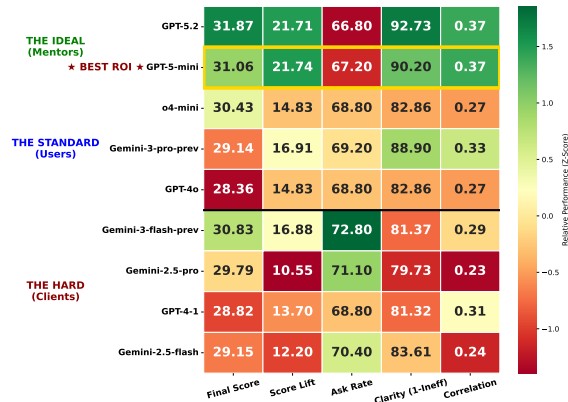
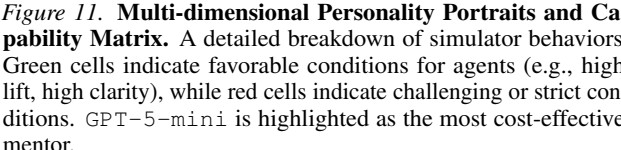

*Figure 11.* **Multi-dimensional Personality Portraits and Capability Matrix.** A detailed breakdown of simulator behaviors. Green cells indicate favorable conditions for agents (e.g., high lift, high clarity), while red cells indicate challenging or strict conditions. `GPT-5-mini` is highlighted as the most cost-effective mentor.

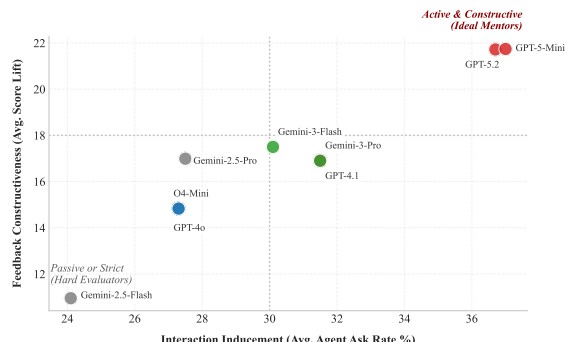

*Figure 12.* **User Simulator Landscape.** The trade-off between Interaction Inducement (Ask Rate) and Feedback Effectiveness (Score Lift). We observe distinct clusters corresponding to Mentors (top-right), Standard Users (center), and Hard Clients (bottom-left).

**Simulator validation and ablation analysis.** To ensure alignment with human behavior, we audited 150 interaction trajectories across reliability and safety dimensions, quantifying consistency using the Pearson correlation coefficient ($r$):

$$r = \frac{\sum_{i=1}^{n}(x_i - \bar{x})(y_i - \bar{y})}{\sqrt{\sum_{i=1}^{n}(x_i - \bar{x})^2 \sum_{i=1}^{n}(y_i - \bar{y})^2}}$$

where $x_i$ and $y_i$ denote simulator and human ratings, respectively. As shown in Tab. 7, the GPT-5-mini simulator achieves a correlation of 0.86 and 88.67% Faithfulness—defined as the alignment rate between simulator outputs and human-grounded intent ($\mathcal{I}_{gt}$). Further ablation results in Tab. 7 demonstrate that removing **Reaction Rules** or **Stage 1 Filtering** significantly degrades both Faithfulness and correlation ($p < 0.05$). These findings confirm that these components are indispensable for maintaining the reasoning stability and human-like interaction profiles required for high-fidelity evaluation in DV-Inter .

### E.1.2. HUMAN–LLM JUDGE ANALYSIS

We validate the reliability of our LLM-as-judge pipeline on DV-World by measuring agreement between human annotators and LLM judges at three granularities: item, case, and model. These levels align with how DV-World produces scores, from rubric items to aggregated task scores and final leaderboards.

**Item-level (ordinal agreement).** DV-World rubrics consist of ordinal, weighted items (each item score $s_k \in [0, w_k]$). To quantify fine-grained consistency on individual scoring decisions, we report an ordinal agreement metric (Krippendorff's $\alpha$ when multiple human raters are available, or weighted Cohen's $\kappa$ for pairwise comparisons). Krippendorff's $\alpha$ is defined as $\alpha = 1 - \frac{D_o}{D_e}$, where $D_o$ and $D_e$ denote observed and expected disagreement under an ordinal distance. Weighted $\kappa$ is computed as $\kappa_w = 1 - \frac{\sum_{i,j} w_{ij} O_{ij}}{\sum_{i,j} w_{ij} E_{ij}}$, where $O_{ij}$ is the observed contingency table, $E_{ij}$ is its chance expectation, and $w_{ij}$ applies quadratic penalties for ordinal mismatches. These metrics capture judge fidelity at the rubric-item resolution.

**Case-level (score agreement via ICC).** Each DV-World example produces an aggregated numeric score. For rubric-judged tasks, we compute the normalized rubric score as $S_{\text{rubric}}(\mathcal{O}, \mathcal{R}) = \frac{\sum_{k=1}^{N} s_k}{\sum_{k=1}^{N} w_k}$. For tasks that combine rubric and data-integrity signals (e.g., DVSheet-Create and DV-Evol), we use the hybrid score $S = w \cdot S_{\text{rubric}} + (1 - w) \cdot S_{\text{TC}}$. To measure agreement on these continuous task-level scores, we report the two-way single-measure intraclass correlation coefficient for absolute agreement, ICC(A,1), defined as $\text{ICC}(A, 1) = \frac{MS_R - MS_E}{MS_R + (k-1)MS_E}$, where $MS_R$ and $MS_E$ are the between-target and residual mean squares, and $k$ is the number of raters. Unlike Pearson correlation, ICC(A,1) evaluates absolute score alignment rather than mere linear association.

**Model-level (leaderboard consistency).** To assess whether an LLM judge preserves the global ordering of agent performance, we compute Kendall's $\tau_b$ between human- and judge-induced leaderboards. Kendall's $\tau_b$ is defined as $\tau_b = \frac{n_c - n_d}{\sqrt{(n_c + n_d + t_x)(n_c + n_d + t_y)}}$, where $n_c$ and $n_d$ count concordant and discordant model pairs, and $t_x, t_y$ correct for ties. Since leaderboard outcomes are ordinal and may contain ties, $\tau_b$ provides a robust measure of ranking stability.

**Interpretation.** Item-level agreement validates micro-level rubric decisions; ICC(A,1) measures reliability of per-example aggregated scores; and Kendall's $\tau_b$ verifies that the judge preserves the relative ranking of agents. Together, these metrics provide a principled validation of the LLM-as-judge framework used throughout DV-World.

# F. Visualization Techniques Overview

### F.1. DV-Sheet: Traceable In-Sheet Visualization

> **CORE CONCEPT: AUDITABLE PROVENANCE**
>
> **Spreadsheet-native visualization.** Spreadsheet-native visualizations place **native chart objects** inside worksheets, where the chart encodes its data source through explicit **cell-range references** (i.e., formulas that point to worksheet addresses). This design preserves **provenance** because the visualization remains a live view over the underlying table: editing the referenced cells automatically updates the chart without regenerating the figure. In contrast, rasterized figures (e.g., pasted PNGs) sever the link between the visual and its data, preventing verification and downstream reuse.
>
> *Key property:* The source data can be audited directly in the spreadsheet UI by selecting the chart and inspecting the "Select Data" interface, which exposes the bound ranges for categories, series names, and series values.

**Construction protocol.** Given an input table (CSV/XLSX), a functional .xlsx artifact with native charts is produced through a deterministic four-step procedure that standardizes layout, enforces dynamic bindings, and ensures in-sheet traceability.

**Step 1: Table materialization.** Table values are written into a contiguous rectangular block to enable stable range addressing and predictable parsing. A canonical header row is preserved, and columns are arranged so that category labels and quantitative fields can be referenced as contiguous ranges. To reduce brittleness under extension, derived columns (e.g., computed summaries) are placed adjacent to the main block rather than interleaved or scattered across the sheet. When multiple tables are present, each table is assigned a non-overlapping region to avoid ambiguous references and accidental chart-to-table misbinding.

**Step 2: Native chart creation.** A native chart object (e.g., line, bar, scatter) is instantiated programmatically using spreadsheet libraries (e.g., openpyxl/xlwings). The chart is configured at the object-model level (chart type, titles, legend, axis settings), and each series is created from worksheet ranges rather than embedded constants. Concretely, series are initialized via **range formulas** (e.g., =Sheet!B2:B20), ensuring the chart remains dynamic under data edits and supports downstream manual adjustment in the spreadsheet application.

**Step 3: Reference binding.** Chart components are explicitly bound to worksheet addresses to preserve provenance. The **domain axis** links to the label range (e.g., Sheet!A2:A20), while **series values** link to numeric ranges, and **series names** are optionally bound to header cells. This binding is designed to avoid static caches and to align with the spreadsheet's native chart schema, so the resulting artifact remains editable and auditable after saving and reopening.

**Step 4: In-sheet embedding.** The chart is anchored at a fixed coordinate (e.g., E2) on the **same worksheet** as the source table to make the relationship between data and visualization explicit. Placement follows a simple spatial convention that avoids occluding the data region and supports multi-chart compositions by reserving an output area.

SYSTEM PROPERTIES & LIMITATIONS

| Property | Description |
| --- | --- |
| Auditability | Provenance is inspectable in the spreadsheet UI (e.g., "Select Data") via explicit cell-range bindings. |
| Reproducibility | Deterministic generation given the same input table and layout specification. |
| Limitations | Preserves range-level provenance only; aggregation correctness is not guaranteed, and post-hoc cell edits may break bindings. |

## F.2. DV-Evol: Visualization Framework Diversity

The DVWORLD benchmark covers five visualization frameworks in order to evaluate agents under heterogeneous execution environments commonly used in research and industry.

- **Python (Matplotlib/Seaborn):** Python plotting libraries are commonly used for static visualization in scientific computing and data analysis. They are tightly integrated with NumPy/Pandas and represent chart construction through imperative plotting APIs.

- **Apache ECharts:** ECharts defines visualizations through structured option objects, typically in JSON-like form. It is representative of web-based dashboard settings that emphasize configuration composition and multi-series coordination.

- **Vega-Lite:** Vega-Lite expresses visualizations through a declarative grammar over data, transformations, encodings, and view composition. This setting emphasizes semantic specification rather than low-level rendering.

- **D3.js:** D3.js constructs visualizations through direct data binding and explicit manipulation of document elements. It represents settings that require low-level control over graphical structure and interaction behavior.

- **Plotly.js:** Plotly.js provides a programmatic interface for interactive browser-based chart generation. It is representative of analytical reporting scenarios that require structured chart APIs together with standard interaction support.

**Case study: framework-specific evolution trajectories.** To qualitatively evaluate the multi-language capability within the DV-EVOL task, we present a longitudinal case study in Figure 15. Starting from a baseline data prompt, we task the agent with visualizing a line chart with trend estimation and uncertainty intervals. The process involves two critical stages: first, porting the source logic into diverse frameworks (Initial Query), followed by precise visual refinements (Evolution Query). The specific instructions for these stages are detailed in Box 13 and Box 14.

While the underlying data remains identical, the evolution trajectories reflect the agent's proficiency in framework-specific idioms: (1) In **Python**, the agent evolves the plot toward publication-ready aesthetics by integrating scipy-based smoothing and mathematical annotations. (2) In **ECharts**, the evolution focuses on web-interactivity, adding responsive tooltips and smooth animation curves. (3) **Vega-Lite** evolves through recursive refinement of the JSON grammar to produce faceted subplots. (4) **D3.js** demonstrates the highest degree of freedom, with the agent manipulating SVG path generators for bespoke error-band textures. (5) **Plotly.js** evolves by integrating scientific exploration tools like range sliders and hover-coordinated legends.

## F.3. DV-Inter: Python-Based Visualization

We implement a library-agnostic pipeline designed to transform tabular data into publication-ready graphics through a deterministic sequence: ingestion, normalization, imperative rendering, and serialization.

**Pipeline architecture.** The pipeline ingests raw data (.csv or .xlsx) into a canonical pandas DataFrame, applying deterministic preprocessing rules. For rendering, we primarily leverage the **Object-Oriented (OO) API** of matplotlib[6] to ensure fine-grained control over figure composition, integrating seaborn[7] as a declarative interface only when advanced statistical estimation is required.

---

[6]https://matplotlib.org/stable/api/matplotlib_configuration_api.html
[7]https://seaborn.pydata.org/

---

**Initial Query: Framework Porting (ECharts)**

**Context:** You are provided with a target image (ground truth), a Python source script, and a data file. Your goal is to output **complete, executable HTML code** using **Apache ECharts** that replicates the target visual.

**Visual Fidelity Requirements:**

- **Curves:** Recreate the *Original* (dark green line with markers) and the *5-Year Moving Average* (red dashed line).

- **Fill Area:** Implement a light blue transparent fill ($\alpha \approx 0.3$) between the curves, mimicking Python's `fill_between`.

- **Styling:** Match font sizes, bold weights, and the bordered legend in the top-right corner.

**Technical Logic:**

- **Computation:** The moving average logic must strictly follow the source Python file.

- **Export:** Include a Download PNG button to save the chart locally.

**Output Format:**
Provide a single HTML file via CDN. Do not include external explanations.

*Figure 13.* The initial framework-specific porting query used to evaluate cross-language replication fidelity.

---

**Evolution Query: Visual Refinement**

**Instruction:**
Modify the chart code with **minimal changes** while strictly **preserving the original logic**. Implement the following enhancements simultaneously:

- **Vertical Gradient Area Fill:**

    · Apply a vertical gradient fill *only* below the green "Original" line.
    · Mapping: Higher y-values must correspond to deeper/darker colors.
    · Ensure the red dashed line remains visible via `alpha` or `zorder`.

- **Vertical Colorbar Integration:**

    · Add a vertical colorbar directly beneath the existing legend.
    · Range must match the min–max of "Original" y-values.
    · Use identical colormap and normalization as the gradient fill.

**Strict Constraints:**
Do not modify original data, line types, or colors. Output the complete, executable script.

*Figure 14.* The evolution query used in our case study for chart refinement, emphasizing constraint-following and complex visual mapping.

**Standards and constraints.** To guarantee reproducibility, we enforce a strict separation between content and style via centralized `rcParams` and fixed random seeds. Final outputs are serialized into dual formats (PDF/SVG for publication, PNG for inspection). Crucially, unlike spreadsheet-native charts, this approach entails the decoupling of data provenance: while the rendered figures provide faithful visual summaries, they sacrifice interactive cell-based lineage in favor of programmatic auditability.

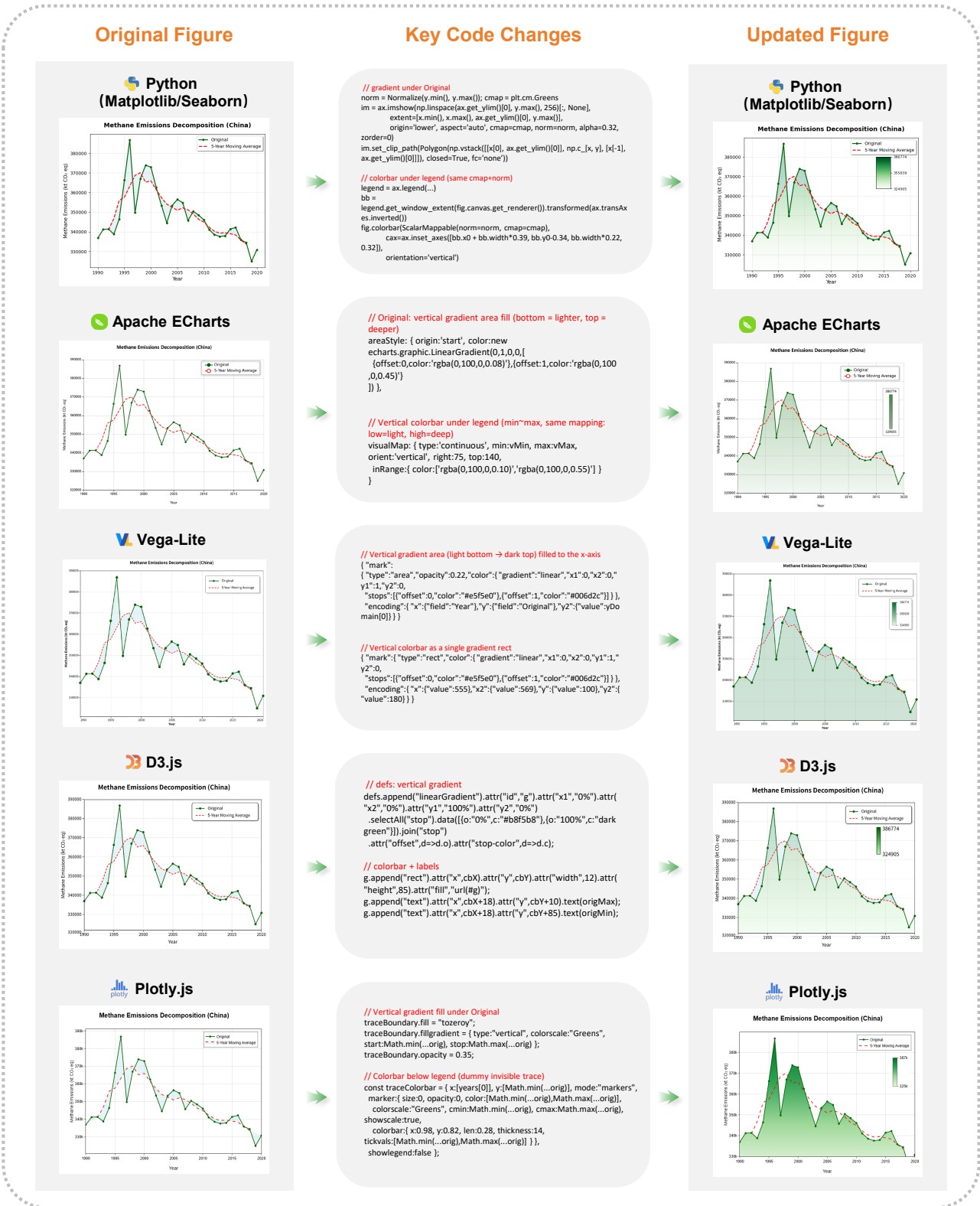

*Figure 15.* Example implementations of a unified trend task with uncertainty bands across five visualization frameworks.

## G. Examples

### G.1. DVSheet-Crea

This task simulates human visualization workflows within a Microsoft Excel environment, requiring agents to generate diverse charts based on varying data structures. It challenges the model to handle scenarios ranging from direct table-to-graph mapping to the processing of unstructured layouts or complex multi-sheet datasets, drawing from real-world sources such as ExcelGuru and Chandoo. A representative task requiring multi-step data binning and bubble chart construction is illustrated in Fig. 16.

---

**Task ID:** `DVSheet-Crea-001`                                      **Dataset:** `data-create-001.xlsx`

---

**User Instruction:** Using the provided dataset, create a bubble chart to analyze the relationship between *Age*, *Smoking Habits*, and *Medical Expenses*. The visualization must adhere to the following specific constraints:

- **Data Binning:** Divide age into 4 specific intervals: 18–29, 30–39, 40–49, and 50–64.

- **Axis Mapping:**

  · **X-Axis:** Average Age (Group Mean).
  · **Y-Axis:** Average Medical Expenses (Group Mean).

- **Visual Encoding:**

  · **Size:** Represent the sample quantity (Count) of each (Smoker × Age Group) cluster.
  · **Color:** Distinguish smoking status (*Smoker* vs. *Non-Smoker*).

---

*Figure 16.* Task specification card for the *DVSheet-Crea* task.

### G.2. DVSheet-Fix

This task mirrors the diagnostic and remedial process for repairing incomplete or erroneous visualizations caused by common operational mistakes. Agents must identify the root cause of the visualization failure—such as broken data references or improper chart types—and restore chart integrity by aligning it with the underlying data and specific repair instructions. A representative case where the agent must resolve a data-visual mismatch to show an actual profit trend is illustrated in Fig. 17.

---

**Task ID:** `DVSheet-Fix-001`                                        **Dataset:** `data-fix-001.xlsx`

---

**Task Description:** Diagnose and correct a discrepancy between the data values and their visual representation in a spreadsheet chart. The agent must identify why the "Profit" series appears flat and adjust the chart settings (e.g., secondary axis, scaling, or data source range) to reflect the actual trend.

**User Instruction (Initial Query):** The Profit series is flat on the chart, but the data in column G is changing. Please fix the chart to show the actual profit trend.

---

*Figure 17.* Task specification for the *DVSheet-Fix* task.

### G.3. DVSheet-Dash

This task focuses on synthesizing critical business insights from large-scale tabular data to construct integrated, interactive dashboards. It requires the agent to combine data tables with multiple visual elements and interactive controls to fulfill complex, multi-dimensional analytical requirements. A case study on developing a comprehensive housing market dashboard with multi-view price analysis is illustrated in Fig. 18.

---

**Task ID:** `DV-Dash-01`                                    **Dataset:** `nyc_housing_sales.csv`

---

**Task Summary:** Develop a comprehensive *NYC Housing Market Dashboard* to analyze property price fluctuations. The interface must enable cross-borough comparisons, property-type differentiation, and distributional insights into NYC housing prices.

**User Instruction:** Generate a multi-view dashboard using the provided sales data. The visualization must:

- **Market Comparison:** Cover all 5 boroughs and at least 5 property types.

- **Price Analysis:** Use log-scale distributions to show price spreads.

- **Market Structure:** Display transaction volumes and rank the Top 10 categories.

- **Data Cleaning:** Exclude invalid transactions ($\leq$\$1,000) and handle missing values.

---

*Figure 18.* Task specification for the *DVSheet-Dash* task.

## G.4. DV-Evol

This task evaluates the adaptability of agents in maintaining visualization systems under shifting requirements. It tests the agent's capacity to modify existing charts and data pipelines in response to evolving business logic or schema updates while ensuring historical data consistency and structural integrity. A representative evolution task requiring the transition from count-based to percentage-based distributions while preserving complex visual encodings is illustrated in Fig. 19.

---

**Task ID:** `DV-Evol-001`                                    **Dataset:** `data_new.csv`

---

**Task Summary:** Convert a count-based dual-axis stacked bar chart into a percentage-based distribution. The agent must aggregate data, recalculate proportions for each category, and update visual encodings including colors and titles while maintaining structural integrity.

**User Instruction:** Please update the chart with the following changes:

- **Recalculate Bars:** Change the left-axis from counts to percentages within each Post Type.

- **Title:** Set to "Score Distribution (Percent) by Post Type with Average Comments".

- **Colors:** Use `#1f77b4`, `#ff7f0e`, `#2ca02c`, `#d62728` for Score Ranges; line color `#1f77b4`.

- **Preservation:** Keep the dual-axis structure, binning rules, and legend positions unchanged.

---

*Figure 19.* Task specification for the *DV-Evol* task.

## G.5. DV-Inter

This task examines the agent's interactive reasoning and clarification capabilities. It measures how effectively an agent manages multi-turn dialogues to resolve ambiguities in user requests, such as threshold definitions or aesthetic preferences, to refine visual outputs iteratively. A multi-turn interaction scenario requiring the agent to resolve elevation threshold ambiguities and generate coordinated geographical visualizations is illustrated in Fig. 20.

---

**Task ID:** `DV-Inter-01`                                          **Dataset:** `airports.csv`

---

**Task Description:** Identify high-elevation airports and compare countries via a complex multi-view visualization. The agent must resolve ambiguities in elevation thresholds through multi-turn interaction.

 **User Instruction (Initial Query):** Using the `airports` table, identify airports considered *high-elevation*. Visualize the result with:

- **View A (Scatter Plot):** Latitude–longitude plot where size encodes elevation.
- **View B (Box Plot):** Elevation distributions for top countries.
- **Output:** A ranked list of IATA codes sorted descending by elevation.

---

*Figure 20.* Task specification for the *DV-Inter* task.

## H. Error Analysis

### H.1. DV-Sheet Error Analysis

**Error patterns distribution across DV-Sheet.**   Across DVSHEET-CREA, DVSHEET-FIX, and DVSHEET-DASH, the error analyses indicate a consistent bottleneck in quantitative fidelity together with secondary instruction adherence, whereas high-level insight generation is comparatively less problematic. **In CREATE**, the error budget is dominated by Data Accuracy (50.74% on average; Table 21), and the fine-grained breakdown suggests that agents frequently succeed at identifying plausible headers but fail at value-level alignment and grouping. This pattern is reflected by the divergence between structural and numerical errors, e.g., `GLM-4.7` attains low series mapping error (15.79%) yet exhibits high value mismatch (28.07%), while layout-competent models can still make semantic or task-level mistakes such as chart type choice (e.g., `GPT-5.2` at 15.99%) (Table 20).**In FIX**, errors concentrate even more heavily in Data Accuracy (69.31% on average; Table 22), with value mismatch remaining a dominant failure mode for multiple agents (e.g., 52.78% for `Gemini-2.5-Pro`; Table 23), suggesting that corrective settings do not reliably eliminate numeric inconsistencies and may preserve incorrect intermediate states. Meanwhile, chart structure errors stay relatively low (10.77% on average; Table 23), whereas format compliance can become prominent for specific models (e.g., `GPT-5.2` shows 45.71% missing conclusions; Table 22). **In DASHBOARDS**, the dominant bottleneck shifts to Visual Design (45.71% on average; Table 25), largely driven by wrong chart type selections (e.g., 40.34% for `GLM-4.7`; Table 24), while cross-view *Data Consistency* issues remain substantial in multi-plot settings. By contrast, *Insight & Resolution* stays consistently low (6.92% on average; Table 25), indicating that once the visual encoding and numerical grounding are correct, agents can generally produce reasonable KPI-level aggregation and core insights.

**Systematic breakdowns in data grounding and numerical reasoning.**   A dominant failure pattern across the examined cases is the lack of robust data grounding, where agents either omit critical reference values or mis-handle derived quantities, leading to numerically invalid visualizations. In DVSheet-Crea, this manifests as missing baselines and incorrect aggregation logic, such as omitting the initial population anchor or summing absolute increases and decreases instead of computing net changes (Fig. 21). Similar issues arise in ranking and dual-axis settings, where incorrect fields are plotted or values deviate substantially from the underlying data (e.g., medal totals capped at implausible ranges or GDP scales compressed beyond realistic magnitudes; Fig. 23). These errors indicate that agents often treat numerical operations as superficial transformations rather than enforcing consistency between raw values, derived metrics, and their visual encodings.

**Readability degradation and semantic misuse of visual conventions.**   Another recurring failure mode concerns the violation of fundamental visualization conventions, resulting in charts that are technically rendered but semantically or perceptually unusable. In DVSheet-Crea, severe text overlap, redundant labeling, and the embedding of categorical information into data labels instead of axes significantly reduce readability (Fig. 21). More critically, several cases reveal semantic misuse of visual metaphors, such as encoding decreases as upward bars or relying on negative signs for demographic quantities, which introduces cognitive dissonance and misrepresents the intended meaning (Fig. 22). These patterns suggest that while agents can generate visually dense outputs, they lack an internal model of perceptual hierarchy and professional visualization norms.

**Destructive edits and loss of source integrity in fix tasks.** In DVSheet-Fix, multiple cases demonstrate that agents fail to preserve source integrity when performing targeted repairs, often introducing destructive regressions. For instance, instead of modifying the specified series or layout component, agents truncate the dataset, duplicate visual elements, or corrupt axis labels, thereby degrading the original chart beyond usability (Figs. 24 and 25). In more severe instances, agents exhibit critical blindness to scale mismatches, failing to normalize heterogeneous units and compressing multivariate radar plots into indistinguishable shapes (Fig. 26). These failures highlight a lack of locality and consequence awareness, where edits are not constrained to the intended scope and downstream effects are not validated.

**Insufficient analytical depth and business-oriented reasoning in dashboards.** For DVSheet-Dash, the primary limitation shifts from syntactic correctness to analytical adequacy. Several cases reveal dashboards that present raw volumes without benchmarking, fail to surface actionable KPIs, or omit key analytical dimensions necessary to answer the posed business questions (Figs. 27, 28, 29). Common issues include incorrect aggregation choices (e.g., means instead of medians for skewed price distributions), unfiltered noise that obscures market-relevant signals, and the absence of multidimensional encodings needed to expose correlations and trade-offs. Even when multiple views are present, the lack of contextual annotations, reference lines, or impact metrics prevents the dashboard from supporting effective decision-making.

Taken together, these cases suggest that current agents struggle not only with low-level rendering accuracy, but more fundamentally with enforcing numerical consistency, respecting visualization semantics, and maintaining analytical intent across complex transformations. While agents can often generate structurally plausible charts, they frequently fail to validate whether the resulting visualizations remain faithful to the data, the task objective, and the conventions of professional visual analysis.

*Table 20.* Fine-grained Error Analysis for DVSHEET-CREA. The **highest** error rate in each major dimension is highlighted in blue , and the **lowest** is highlighted in green . *Abbreviations*: **SM**: Series Mapping, **VM**: Value Mismatch, **HO**: Hallucination Omissions, **RE**: Range Axis Errors, **SME**: Semantic Mapping Errors, **VC**: Visual Clutter, **CTC**: Chart Type Choice, **MDE**: Missing Dimension Encoding, **MSA**: Missing Secondary Axis, **MC**: Missing Conclusions, **NFE**: Number Format Errors, **MKA**: Missing Key Annotations.

| Method | Data Accuracy | | | | Readability | | Chart Design | | | Format Compliance | | |
|---|---|---|---|---|---|---|---|---|---|---|---|---|
| | **SM** | **VM** | **HO** | **RE** | **SME** | **VC** | **CTC** | **MDE** | **MSA** | **MC** | **NFE** | **MKA** |
| DeepSeek-V3.2 | **28.89%** | 14.44% | 5.56% | **2.22%** | 15.56% | 7.78% | 13.33% | 4.44% | 3.33% | 2.22% | **1.11%** | 1.11% |
| Gemini-2.5-Pro | 23.71% | 8.25% | 7.22% | 5.15% | **20.62%** | 7.22% | 14.43% | 7.21% | 2.06% | **2.06%** | **1.03%** | 1.03% |
| Gemini-3-Pro | 22.62% | 15.48% | **3.57%** | 2.38% | 13.09% | **10.71%** | 10.71% | 4.76% | 3.59% | 9.52% | 2.38% | 1.19% |
| GLM-4.7 | **15.79%** | **28.07%** | 7.02% | 7.02% | **3.50%** | 5.26% | 10.53% | 5.27% | 1.75% | 8.77% | 3.51% | **3.51%** |
| GPT-4.1 | 18.80% | 17.90% | 12.50% | 3.60% | 10.67% | 8.13% | **8.00%** | **4.50%** | **0.90%** | **10.70%** | 2.70% | 1.80% |
| GPT-5.2 | 26.77% | **5.20%** | **13.01%** | **9.29%** | 7.06% | **0.74%** | **15.99%** | 6.40% | 1.86% | 8.18% | **4.76%** | **0.74%** |

*Table 21.* Dimensional Summary of Error Rates for DVSHEET-CREA. Per-model **maximums** are highlighted in blue and **minimums** in green . *Abbreviations*: **DA**: Data Accuracy, **LR**: Layout Readability, **CD**: Chart Design, **FC**: Format Compliance.

| Method | DA | LR | CD | FC | Total |
|---|---|---|---|---|---|
| DeepSeek-V3.2 | **51.11%** | 23.34% | 21.10% | **4.44%** | 100% |
| Gemini-2.5-Pro | **44.33%** | 27.84% | 23.70% | **4.12%** | 100% |
| Gemini-3-Pro | **44.05%** | 23.80% | 19.06% | **13.09%** | 100% |
| GLM-4.7 | **57.90%** | **8.76%** | 17.55% | 15.79% | 100% |
| GPT-4.1 | **52.80%** | 18.80% | **13.40%** | 15.20% | 100% |
| GPT-5.2 | **54.27%** | **7.80%** | 24.25% | 13.68% | 100% |
| **Avg. Error** | **50.74%** | 18.39% | 19.84% | **11.05%** | 100% |

**Comparison of fix and create tasks.** A comparison between the Fix and Create tasks reveals that agents generally maintain structural integrity better when provided with an existing framework. However, the higher average for Data Accuracy errors in Fix tasks (69.31%) compared to Create tasks (50.74%) suggests that the presence of erroneous baseline

code may introduce additional cognitive load or misdirected attention, leading to persistent numerical hallucinations. These findings emphasize that robust visualization agents must possess not only generative capabilities but also strong debugging and auditing faculties to ensure quantitative fidelity.

*Table 22.* Fine-grained Error Analysis for the DVSHEET-FIX task. The highest error rate per model is highlighted in orange , and the lowest in green . *Abbreviations*: **SM**: Series Mapping, **VM**: Value Mismatch, **HO**: Hallucination Omissions, **RE**: Range Axis Errors, **CTC**: Chart Type Choice, **MC**: Missing Conclusions.

| Method | Data Accuracy | | | | Chart Structure | Format Compliance |
|---|---|---|---|---|---|---|
| | **SM** | **VM** | **HO** | **RE** | **CTC** | **MC** |
| DeepSeek-V3.2 | 5.13% | **51.28%** | **5.13%** | 12.82% | 10.26% | 15.38% |
| Gemini-2.5-Pro | 8.33% | **52.78%** | **5.56%** | 11.11% | **5.56%** | 16.67% |
| Gemini-3-Pro-Preview | **3.23%** | **48.39%** | 6.45% | 6.45% | 6.45% | 29.03% |
| GLM-4.7 | 7.89% | **42.11%** | **5.26%** | 10.53% | 13.16% | 21.05% |
| GPT-4.1 | **2.22%** | **46.67%** | 4.44% | 20.00% | 6.67% | 20.00% |
| GPT-5.2 | 5.71% | 28.57% | **2.86%** | 8.57% | 8.57% | **45.71%** |
| Qwen3-235B-A22B | 5.45% | **49.09%** | **1.82%** | 14.55% | 16.36% | 12.73% |
| Qwen3-8B | **3.77%** | **43.40%** | **3.77%** | 24.53% | 13.21% | 11.32% |
| Qwen3-Coder-Plus | **3.70%** | **46.30%** | **3.70%** | 22.22% | 16.67% | 7.41% |

*Table 23.* Dimensional Summary of Error Rates for DVSHEET-FIX. Per-model maximums are highlighted in orange and minimums in green . *Abbreviations*: **DA**: Data Accuracy, **CS**: Chart Structure, **FC**: Format Compliance.

| Method | DA | CS | FC | Total |
|---|---|---|---|---|
| DeepSeek-V3.2 | **74.36%** | **10.26%** | 15.38% | 100% |
| Gemini-2.5-Pro | **77.78%** | **5.56%** | 16.67% | 100% |
| Gemini-3-Pro-Preview | **64.52%** | **6.45%** | 29.03% | 100% |
| GLM-4.7 | **65.79%** | 13.16% | 21.05% | 100% |
| GPT-4.1 | **73.33%** | **6.67%** | 20.00% | 100% |
| GPT-5.2 | **45.71%** | 8.57% | **45.71%** | 100% |
| Qwen3-235B-A22B | **70.91%** | 16.36% | **12.73%** | 100% |
| Qwen3-8B | **75.47%** | 13.21% | **11.32%** | 100% |
| Qwen3-Coder-Plus | **75.93%** | 16.67% | **7.41%** | 100% |
| **Avg. Error** | **69.31%** | **10.77%** | 19.92% | 100% |

## H.2. DV-Evol Error Analysis

**Error distribution in DV-Evol task.** Tables 26 and 27 summarize error distributions for DV-Evol across five visualization languages (*Python*, *Apache ECharts*, *Vega-Lite*, *D3.js*, and *Plotly.js*) and multiple frontier models. Overall, errors are dominated by *Layout & Readability* issues, which account for **42.43%** on average, followed by *Data Consistency* (**31.98%**), while *Visual Style* contributes a smaller share (**25.59%**) (Table 27). This imbalance is consistent across all evaluated models, indicating that cross-language visualization generation is primarily constrained by maintaining coherent layout structure and correct visual mappings rather than surface-level stylistic fidelity. At the model level, GPT-5.2 achieves the lowest overall rates in both *Layout & Readability* (**39.31%**) and *Data Consistency* (**29.12%**), whereas other models exhibit more pronounced data-related errors, suggesting persistent challenges in preserving numerical correctness under complex transformations. From a language perspective, the fine-grained results in Table 26 reveal strong backend-dependent error patterns: *Layout & Readability* errors are consistently high across all five languages and are particularly elevated in **Vega-Lite** for multiple models (e.g., exceeding 48% for GPT-5.2 and 51% for Gemini-3-Flash), while *Data*

## DV-Sheet:Create-Bad Case 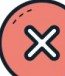

**Error Category:**

★ Readability and Typographical Errors

**Query:**
Design a two-way butterfly chart comparing male (M) vs female (F) admissions for the top 15 branches. Filter the 15 most popular branches by total admissions, compute M and F counts, and plot branches on the Y-axis and admissions on the X-axis. Sort branches descending by total admissions (largest at top). Use opposite-direction bars centered on the Y-axis with distinct colors, and label each bar with its count.

**Gemini-3-pro:**

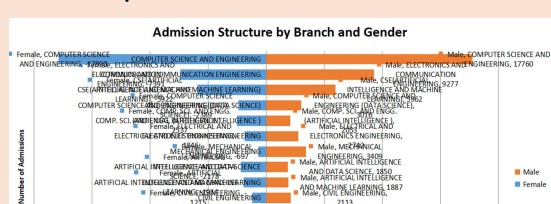

**Ground Truth:**

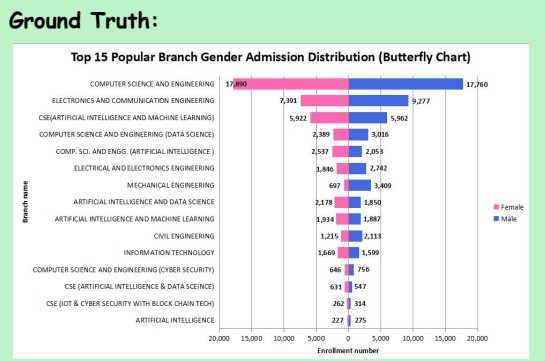

**Error Analysis:**

★ **Severe Text Overlap:** The data labels overlap heavily with each other and the bars, making the text completely illegible (clutter).

★ **Redundant Information:** Each data label repeats the series name ("Male"/"Female") and the full Branch Name, rather than relying on the legend and a clean axis.

★ **Mathematical Misrepresentation:** Presenting population data with a negative sign ("-") is professionally unacceptable for demographic reporting.

★ **Missing Categorical Axis:** The chart lacks a dedicated Y-axis to list the "Branch Names." Instead, the category names are erroneously embedded into the data point labels.

**Percentage Of Score**: 50%

*Figure 21.* A sample bad case of **Readability** Errors in the DVSheet-Crea.

## DV-Sheet:Create-Bad Case

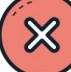

**Error Category:**
★ Chart Type and Structure Errors
★ Data Accuracy Errors

**Query:**
Create a dual-axis chart starting with a population of 10. Map Year to the X-axis. On the primary axis, plot a stacked waterfall of yearly changes (Green=Increase, Red=Decrease). On the secondary axis, plot a line of the total year-end population. Label data and analyze the trends.

**GPT-4.1:**

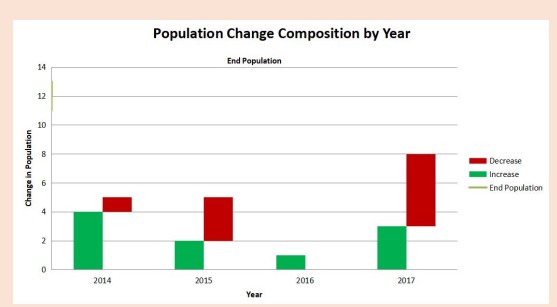

**Ground Truth:**

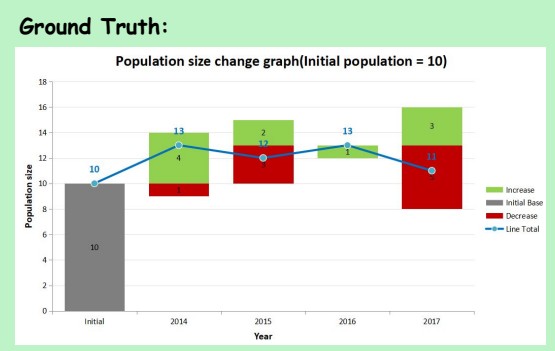

**Error Analysis:**

★ **Missing Baseline Data:** The critical "Initial Base" population of 10 is completely omitted from the X-axis in the first chart. Without this starting anchor, the subsequent changes float without context, making it impossible to determine actual population size.
★ **Logic Error in Aggregation:** The chart incorrectly sums the absolute values of "Increase" and "Decrease" instead of calculating the net change.
★ **Incorrect Calculated Derived Values:** The visual representation fails to reflect the "Line Total" or running balance of the population. The first chart completely lacks this derived series, showing only isolated components of change without their cumulative impact.
★ **Contradictory Directionality:** "Decrease" implies a downward movement or negative value. Plotting it as an upward extension of the bar creates a cognitive dissonance where "loss" is visualized as "gain" in height.
★ **Absence of Trend Indicators:** The first chart lacks the specific blue line markers that serve as the primary conclusion of the analysis (the actual population size at the end of each year).

**Percentage Of Score:** 46.15%

*Figure 22.* A sample bad case of **Structure** and **Data Accuracy** Errors in the DVSheet-Crea.

## DV-Sheet:Create-Bad Case

**Error Category:**
★ Data Accuracy Errors
★ Professional Format and Compliance

**Query:**
Please create a clear and easy-to-read dual-axis ranking comparison chart for the top 30 countries in terms of the total number of medals (`total`): The X-axis should display the countries sorted by `total` from high to low (only the top 30), with the main (left) axis using a bar chart to represent `total`, and the secondary (right) axis using a line chart to represent `gdp`.

**Gemini-2.5-pro:**

**Ground Truth:**

**Error Analysis:**

★ **Wrong fields plotted:** Chart shows Silver/Bronze counts, but labels/requirements are Total Medals and GDP.

★ **Values don't match data:** Y-axis peaks ~45, but USA Total Medals should be 120+ → likely plotting a medal subset (e.g., silver) as total.

★ **Dual-axis setup is conceptually incorrect:** A dual-axis chart is fine in principle, but the variables assigned are wrong—both lines/bars are medal counts with different labels, instead of two different measures (Medals on one axis, GDP on the other).

★ **Not sorted as required:** X-axis isn't in descending order by Total Medals.

★ **GDP axis scale/format is wrong:** GDP is shown with a "billion" unit but only single/double-digit tick values, and the range is too small. The axis should span a much larger range and display values that reflect real GDP gaps across countries.

**Percentage Of Score**: 38.46%

*Figure 23.* A sample bad case of **Data Accuracy** Errors in the DVSheet-Crea.

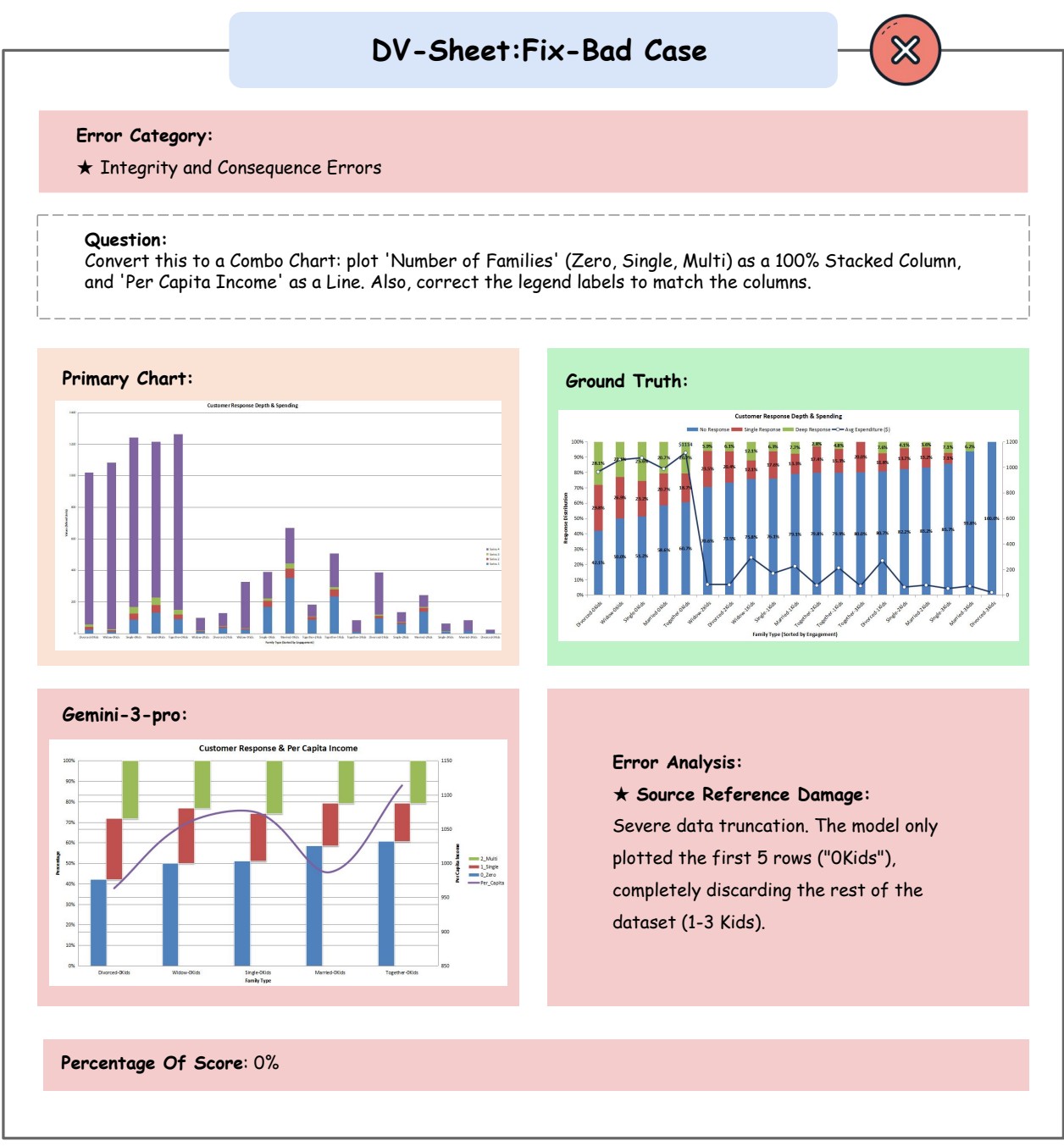

*Figure 24.* A sample bad case of **Integrity and Consequence** Errors in the DVSheet-Fix.

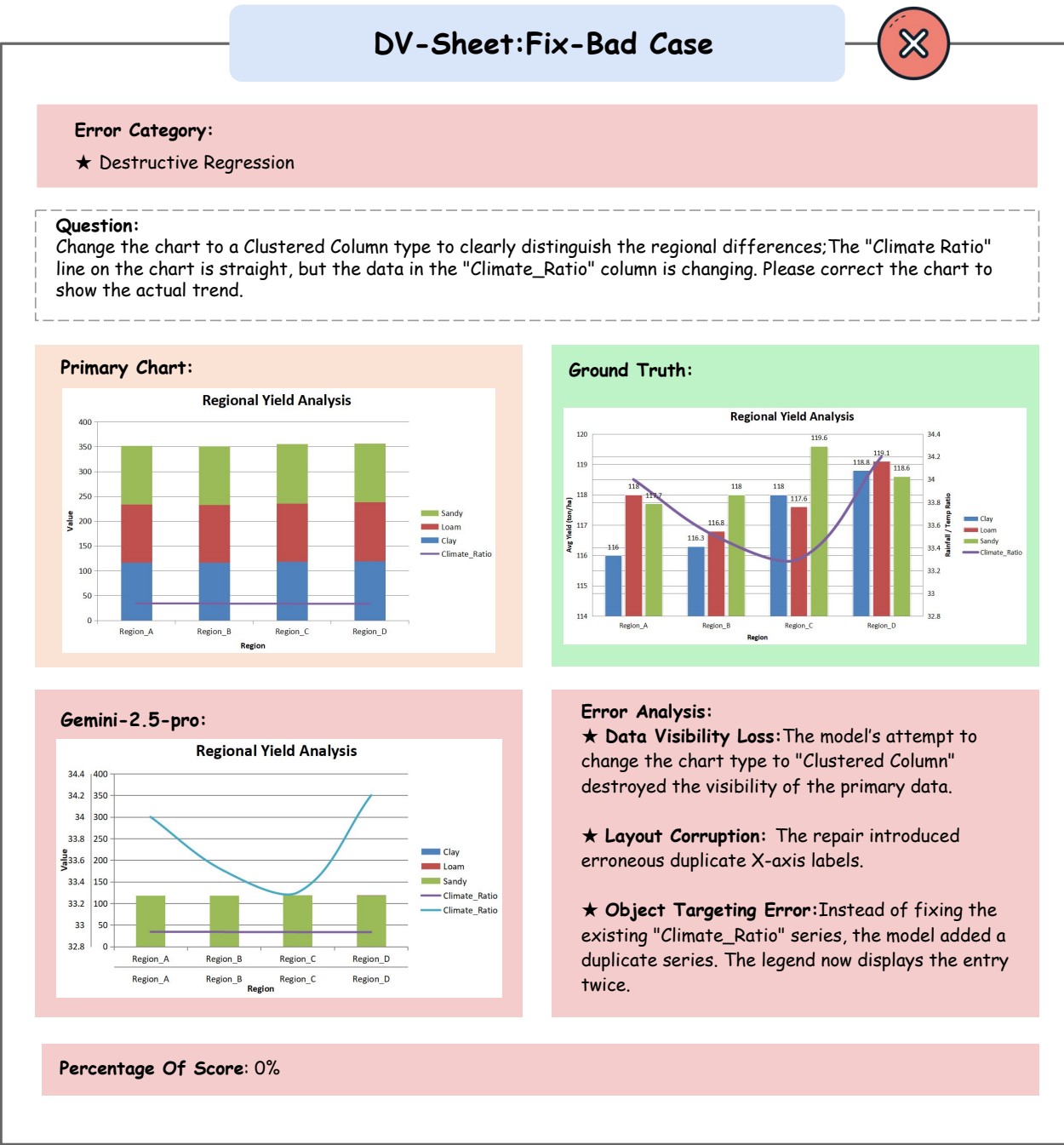

*Figure 25.* A sample bad case of **Destructive Regression** Errors in the DVSheet-Fix.

## DV-Sheet:Fix-Bad Case

**Error Category:**

★ Critical Blindness

**Question:**
This radar chart has visual overlaps and occlusions, and some dimensions are not fully displayed. Please make the modifications to facilitate my comparison of the source structure characteristics of candidates among different professional branches.

**Primary Chart:**

**Ground Truth:**

**GPT-5.2:**

**Error Analysis:**

★ **Failure to Identify Scale Variance:**

The model ignored vast unit disparities (e.g., thousands vs. decimals) and failed to normalize the data to a 0-100 scale. This omission caused incompatible scales to compress the lines into an indistinguishable bunch, rendering the repair useless.

**Percentage Of Score**: 0%

*Figure 26.* A sample bad case of **Critical Blindness** Errors in the DVSheet-Fix.

## DV-Sheet:Dashboards-Bad Case

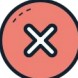

**Error Category:**
★ Business Insights and Problem Solving

**Query:**

Currently, there's a lack of intuitive comparison for the performance of pizza categories and daily order fluctuations. Could you create a dashboard to conduct a benchmark analysis across four dimensions—category sales benchmarks, daily order trends, size preferences, and top-selling items—to optimize product strategies and operational cadence?

**Gemini-3-pro:**

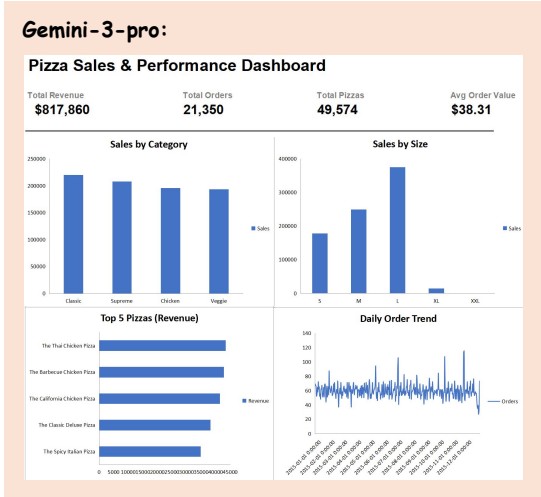

**Ground Truth:**

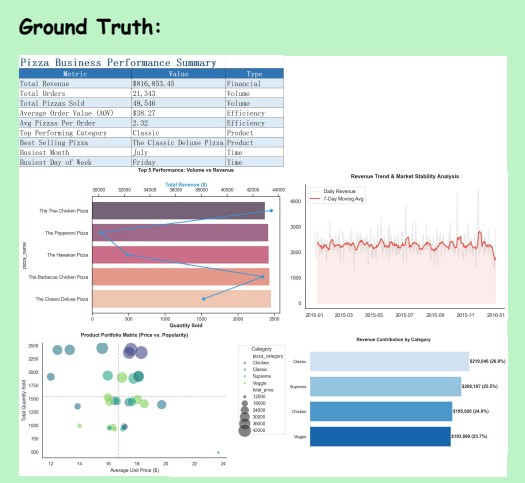

**Error Analysis:**

★ **Lack of KPI Elements:** Missing specific callouts for "Peak Activity Periods" (July/Friday) and "Price Elasticity" analysis.
★ **Incorrect Aggregation:** Categorical charts provide raw volume but fail to show the percentage contribution or benchmark comparisons.
★ **KPI Data Mismatch:** Summary values are incorrect (e.g., Total Revenue $817,860 vs. standard $816,853.45).
★ **Incorrect Chart Selection:** Uses basic bars instead of a Product Portfolio Matrix(Scatter/Bubble), losing the price vs. popularity correlation.
★ **Readability Issues:** "Daily Order Trend" is compressed and cluttered with redundant "0:00:00" timestamps, obscuring date intervals.

**Percentage Of Score:** 25%

*Figure 27.* A sample bad case of **Business Insights and Problem Solving** Errors in the DVSheet-Dash.

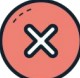

# DV-Sheet:Dashboards-Bad Case

**Error Category:**
★ Data Logical Consistency
★ Visualization Design

**Query:**
I've noticed that the property prices in New York vary greatly. This table is not very intuitive. Could you please create a dashboard for me to analyze the fluctuations in property prices? This would help me compare the market performance of different property types in various districts.

**Gemini-3-pro:**

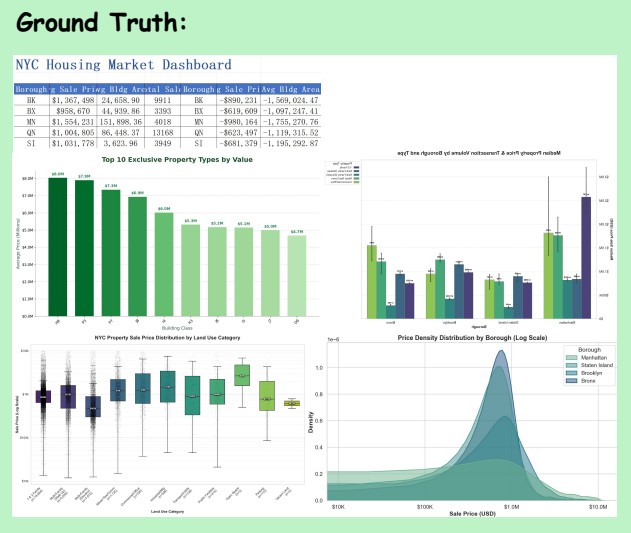

**Ground Truth:**

**Error Analysis:**

★ **Incorrect Data Aggregation:** The "Price Fluctuation by Building Age" chart uses averages which are highly sensitive to outliers in real estate. The standard dashboard correctly prioritizes medians to represent the "typical" price.

★ **Unfiltered Data Noise:** The first picture likely includes low-value non-market transfers (e.g., $0 or $1 sales), whereas the standard dashboard applies a Price Validity Filter (>$1,000) to ensure market relevance.

★ **Missing Multidimensional Analysis:** The charts lack secondary encodings (like color-coded legends for property types within borough charts) required to answer complex market questions at a glance.

★ **Vague Categorization:** The "Avg Price by Property Type" uses cryptic codes like B9, B3, A1 without providing the full descriptive labels (e.g., "Walk-up" or "Elevator Apartments") found in the standard dashboard, reducing readability for non-experts.

**Percentage Of Score**: 21.43%

*Figure 28.* A sample bad case of **Data Logical Consistency** and **Visualization Design** Errors in the DVSheet-Dash.

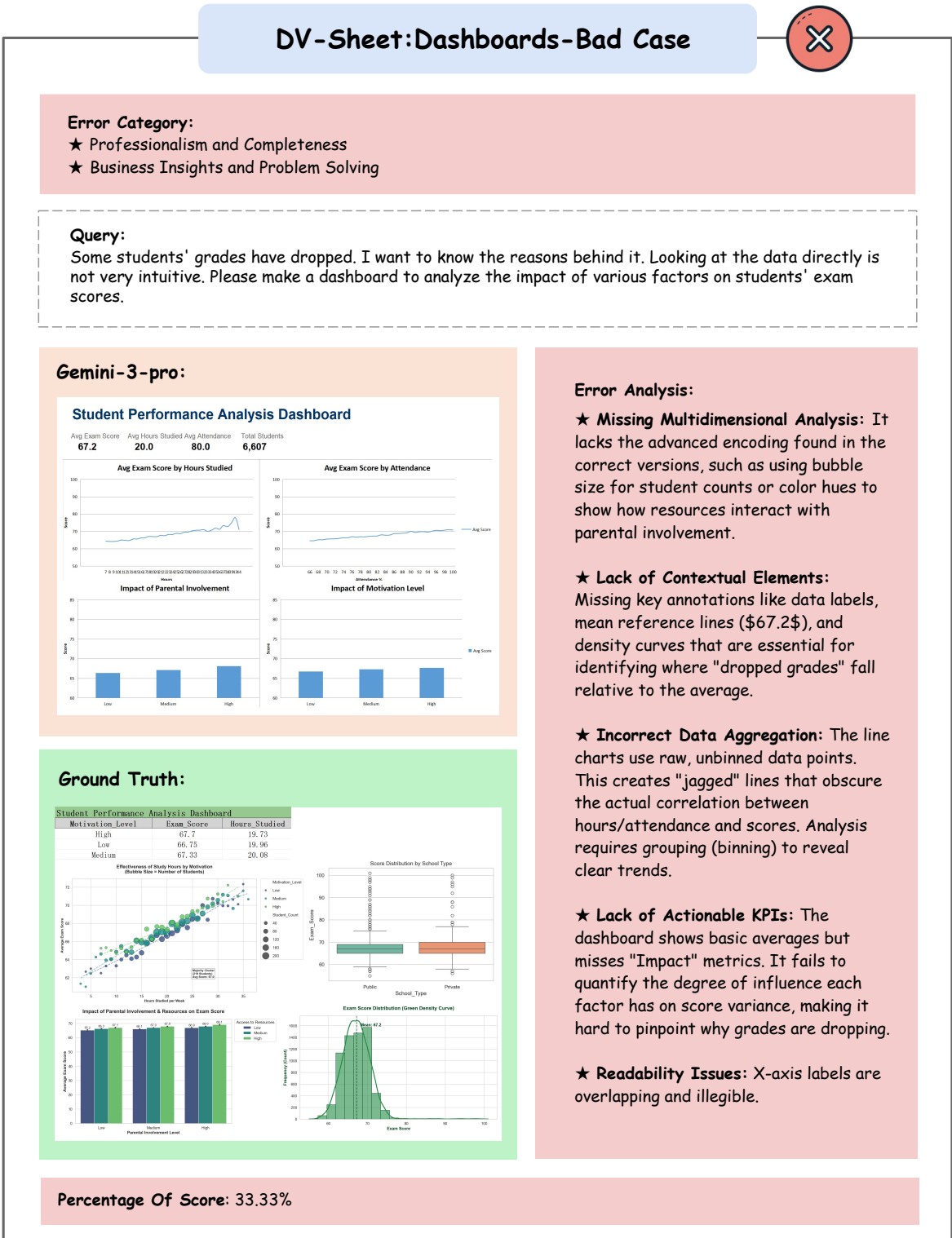

*Figure 29.* A sample bad case of **Professionalism and Completeness** and **Business Insights and Problem Solving** Errors in the DVSheet-Dash.

*Table 24.* Fine-grained Error Analysis for the DVSHEET-DASH task. Values represent error occurrence rates per category. The highest error rate for each model is highlighted in orange , and the lowest in blue . *Abbreviations*: **WCT**: Wrong Chart Type, **MVE**: Missing Visual Elements, **LI**: Layout Issues, **RI**: Rendering Issues, **MUS**: Missing Unit Symbols, **MDA**: Missing Dimension Analysis, **NFE**: Number Format Error, **MCE**: Missing Context Elements, **CE**: Calculation Errors, **FRI**: Filter Range Issues, **CCM**: Cross-Check Missing, **KDM**: KPI Data Mismatch, **MKE**: Missing KPI Elements, **ALE**: Aggregation Level Error.

| Method | Visual Design (VD) | | | | Rigor & Completeness (RC) | | | | Data Consistency (DC) | | | | Insight & Res. (IR) | |
|---|---|---|---|---|---|---|---|---|---|---|---|---|---|---|
| | WCT | MVE | LI | RI | MUS | MDA | NFE | MCE | CE | FRI | CCM | KDM | MKE | ALE |
| DeepSeek-V3.2 | **24.07%** | 9.88% | 7.41% | 0.62% | 12.35% | 10.49% | 4.32% | 2.47% | 8.02% | 4.32% | 3.09% | 1.23% | 11.11% | **0.62%** |
| Gemini-2.5-Pro | **35.98%** | 3.05% | 14.02% | 6.10% | 2.44% | 1.83% | 9.15% | 7.32% | 4.27% | 0.61% | 6.10% | 4.27% | 4.27% | 1.22% |
| Gemini-3-Pro | **33.33%** | 2.22% | 18.52% | 2.96% | 1.48% | **0.74%** | 10.37% | 8.15% | 1.48% | 0.74% | 7.41% | 6.67% | 5.19% | 0.74% |
| GLM-4.7 | **40.34%** | 0.57% | 6.25% | 11.36% | 2.84% | 1.70% | 10.80% | 6.25% | 3.41% | 0.00% | 1.70% | 7.95% | 0.57% | 6.25% |
| GPT-4.1 | 25.95% | 0.54% | 3.78% | 9.19% | 4.86% | 2.70% | 9.19% | **18.38%** | 12.98% | 1.62% | 4.32% | 2.70% | 0.54% | 3.24% |
| GPT-5.2 | 6.90% | 0.86% | 0.86% | 9.49% | 6.03% | 12.07% | 13.79% | 10.34% | **14.66%** | 3.45% | 4.31% | 9.48% | 4.31% | 3.45% |

*Table 25.* Dimensional Summary of Error Rates for DVSHEET-DASH. Per-model maximums are highlighted in orange and minimums in blue . *Abbreviations*: **VD**: Visual Design, **RC**: Rigor & Completeness, **DC**: Data Consistency, **IR**: Insight & Resolution.

| Method | VD | RC | DC | IR | Total |
|---|---|---|---|---|---|
| DeepSeek-V3.2 | **41.98%** | 29.63% | 16.66% | **11.73%** | 100% |
| Gemini-2.5-Pro | **59.15%** | 20.74% | 15.25% | **5.49%** | 100% |
| Gemini-3-Pro | **57.03%** | 20.74% | 16.30% | **5.93%** | 100% |
| GLM-4.7 | **58.52%** | 21.59% | 13.06% | **6.82%** | 100% |
| GPT-4.1 | **39.46%** | 35.13% | 21.62% | **3.78%** | 100% |
| GPT-5.2 | **18.11%** | 42.23% | 31.90% | 7.76% | 100% |
| **Avg. Error** | **45.71%** | 28.34% | 19.13% | **6.92%** | 100% |

*Consistency* errors peak in **ECharts** for several agents (e.g., 55.15% for `Gemini-3-Flash` and 47.63% for `GPT-5.1`). In contrast, *Visual Style* errors show lower averages and larger variance across languages, indicating that stylistic compliance is comparatively less systematic and more model- and backend-specific.

**Cross-backend visual mapping failures.** A recurring failure pattern across the ten cases is visual mapping breakdown, where agents preserve the high-level chart intent but fail to realize key marks, layers, or structural constraints required by the target backend. This is most evident in the ridgeline task across multiple libraries: outputs often capture only partial shape trends yet violate stacking order, spacing, proportionality, and axis alignment, resulting in overlapped ridges and misaligned legends/titles (Figs. 31,33,35,37). These qualitative failures align with the table-level finding that *Layout & Readability* dominates overall errors, since a small number of layout mismatches (e.g., ridge overlap, incorrect aspect ratio, mispositioned legend/title) can render an otherwise plausible distribution plot difficult to interpret.

**Transformation and scale inconsistencies as the core data-consistency bottleneck.** A second prominent error mode is incorrect transformations and scale choices, which directly produce *Data Consistency* violations even when the chart type is nominally correct. Typical examples include using the wrong normalization domain on the secondary axis (e.g., plotting normalized 0–100% scores instead of the required raw median scale and altering the country set/order), or applying an unintended log transform that shifts the entire axis range into an incompatible regime (Figs. 32 and 34). Similar scale/encoding failures appear when trendlines or reference thresholds are truncated or stop mid-range rather than spanning the full domain required by the specification, which breaks both semantic correctness and interpretability (Figs. 30 and 38). These cases provide concrete explanations for why *Data Consistency* remains substantial in Table 27 despite style being the smallest dimension.

**Backend-specific rendering omissions and chart-type degeneration.** Beyond mapping and scaling, several cases show backend-specific rendering omissions where essential graphical components are missing, causing the visualization to degenerate into an unintended form. For example, a correlation heatmap can collapse into a diagonal line plot with missing matrix cells, missing colormap encoding, and absent colorbar/annotations, indicating that the agent failed to instantiate the

correct mark/trace structure under Plotly (Fig. 36). In ECharts, composite statistical graphics may lose entire distribution layers (e.g., violin not rendered, leaving only boxplots), accompanied by legend-style mismatches and axis range drift that introduces unnecessary whitespace (Fig. 30). These failures suggest that correctness in DV-Evol is not only about choosing the right chart, but about reliably realizing multi-layer specifications and backend-specific primitives.

**Model-dependent error tendencies across five backends.** Comparing the two-model, five-backend case set reveals complementary weaknesses. GPT-5.2 tends to be structure-first but incomplete in realization: its outputs often resemble the target layout yet omit crucial layers or truncate global references (e.g., missing violin layer or truncated percentile/trend lines in ECharts; Fig. 30, and truncated trendline/insufficient points in Vega-Lite; Fig. 38). In contrast, Gemini-3-Pro more frequently exhibits pattern-preserving but layout-unstable behavior in ridgeline-style tasks, where the overall distribution trends appear plausible but labels, spacing, and proportionality drift substantially from the intended design, reducing readability and professional quality (Figs. 31,33,35,37). Together, these cases illustrate how the same DV-Evol intent can fail through different mechanisms depending on both the agent and the backend.

*Table 26.* Fine-grained Error Analysis for the DV-EVOL task. Values represent error rates per library. The highest error rate in each column is highlighted in orange , and the lowest in green . *Abbreviations*: **Py**: Python (Matplotlib/Seaborn), **EC**: Apache ECharts, **VL**: Vega-Lite, **D3**: D3.js, **PL**: Plotly.js.

| Agent | Layout & Readability Issues | | | | | Visual Style Issues | | | | | Data Consistency Issues | | | | |
|---|---|---|---|---|---|---|---|---|---|---|---|---|---|---|---|
| | Py | EC | VL | D3 | PL | Py | EC | VL | D3 | PL | Py | EC | VL | D3 | PL |
| Gemini-2.5-Pro | **34.68%** | **47.15%** | 46.09% | 45.25% | 38.77% | **34.32%** | 27.73% | **20.59%** | 26.40% | 27.47% | 31.00% | **27.12%** | **35.32%** | 30.35% | 27.75% |
| Gemini-3-Flash | 46.61% | **33.28%** | 51.80% | 42.92% | 42.92% | 19.80% | **9.67%** | 22.00% | **40.96%** | 30.61% | 33.44% | **55.15%** | 26.20% | **20.14%** | 24.47% |
| Gemini-3-Pro | 58.70% | 36.65% | 45.17% | **30.00%** | 44.87% | 18.51% | **17.15%** | 20.66% | 24.92% | **28.99%** | 21.89% | **46.20%** | 34.17% | 45.93% | 26.14% |
| GPT-5.1 | 42.68% | 41.39% | **40.71%** | 46.75% | **47.84%** | 21.79% | **9.98%** | **31.29%** | 28.85% | 20.14% | 34.53% | **47.63%** | 29.00% | **26.40%** | 31.02% |
| GPT-5.2 | 38.14% | 38.60% | **48.79%** | **35.27%** | 35.77% | 34.02% | 33.38% | **24.20%** | 30.14% | **36.11%** | 27.84% | 28.02% | **26.81%** | 34.59% | 28.32% |

*Table 27.* Dimensional Summary of Error Rates for DV-EVOL task. Per-model maximums are highlighted in orange and minimums in green . *Abbreviations*: **LR**: Layout & Readability, **VS**: Visual Style, **DC**: Data Consistency.

| Agent | LR | VS | DC | Total |
|---|---|---|---|---|
| Gemini 2.5 Pro | **42.39%** | **27.30%** | 30.31% | 100.00% |
| Gemini 3 Flash | **43.51%** | **24.61%** | 31.88% | 100.00% |
| Gemini 3 Pro | **43.08%** | **22.05%** | 34.87% | 100.00% |
| GPT-5.1 | **43.87%** | **22.41%** | 33.72% | 100.00% |
| GPT-5.2 | **39.31%** | 31.57% | **29.12%** | 100.00% |
| **Avg** | **42.43%** | **25.59%** | 31.98% | — |

## H.3. DV-Inter Error Analysis

**Fine-grained error analysis for DV-Inter.** Table 28 and Table 29 show that DV-Inter failures are primarily driven by the Cognitive-Execution Gap (CEG) (**38.44%** on average), where Intent Drift (ID) is especially severe for frontier models (e.g., Gemini-2.5-Pro: **45.38%**, GPT-4.1: **41.74%**), indicating that multi-turn constraint retention remains a key bottleneck despite strong code generation. Secondary error profiles differ by model family: GLM-4.7 and Qwen3-235B-A22B lean toward Interactive Avoidance (IA) with high Speculative Logic (SL) (33.32% and 30.11%), suggesting assumption-first completion instead of clarification; GPT-5.2 concentrates on Heuristic Logic Approximation (HL) (24.74%), reflecting simplified but brittle reasoning; and architecture/scale effects are visible at both ends, with DeepSeek-V3.2 peaking in Visual Design Mismatch (MM: 13.61%, LC: 16.00%) and the small Qwen3-8B exhibiting pronounced Technical Collapse via Debugging Deadlock (DD) (33.40%), underscoring that interactive refinement stresses both long-horizon intent tracking and tool-facing error recovery.

**Interactive avoidance through assumption-first execution.** A recurring failure pattern in DV-Inter is Interactive Avoidance (IA), where agents proceed with execution despite unresolved ambiguity instead of seeking clarification. As illustrated in Fig. 40, when asked to create a pie chart over a high-cardinality entity set, the agent directly visualizes

## DV-Evol Bad Case——Apache Echarts

**Error Categooy:**
- Visual Style Issues
- Layout & Readability Issues

**Question:**
- Sort countries by descending MEDIAN citations per paper.
- Plot violins with box overlays for the same country set (order only changes).
- Add a red dashed global 75th-percentile line with value in legend.
- Add a global linear trendline across the sorted countries with legend label "Trendline".
- Keep title/axis labels; y-axis must cover full data range.

**GPT-5.2:**

**Ground Truth:**

**Error Analysis:**
- Missing visual layer: only boxplots are shown; the violin distribution layer is not rendered, so distribution shapes are lost.
- Incomplete/abnormal data display: some countries (e.g., Singapore, Hong Kong) collapse into a near-flat line/tiny box, suggesting missing samples or over-aggregation.
- Truncated 75th-percentile line: the red dashed reference line does not span the full width of all countries.
- Truncated trendline: the black trendline stops around the mid-x range (near Taiwan) instead of covering the full x-axis.
- Legend mismatch: legend uses point markers (green/orange) and does not clearly match the "red dashed + black solid" line styles seen in the reference.
- Axis range/whitespace issue: the y-axis extends into negatives (e.g., -30), creating excessive empty space; the reference is tighter and starts near 0.
- Overall styling drift: country color mapping, gridline density/orientation, and typography differ from the reference.

**Percentage Of Score: 47.5%**

*Figure 30.* **Detailed Error Analysis of an Apache Echarts Failure Case (Score: 47.5%).**

## DV-Evol Bad Case echarts：Visual mapping failure 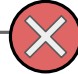

**Question: Generate a ridgeline plot comparing life expectancy (1990–2014) across income groups. Use fixed 5-year bins from 40 to 85, normalize to percent frequency, and stack the ridges from "Low income" to "High income" using designated HEX colors.**

**Gemini-3-pro-preview:**

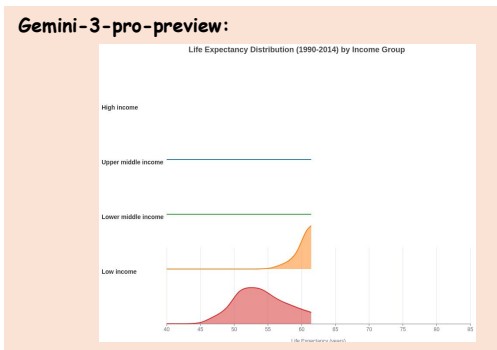

**Ground Truth:**

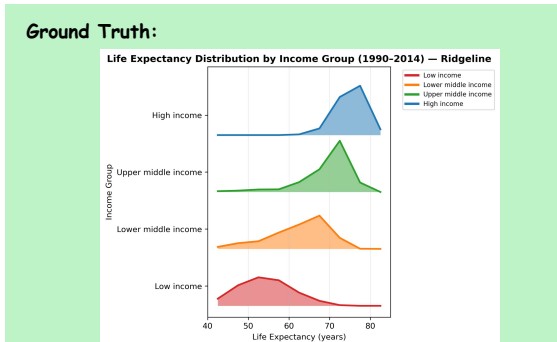

**Plot code:**
```
const colors = ["#1f77b4", "#2ca02c", "#ff7f0e", "#d62728"];
// High to Low
const groups = ["High income", "Upper middle income", "Lower middle income", "Low income"];

option = {
    title: { text: 'Life Expectancy Distribution (1990-2014)', left:
'center' },
    grid: groups.map((_, i) => ({
        left: '20%', right: '10%', top: `${15 + i * 18}%`, height:
'25%'
    })),
    xAxis: groups.map((_, i) => ({
        type: 'value', min: 40, max: 85, gridIndex: i, show: i === 3,
        name: i === 3 ? 'Years' : ''
    })),
    yAxis: groups.map((_, i) => ({
        type: 'value', gridIndex: i, show: false, max: 80
    })),
    series: groups.map((group, i) => ({
        name: group, type: 'line', smooth: true, xAxisIndex: i,
yAxisIndex: i,
        symbol: 'none',
        areaStyle: { color: colors[i], opacity: 0.5 },
        lineStyle: { color: colors[i], width: 2 },
        data: [[40, 0], ...results.filter(r => r.group === group).map(r
=> [r.x, r.y]), [85, 0]]
    }))
};
```

**Error Analysis:** 1. Data Integrity
❌1.1 Trend/Pattern: 0/1 The distribution curves successfully mirror the multi-modal trends of the Ground Truth.
❌1.2 Completeness: 0/1 Critical data series or specific years are missing from the visualization.
❌1.3 Label Accuracy: 0/1 Axis tick values and labels do not precisely match the original data points.
❌1.4 Data Consistency: 0/1 Proportional relationships between income groups remain accurate across the X-axis.
2. Style Imitation
✅2.1 Color Palette: 1/1 The model correctly applied the designated HEX codes for all income categories.
❌2.2 Chart Elements: 0/1 Specific ridgeline outlines and gridline styles were not properly reproduced.
✅2.3 Background: 1/1 The background color and light/dark mode are consistent with the Ground Truth.
❌2.4 Font and Text Styles: 0/1 Font weight and sizing for titles and labels deviate from the original style.
❌2.5 Element Spacing: 0/1 Distances between labels and the plot area are uneven, creating a cluttered look.
❌2.6 Borders and Shadows: 0/1 Visual effects like drop shadows or specific border strokes were omitted.

3. Layout & Aesthetics
❌3.1 Labels: 0/1 Legend and title positioning are misaligned compared to the standard layout.
✅3.2 Readability: 1/1 Excessive ridge overlap and small font sizes compromise the overall legibility.
❌3.3 Axis Alignment and Spacing: 0/1 The axes are poorly calibrated, causing data to shift away from intended baselines.
❌3.4 Element Alignment: 0/1 Graphical components fail to align precisely with their respective axis ticks.
❌3.5 Proportionality and Aspect Ratio: 0/1 An incorrect aspect ratio has led to vertical distortion of the ridge shapes.
❌3.6 Overall Balance: 0/1 The composition lacks visual equilibrium, making the chart difficult to interpret quickly.

**Score: 3/16**

*Figure 31.* A sample bad case of **Visual mapping failure** in DV-Evol using **ECharts**.

*Figure 32.* A sample bad case in DV-Evol using **Python**.

## DV-Evol Bad Case python：Visual mapping failure ❌

**Question: Generate a ridgeline plot comparing life expectancy (1990–2014) across income groups. Use fixed 5-year bins from 40 to 85, normalize to percent frequency, and stack the ridges from "Low income" to "High income" using designated HEX colors.**

**Gemini-3-pro-preview:**

**Ground Truth:**

**Plot code:**

```python
df_plot = pd.read_csv('result.csv')
colors = {"Low income": "#d62728", "Lower middle income":
"#ff7f0e",
        "Upper middle income": "#2ca02c", "High income":
"#1f77b4"}
bin_centers = np.linspace(42.5, 82.5, 9)
x_coords = np.concatenate(([37.5], bin_centers, [87.5]))

fig, ax = plt.subplots(figsize=(10, 6))
for i, group in enumerate(groups):
    base_y = i * 1.0  # Vertical offset
    pcts = df_plot[df_plot['Income Group'] ==
group]['Percentage'].values
    y_coords = base_y + np.concatenate(([0], pcts * 0.025, [0]))
# Scaled heights

    # Render ridge
    ax.fill_between(x_coords, base_y, y_coords,
color=colors[group], alpha=0.5, zorder=10-i)
    ax.plot(x_coords, y_coords, color=colors[group], lw=2,
zorder=10-i)

# Formatting
ax.set_yticks([i * 1.0 for i in range(len(groups))])
ax.set_yticklabels(groups)
ax.set_xlabel("Life Expectancy (years)")
ax.set_title("Life Expectancy Distribution (1990-2014)")
ax.spines[['top', 'right', 'left']].set_visible(False)
plt.savefig('result.png', dpi=150)
```

**Error Analysis:** 1. Data Integrity
✅1.1 Trend/Pattern: 1/1  The distribution curves successfully mirror the multi-modal trends of the Ground Truth.
❌1.2 Completeness: 0/1 Critical data series or specific years are missing from the visualization.
✅1.3 Label Accuracy: 1/1 Axis tick values and labels do not precisely match the original data points.
❌1.4 Data Consistency: 0/1 Proportional relationships between income groups remain accurate across the X-axis.
2. Style Imitation
✅2.1 Color Palette: 1/1 The model correctly applied the designated HEX codes for all income categories.
❌2.2 Chart Elements: 0/1 Specific ridgeline outlines and gridline styles were not properly reproduced.
✅2.3 Background: 1/1 The background color and light/dark mode are consistent with the Ground Truth.
❌2.4 Font and Text Styles: 0/1 Font weight and sizing for titles and labels deviate from the original style.
✅2.5 Element Spacing: 1/1 Distances between labels and the plot area are uneven, creating a cluttered look.
❌2.6 Borders and Shadows: 0/1 Visual effects like drop shadows or specific border strokes were omitted.

3. Layout & Aesthetics
❌3.1 Labels: 0/1 Legend and title positioning are misaligned compared to the standard layout.
✅3.2 Readability: 1/1 Excessive ridge overlap and small font sizes compromise the overall legibility.
❌3.3 Axis Alignment and Spacing: 0/1  The axes are poorly calibrated, causing data to shift away from intended baselines.
❌3.4 Element Alignment: 0/1  Graphical components fail to align precisely with their respective axis ticks.
❌3.5 Proportionality and Aspect Ratio: 0/1 An incorrect aspect ratio has led to vertical distortion of the ridge shapes.
❌3.6 Overall Balance: 0/1 The composition lacks visual equilibrium, making the chart difficult to interpret quickly.

**Score: 6/16**

*Figure 33.* A sample bad case of **Visual mapping failure** in DV-Evol using **Python**.

## DV-Evol Bad Case——d3.js 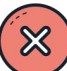

Erryor Category: Layout & Readability Issues

Question:
- Use normalized rider percentages instead of log-transformed counts.
- Rename: Rider_On → Active Riders, Rider_Off → Inactive Riders.
- Add vertical thresholds: 90% cumulative for Active, 10% cumulative for Inactive.
- Title: "Ridgeline Plot of Normalized Rider Counts with Threshold Lines".
- KDE smoothing: bw_adjust = 1.
- Keep ridgeline style, x-axis label "Log of Number of Riders", and stack order: Inactive above Active.

GPT-5.2:

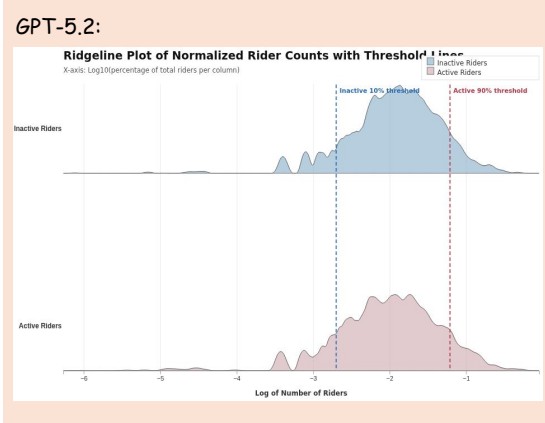

Ground Truth:

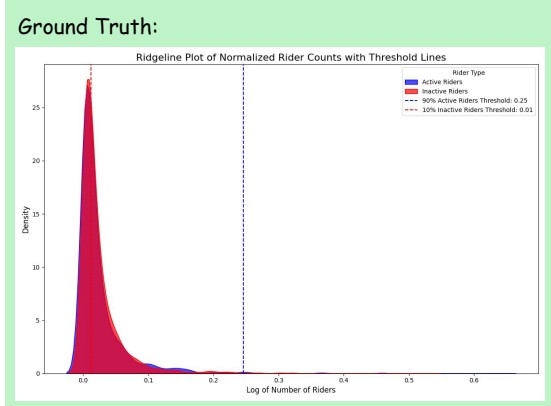

Error Analysis:
- Incorrect X-axis transform: using Log10(percentage) shifts the range to about -6 to -1, while the reference stays around 0 to 0.7.
- Layout mismatch: the reference overlays both KDE curves in a single axis, but the output uses two separate ridgeline panels (stacked vertically).
- Threshold lines incorrect: positions fall in the negative range and the Active 90% / Inactive 10% mapping (colors/association) does not match the reference.
- Color scheme drift: the reference uses blue/red with strong overlap, but the output uses light blue/pink.
- Extra/missing text elements: an additional "X-axis: Log10(...)" note appears and the overall axis/legend presentation differs from the reference.

Percentage Of Score: 12.5%

*Figure 34.* A sample bad case in DV-Evol using **D3.js**.

## DV-Evol Bad Case d3.js: Visual mapping failure ❌

**Question: Generate a ridgeline plot comparing life expectancy (1990–2014) across income groups. Use fixed 5-year bins from 40 to 85, normalize to percent frequency, and stack the ridges from "Low income" to "High income" using designated HEX colors.**

**Gemini-3-pro-preview:**

**Ground Truth:**

**Plot code:**

```
const x = d3.scaleLinear()
    .domain([35, 90])
    .range([margin.left, width - margin.right]);

const y = d3.scalePoint()
    .domain(["Low income", "Lower middle income", "Upper middle
income", "High income"])
    .range([height - margin.bottom, margin.top]);

const yHeight = d3.scaleLinear()
    .domain([0, maxPercent])
    .range([0, spacing * 1.5]);

const area = d3.area()
    .curve(d3.curveMonotoneX)
    .x(d => x(d.Bin_Mid))
    .y0(0)
    .y1(d => -yHeight(d.Percent));

categories.reverse().forEach(group => {
    const g = svg.append('g')
        .attr('transform', `translate(0, ${y(group)})` );
    g.append('path')
        .datum(groupedData[group])
        .attr('fill', colors[group])
        .attr('fill-opacity', 0.6)
        .attr('stroke', colors[group])
        .attr('stroke-width', 2)
        .attr('d', area);
});
```

**Error Analysis:** 1. Data Integrity
✅1.1 Trend/Pattern: 1/1  The distribution curves successfully mirror the multi-modal trends of the Ground Truth.
❌1.2 Completeness: 0/1 Critical data series or specific years are missing from the visualization.
❌1.3 Label Accuracy: 0/1 Axis tick values and labels do not precisely match the original data points.
✅1.4 Data Consistency: 1/1 Proportional relationships between income groups remain accurate across the X-axis.
2. Style Imitation
✅2.1 Color Palette: 1/1 The model correctly applied the designated HEX codes for all income categories.
❌2.2 Chart Elements: 0/1 Specific ridgeline outlines and gridline styles were not properly reproduced.
✅2.3 Background: 1/1 The background color and light/dark mode are consistent with the Ground Truth.
❌2.4 Font and Text Styles: 0/1 Font weight and sizing for titles and labels deviate from the original style.
❌2.5 Element Spacing: 0/1 Distances between labels and the plot area are uneven, creating a cluttered look.
❌2.6 Borders and Shadows: 0/1 Visual effects like drop shadows or specific border strokes were omitted.

3. Layout & Aesthetics
❌3.1 Labels: 0/1 Legend and title positioning are misaligned compared to the standard layout.
❌3.2 Readability: 0/1 Excessive ridge overlap and small font sizes compromise the overall legibility.
❌3.3 Axis Alignment and Spacing: 0/1  The axes are poorly calibrated, causing data to shift away from intended baselines.
❌3.4 Element Alignment: 0/1  Graphical components fail to align precisely with their respective axis ticks.
❌3.5 Proportionality and Aspect Ratio: 0/1 An incorrect aspect ratio has led to vertical distortion of the ridge shapes.
❌3.6 Overall Balance: 0/1 The composition lacks visual equilibrium, making the chart difficult to interpret quickly.

**Score: 4/16**

*Figure 35.* A sample bad case of **Visual mapping failure** in DV-Evol using **D3.js**.

## DV-Evol Bad Case——plotly.js

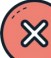

Error Category:
- Data Consistency Issues
- Layout & Readability Issues

Question:
- Z-score the 9 sensors (μ, σ from full data), keep rows with any |z|>1.0, then group by hour and take mean.
- Compute Pearson correlations (9 sensors + quality), sort sensors by |corr(sensor, quality)| desc, keep quality last.
- Heatmap style: title exactly "Refined Outlier Correlation: Sorted by Impact", annotate + colorbar to 4 decimals, cmap='RdBu_r', vmin=-1/vmax=1, grid=#4682B4 (lw=1.2), output 10×10.

GPT-5.2:

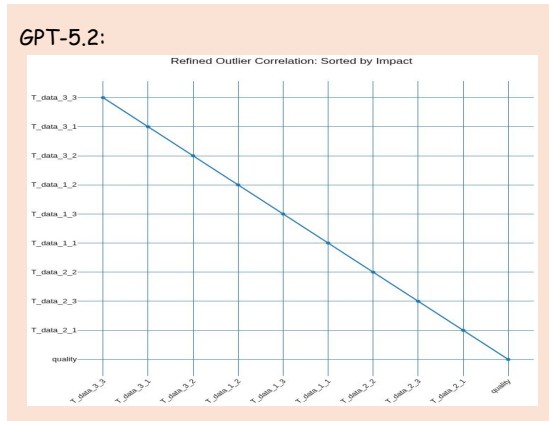

Ground Truth:

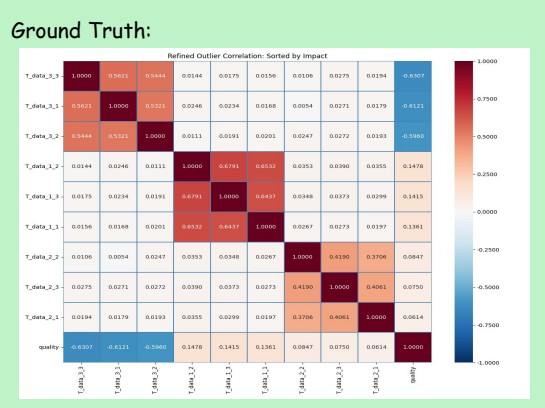

Error Analysis:
- Wrong chart type: it became a diagonal line plot instead of a correlation heatmap.
- Missing core content: the 10×10 correlation matrix cells (colors + values) are not rendered, so correlation patterns are lost.
- Missing color encoding: no proper RdBu_r diverging colormap with vmin=-1 and vmax=1.
- Missing colorbar: the right-side correlation color scale (-1 to 1, 4-decimal ticks) is absent.
- Missing annotations: per-cell Pearson correlation values (4 decimals) are not shown.
- Wrong grid style: only axis background gridlines appear, not the heatmap cell borders (the #4682B4 boxed grid effect is not implemented).

Percentage Of Score: 0%

*Figure 36.* A sample bad case in DV-Evol using **Plotly.js**.

## DV-Evol Bad Case ploty.js：Visual mapping failure 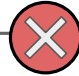

**Question: Generate a ridgeline plot comparing life expectancy (1990–2014) across income groups. Use fixed 5-year bins from 40 to 85, normalize to percent frequency, and stack the ridges from "Low income" to "High income" using designated HEX colors.**

**Gemini-3-pro-preview:**

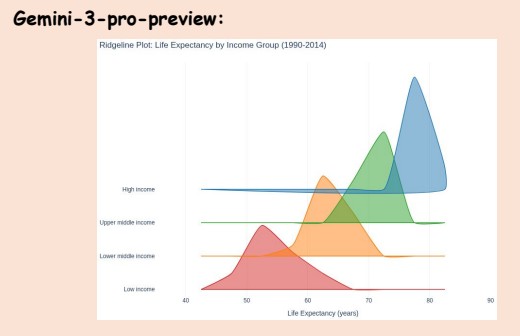

**Ground Truth:**

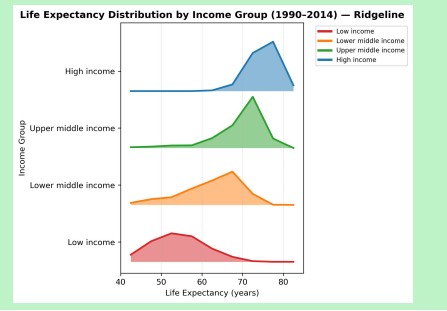

**Plot code:**
```
for g in groups:
    d = result_df[result_df['Group'] == g].sort_values('BinMid')
    off = y_offsets[g]
    px = [d['BinMid'].iloc[0]] + d['BinMid'].tolist() +
[d['BinMid'].iloc[-1]]
    py = [off] + (d['Percent'] + off).tolist() + [off]

    fig.add_trace(go.Scatter(
        x=px, y=py, fill='toself', mode='lines',
        line=dict(color=colors[g], width=1.5, shape='spline'),
        fillcolor=f"rgba{tuple(int(colors[g][i:i+2], 16) for i in (1, 3,
5)) + (0.5,)}"
    ))

fig.update_layout(
    xaxis_range=[35, 90],
    yaxis=dict(tickvals=list(y_offsets.values()), ticktext=groups,
range=[-5, 170]),
    plot_bgcolor="white"
)
```

**Error Analysis:** 1. Data Integrity
✅1.1 Trend/Pattern: 1/1 The distribution curves successfully mirror the multi-modal trends of the Ground Truth.
❌1.2 Completeness: 0/1 Critical data series or specific years are missing from the visualization.
❌1.3 Label Accuracy: 0/1 Axis tick values and labels do not precisely match the original data points.
❌1.4 Data Consistency: 0/1 Proportional relationships between income groups remain accurate across the X-axis.
2. Style Imitation
✅2.1 Color Palette: 1/1 The model correctly applied the designated HEX codes for all income categories.
❌2.2 Chart Elements: 0/1 Specific ridgeline outlines and gridline styles were not properly reproduced.
✅2.3 Background: 1/1 The background color and light/dark mode are consistent with the Ground Truth.
❌2.4 Font and Text Styles: 0/1 Font weight and sizing for titles and labels deviate from the original style.
❌2.5 Element Spacing: 0/1 Distances between labels and the plot area are uneven, creating a cluttered look.
❌2.6 Borders and Shadows: 0/1 Visual effects like drop shadows or specific border strokes were omitted.

3. Layout & Aesthetics
❌3.1 Labels: 0/1 Legend and title positioning are misaligned compared to the standard layout.
✅3.2 Readability: 1/1 Excessive ridge overlap and small font sizes compromise the overall legibility.
❌3.3 Axis Alignment and Spacing: 0/1 The axes are poorly calibrated, causing data to shift away from intended baselines.
❌3.4 Element Alignment: 0/1 Graphical components fail to align precisely with their respective axis ticks.
❌3.5 Proportionality and Aspect Ratio: 0/1 An incorrect aspect ratio has led to vertical distortion of the ridge shapes.
❌3.6 Overall Balance: 0/1 The composition lacks visual equilibrium, making the chart difficult to interpret quickly.

**Score: 4/16**

*Figure 37.* A sample bad case of **Visual mapping failure** in DV-Evol using **Ploty.js**.

## DV-Evol Bad Case——vega-lite

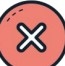

Error Category:
- Data Consistency Issues
- Layout & Readability Issues

Question:
- Bin Rank into 4 tiers: [0–20], (20–40], (40–70], (70–100] (each tier ≥ 5 schools)
- Aggregate Overall Score using MEDIAN (not mean)
- Color encoding: Country → Ranking_Tier
- Add red solid trendline (#FF0000)
- Title: "Academic vs Employer Reputation by Performance Tier\n(Median Overall Score)"
- Keep scatter/bubble: x=Academic Reputation, y=Employer Reputation, size ∝ (median) Overall Score, legend = Ranking_Tier

GPT-5.2:

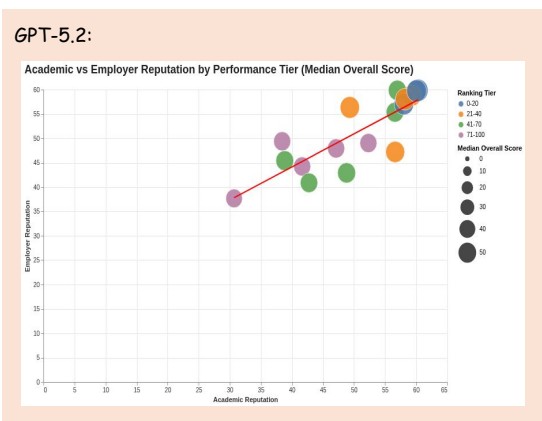

Ground Truth:

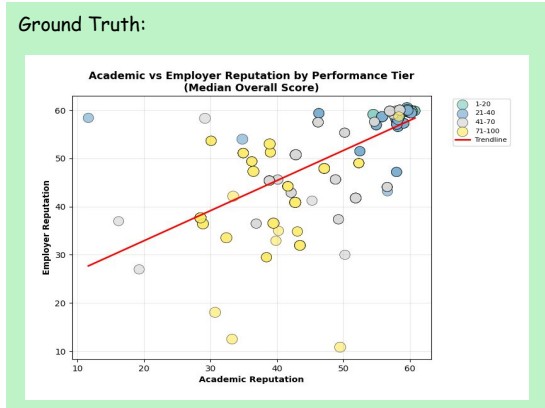

Error Analysis:
- Too few points plotted → many groups/samples are missing.
- Tier label mismatch: uses "0–20" instead of "1–20".
- Axis ranges differ: both axes start at 0, creating large empty space.
- Trendline is truncated (starts mid-range) and does not span the full x-range like gold.
- Trendline is not labeled in the legend (gold shows "Trendline").
- Bubble sizing is overly large and inconsistent, causing heavy overlap near the top-right.
- Title formatting differs: missing the two-line layout used in gold.

Percentage Of Score: 40.625%

*Figure 38.* A sample bad case of in DV-Evol using **Vega-lite**.

## DV-Evol Bad Case vega-lite : Visual mapping failure

**Question: Convert the 1990–2014 regional CO2 emissions data from mean to country-level median aggregation and replace the stacked area chart with a faceted line chart using specific HEX color mappings.**

**Gemini-3-pro-preview:**

**Ground Truth:**

**Plot code:**

```
{
  "$schema": "https://vega.github.io/schema/vega-lite/v5.json",
  "data": {"url": "result.csv"},
  "facet": {"field": "Region", "columns": 2, "header": {"title": null}},
  "spec": {
    "width": 300,
    "height": 200,
    "mark": {"type": "line", "point": true},
    "encoding": {
      "x": {"field": "Year", "type": "quantitative", "scale": {"domain": [1990, 2014]}, "axis": {"format": "d"}},
      "y": {"field": "CO2", "type": "quantitative", "title": "Median CO2"},
      "color": {
        "field": "Region",
        "scale": {
          "domain": ["North America", "Middle East & North Africa", "Europe & Central Asia", "Latin America & Caribbean"],
          "range": ["#9467bd", "#d62728", "#ff7f0e", "#2ca02c"]
        },
        "legend": null
      }
    }
  },
  "resolve": {"scale": {"y": "independent"}}
}
```

**Error Analysis:** 1. Data Integrity
❌1.1 Trend/Pattern: 0/1 The distribution curves successfully mirror the multi-modal trends of the Ground Truth.
❌1.2 Completeness: 0/1 Critical data series or specific years are missing from the visualization.
❌1.3 Label Accuracy: 0/1 Axis tick values and labels do not precisely match the original data points.
❌1.4 Data Consistency: 0/1 Proportional relationships between income groups remain accurate across the X-axis.
2. Style Imitation
✅2.1 Color Palette: 1/1 The model correctly applied the designated HEX codes for all income categories.
❌2.2 Chart Elements: 0/1 Specific ridgeline outlines and gridline styles were not properly reproduced.
✅2.3 Background: 1/1 The background color and light/dark mode are consistent with the Ground Truth.
❌2.4 Font and Text Styles: 0/1 Font weight and sizing for titles and labels deviate from the original style.
❌2.5 Element Spacing: 0/1 Distances between labels and the plot area are uneven, creating a cluttered look.
❌2.6 Borders and Shadows: 0/1 Visual effects like drop shadows or specific border strokes were omitted.

3. Layout & Aesthetics
❌3.1 Labels: 0/1 Legend and title positioning are misaligned compared to the standard layout.
✅3.2 Readability: 1/1 Excessive ridge overlap and small font sizes compromise the overall legibility.
❌3.3 Axis Alignment and Spacing: 0/1 The axes are poorly calibrated, causing data to shift away from intended baselines.
❌3.4 Element Alignment: 0/1 Graphical components fail to align precisely with their respective axis ticks.
❌3.5 Proportionality and Aspect Ratio: 0/1 An incorrect aspect ratio has led to vertical distortion of the ridge shapes.
❌3.6 Overall Balance: 0/1 The composition lacks visual equilibrium, making the chart difficult to interpret quickly.

**Score: 3/16**

*Figure 39.* A sample bad case of **Visual mapping failure** in DV-Evol using **Vega-lite**.

individual players using an arbitrary top-$k$ filter rather than questioning whether aggregation at a higher semantic level is required. This assumption-first behavior leads to charts with low semantic value and violates the rubric's expectation of logical aggregation, highlighting a systematic reluctance to engage in disambiguating interaction.

**Inquiry deficit and visualization strategy misalignment.** Beyond avoidance, some failures stem from Inquiry Deficit (IDf), where agents make unilateral design choices in the presence of layout or analytical ambiguity. In Fig. 41, the agent selects a dual-axis bar–line chart to compare revenue and sales volume, despite the task implicitly requiring a comparative decomposition of volume versus value across tiers. The absence of clarification leads to both visual and logical misalignment, compounded by metric misuse (e.g., summing quantities instead of counting identifiers), demonstrating how insufficient inquiry propagates errors across reasoning and presentation.

**Cognitive–execution gaps from brittle data parsing.** A third major error source is the Cognitive-Execution Gap (CEG), where correct high-level intent fails to translate into executable logic due to brittle intermediate reasoning. Fig. 42 shows a representative case in which an incorrect year-extraction heuristic assigns a constant timestamp to all rows, causing a subsequent temporal filter to discard the entire dataset and yield an empty visualization. Such failures indicate that agents often lack robust checks between reasoning and execution, allowing small parsing mistakes to cascade into total task failure.

**Visual design mismatch under complex semantic requirements.** Errors categorized as Visual Design Mismatch (VDM) arise when agents fail to instantiate the requested visual semantics, even if execution succeeds. In Fig. 43, a prompt requiring a complex distribution chart with statistical summaries is reduced to a simple bubble scatter plot, omitting medians, IQRs, and appropriate categorical color encoding. Additional legend occlusion and inappropriate use of continuous color for categorical variables further degrade readability, indicating a gap between semantic intent and visual encoding choices.

**Technical collapse and tool-facing instability.** Finally, Technical Collapse (TC) represents a hard failure mode in which repeated runtime errors and environment mismanagement prevent task completion altogether. As shown in Fig. 44, persistent exceptions during data filtering and missing execution outputs result in no visualization being produced. These cases underscore the fragility of interactive agents when faced with tool-level errors, where insufficient debugging and state inspection capabilities force premature task abandonment.

*Table 28.* Fine-grained Error Analysis for the DV-INTER task. The **highest** error rate per metric is highlighted in  orange , and the **lowest** in  green . *Abbreviations*: **SL**: Speculative Logic, **TS**: Task Simplification, **PB**: Parametric Bias, **UM**: UI Mismatch, **DB**: Default Bias, **FD**: Focus Dissipation, **SF**: Semantic Fragmentation, **HL**: Heuristic Logic Approx, **ID**: Intent Drift, **PE**: Precision Error, **MM**: Metaphor Mismatch, **LC**: Layout Conflict, **DD**: Debugging Deadlock, **TR**: Tooling Rigidity.

| Method | Interactive Avoidance | | | | Inquiry Deficit | | | Cognitive-Execution Gap | | | Visual Design | | Technical Collapse | |
|---|---|---|---|---|---|---|---|---|---|---|---|---|---|---|
| | SL | TS | PB | UM | DB | FD | SF | HL | ID | PE | MM | LC | DD | TR |
| DeepSeek-V3.2 | 17.04% | 1.42% | 7.94% | **8.51%** | 1.19% | 12.70% | 6.15% | 1.13% | 12.36% | 0.87% | **13.61%** | **16.00%** | 1.44% | 0.64% |
| Gemini-2.5-Pro | 15.40% | 1.08% | 5.35% | 1.23% | 2.00% | 6.57% | 0.79% | 3.32% | **45.38%** | 2.12% | 7.42% | 1.05% | 2.22% | 14.36% |
| Gemini-3-Pro-Proview | 16.59% | 8.11% | 8.73% | 0.96% | **12.34%** | 7.47% | 0.44% | 16.43% | 16.90% | 8.27% | 0.38% | 0.80% | 1.59% | 0.99% |
| GLM-4.7 | **33.32%** | **13.85%** | 7.06% | 5.24% | 1.59% | 1.13% | 1.02% | 6.48% | 14.94% | 5.55% | 8.12% | 0.34% | 0.71% | 0.65% |
| GPT-4.1 | 27.84% | 1.01% | 4.59% | 1.07% | 1.35% | 3.31% | 1.35% | 6.11% | 41.74% | 0.50% | 1.54% | 0.74% | 0.42% | 8.43% |
| GPT-5.2 | 8.12% | 0.68% | **16.33%** | 0.60% | 8.01% | 0.66% | 1.13% | **24.74%** | 22.90% | **12.43%** | 0.65% | 1.52% | 1.14% | 1.09% |
| Qwen3-235B-A22B | 30.11% | 0.33% | 7.82% | 1.78% | 1.26% | 4.14% | 1.46% | 1.59% | 33.54% | 0.34% | 0.58% | 0.96% | 0.33% | **15.76%** |
| Qwen3-8B | 0.57% | 1.53% | 1.44% | 0.71% | 1.15% | **13.40%** | **7.66%** | 0.90% | 23.81% | 5.13% | 1.03% | 0.62% | **33.40%** | 8.65% |

*Table 29.* Dimensional Summary of Error Rates for DV-INTER. **Maximum** error modes per agent are highlighted in orange , and **minimums** in green . *Abbreviations*: **IA**: Interactive Avoidance, **ID**: Inquiry Deficit, **CEG**: Cognitive-Execution Gap, **VDM**: Visual Design Mismatch, **TC**: Technical Collapse.

| Method | IA | ID | CEG | VDM | TC | Total |
|---|---|---|---|---|---|---|
| DeepSeek-V3.2 | 34.91% | 19.04% | 14.36% | **29.61%** | 2.08% | 100% |
| Gemini-2.5-Pro | 23.06% | 9.36% | **50.82%** | 8.47% | 8.29% | 100% |
| Gemini-3-Pro-Proview | 34.39% | 20.25% | 41.60% | 1.18% | 2.58% | 100% |
| GLM-4.7 | **59.47%** | 3.74% | 26.97% | 8.46% | 1.36% | 100% |
| GPT-4.1 | 34.51% | 6.01% | 48.35% | 2.28% | 8.85% | 100% |
| GPT-5.2 | 25.73% | 9.80% | **60.07%** | 2.17% | 2.23% | 100% |
| Qwen3-235B-A22B | 40.04% | 6.86% | 35.47% | 1.54% | 16.09% | 100% |
| Qwen3-8B | 4.25% | **22.21%** | 29.84% | 1.65% | **42.05%** | 100% |
| **Avg. Error** | 32.05% | 12.16% | **38.44%** | 6.92% | 10.44% | – |

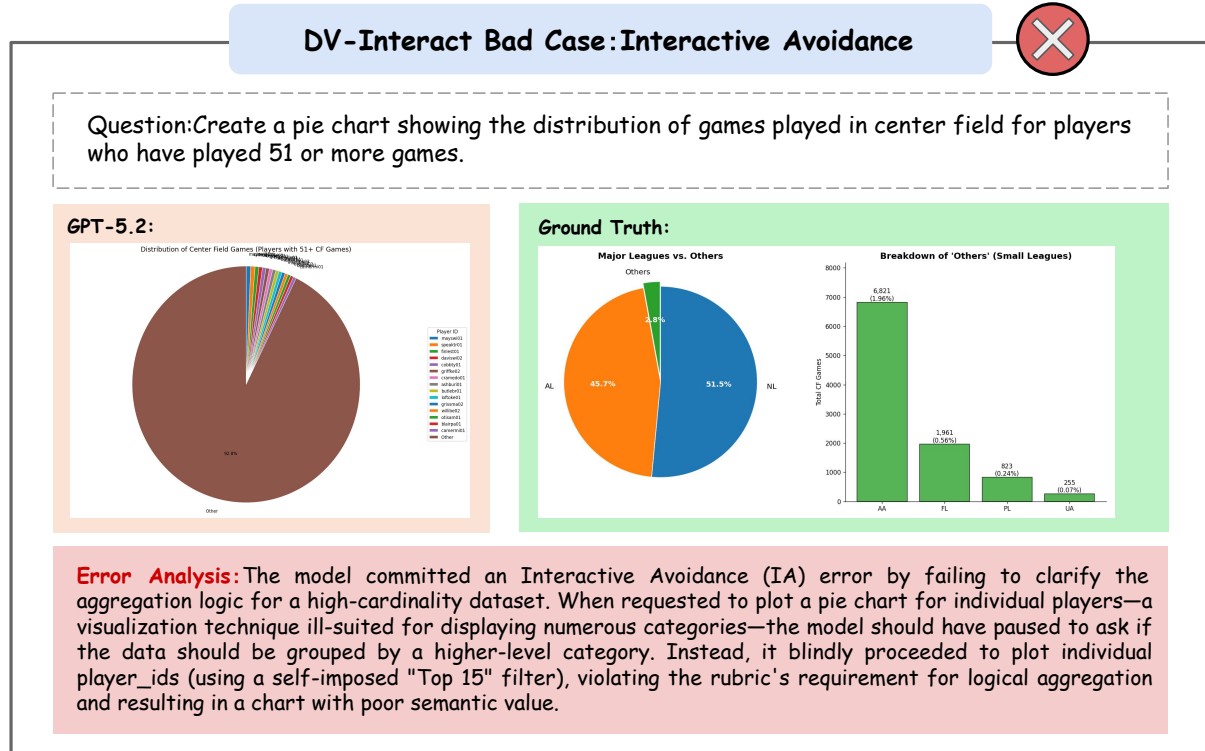

*Figure 40.* A sample bad case of **Interactive Avoidance** in the DV-Inter.

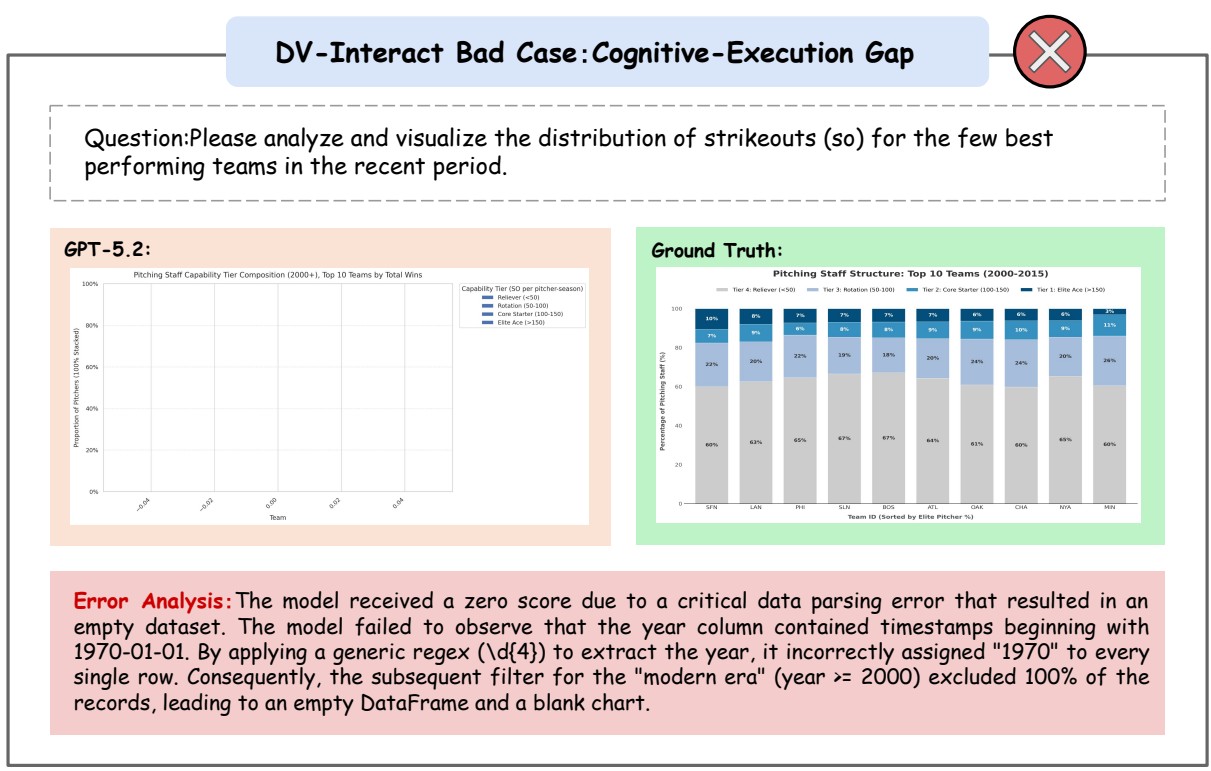

*Figure 41.* A sample bad case of **Inquiry Deficit** in the DV-Inter.

*Figure 42.* A sample bad case of **Cognitive-Execution Gap** in the DV-Inter.

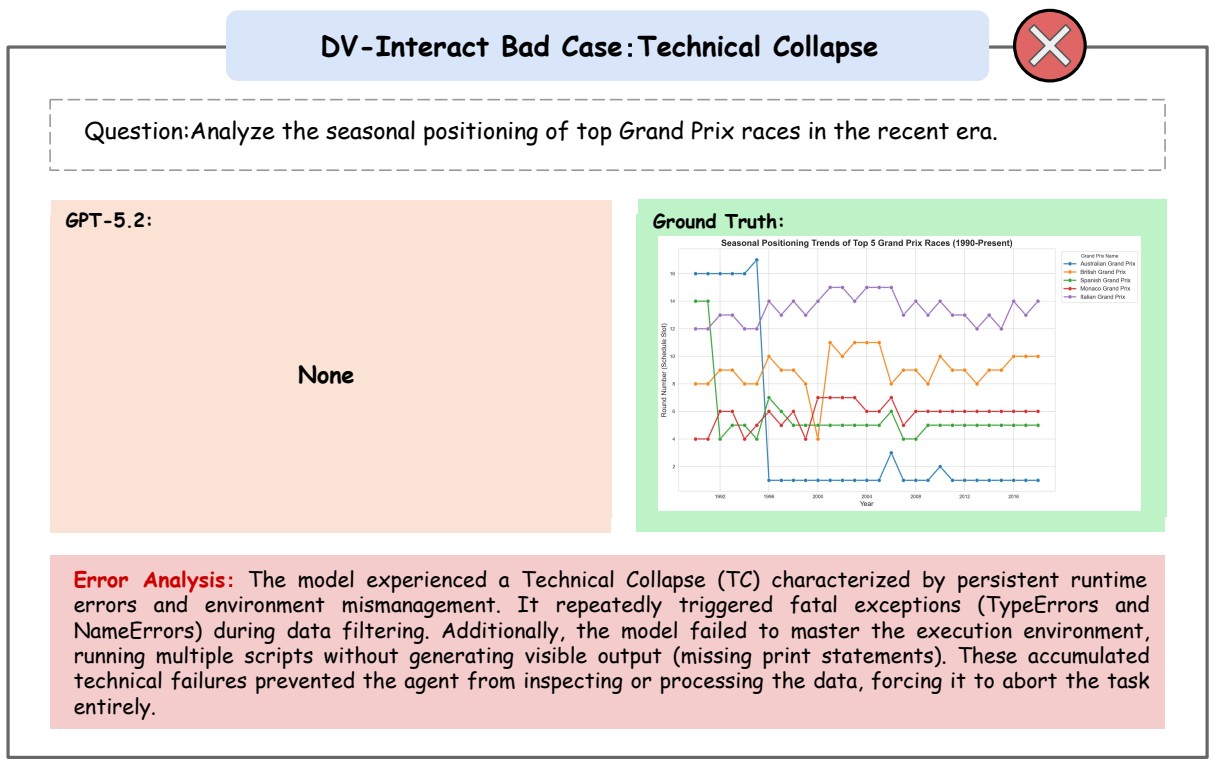

**DV-Interact Bad Case:Visual Design Mismatch** ✖

Question:Analyze the postal code values for the high-density districts based on the most recent updates. Create a complex distribution chart where the communication identifier weights the marker size.

**GPT-5.2:**

**Ground Truth:**

**Error Analysis:** The model exhibited a clear Visual Design Mismatch (VDM). It failed to reproduce the requested "complex distribution chart," generating only a basic bubble scatter plot that lacked statistical distribution details (e.g., medians and IQRs). Additionally, the model made critical errors in data-to-visual mapping by applying a continuous color scale to an invariant time variable, resulting in a monotonous palette devoid of categorical distinctiveness. The readability was further degraded by legend occlusion issues.

*Figure 43.* A sample bad case of **Visual Design Mismatch** in the DV-Inter.

**DV-Interact Bad Case:Technical Collapse** ✖

Question:Analyze the seasonal positioning of top Grand Prix races in the recent era.

**GPT-5.2:**

None

**Ground Truth:**

**Error Analysis:** The model experienced a Technical Collapse (TC) characterized by persistent runtime errors and environment mismanagement. It repeatedly triggered fatal exceptions (TypeErrors and NameErrors) during data filtering. Additionally, the model failed to master the execution environment, running multiple scripts without generating visible output (missing print statements). These accumulated technical failures prevented the agent from inspecting or processing the data, forcing it to abort the task entirely.

*Figure 44.* A sample bad case of **Technical Collapse** in the DV-Inter.

