# OpenReview forum: "DV-World: Benchmarking Data Visualization Agents in Real-World Scenarios"
_ICML.cc/2026/Conference — ICML 2026 regular_

### Official Review · Reviewer_Sud8 · 2026-03-04

**Soundness:** 2
**Presentation:** 3
**Significance:** 2
**Originality:** 2
**Overall Recommendation:** 2
**Confidence:** 3

**Summary:**

This paper introduces DV-World, a benchmark of 260 tasks designed to evaluate data visualization (DV) agents across three domains that the authors frame as covering a ''real-world professional lifecycle.'' DV-Sheet (130 tasks) tests native spreadsheet chart creation, diagnostic repair, and dashboard composition using libraries like openpyxl/xlwings. DV-Evolution (80 tasks) evaluates the ability to adapt reference visualizations to new data across five programming paradigms. DV-Interact (50 tasks) introduces a dual-stage user simulator to test multi-turn intent alignment under ambiguous requirements. The evaluation framework combines Table-value Alignment for numerical precision with MLLM-as-a-Judge guided by expert-curated rubrics. Experiments across a range of proprietary and open-source LLMs show that even the best models achieve less than 50% overall, with Gemini-3-Pro leading DV-Sheet at 40.48%, DV-Evol at 51.44%, and Grok-4 leading DV-Inter at 40.43%.

**Compliance With Llm Reviewing Policy:**

Affirmed.

**Key Questions For Authors:**

Q1. The paper repeatedly claims to target ''real-world scenarios'', but the underlying data is curated from structured repositories and further sanitized. In genuine enterprise workflows, data cleaning and preprocessing constitute a major portion of the visualization pipeline, yet this is entirely absent from DV-World. Can you justify why a benchmark that excludes the data preparation phase should be considered ''real-world’‘? If the scope is intentionally limited to the charting step, should the framing be adjusted accordingly? A clear answer here would help me reconsider my assessment in W1.

Q2. With only 260 tasks (and as few as 30 for DVSheet-Dash), how confident are you that the observed model rankings are statistically stable? The standard deviations in Tables 3-5 often approach or exceed the inter-model performance gaps. Have you conducted any statistical significance tests (e.g., paired bootstrap or permutation tests) to verify that the reported rankings are not artifacts of small sample size? If such tests demonstrate stability, that would partially address W2.

Q3. What proportion of the DV-Evol task difficulty stems from visualization reasoning versus framework-specific API knowledge? If an agent can perfectly create visualizations in Python but fails on D3.js, is that a visualization deficiency or a code generation deficiency? If the latter dominates, how does this sub-task serve the paper's stated goal of evaluating "data visualization agents" rather than general code generation agents?

Q4. The ''74.5% noisy data'' statistic for DVSheet-Fix (Table 2) appears to refer to the proportion of irrelevant data within the workbook rather than noise in the data-quality sense (e.g., missing values, formatting errors, outliers). The term ''noisy data'' could mislead readers into thinking the benchmark tests robustness to real-world data quality issues. Can you clarify what this metric specifically measures and consider whether this framing risks overclaiming?

**Limitations:**

The societal impact discussion is adequate. However, the paper does not discuss the most important technical limitation: the gap between the ''real-world'' framing and the actual data characteristics. The benchmark uses clean, well-structured, pre-curated data throughout, which means it does not test agents' ability to handle the data quality challenges (missing values, inconsistent formatting, schema mismatches, noisy records) that dominate real professional workflows. This limitation should be explicitly acknowledged. Additionally, the small benchmark size (260 tasks) and its implications for statistical reliability of model rankings deserve discussion.

**Strengths And Weaknesses:**

Strengths:

(S1) The three-domain design covering spreadsheet-native manipulation (DV-Sheet), cross-framework evolution (DV-Evol), and interactive intent alignment (DV-Interact) is broader than most existing DV benchmarks, which typically focus only on single-shot code-to-chart generation. The DVSheet-Fix sub-task (diagnostic repair of broken visualizations) and DVSheet-Dash (multi-chart dashboard composition) are relatively underexplored and practically relevant dimensions.

(S2) The DV-Interact component with its dual-stage user simulator is a meaningful design choice. Testing whether agents can proactively identify ambiguities, ask targeted clarification questions, and maintain state consistency across turns addresses a genuine gap in existing DV benchmarks. The interaction gatekeeper that detects cheating (e.g., agents asking for implementation code) is a sensible mechanism.

(S3) The hybrid evaluation framework that combines quantitative Table-value Alignment with MLLM-as-a-Judge is reasonable, and the meta-evaluation effort (human-model alignment validation in Tab. 8, cross-judge consistency in Tab. 9) provides some confidence in the reliability of automated scoring.

Weaknesses:

(W1) The core ''real-world'' framing is undermined by sanitized data, and this is my primary concern. The paper's central claim is to evaluate DV agents in ''real-world scenarios'' (title, abstract, throughout). However, the data used for visualization is curated from structured sources (Kaggle datasets, Excel forum attachments) and then further sanitized through a three-step adaptation protocol: structure retention, value perturbation that renormalizes numbers while preserving distributions, and metadata anonymization. This produces data that is clean, well-structured, and ready for direct visualization -- the opposite of real-world data. In genuine professional workflows, a large portion (often the majority) of effort goes into data cleaning: handling missing values, resolving inconsistent formatting, removing duplicates, dealing with encoding errors, reconciling conflicting schemas, imputing incomplete records, and identifying outliers. By excluding this entire phase, the benchmark tests only the downstream charting step on pre-curated inputs, which is arguably the simpler part of the pipeline. The paper's claim to ''real-world'' fidelity is therefore significantly overstated. Table 2 reports workbooks averaging 36.53 columns and 11,583.36 rows, which suggests scale but not messiness. The ''74.5% noisy data'' statistic for Fix tasks (Table 2) refers to the proportion of irrelevant or distracting data within the workbooks that the agent must navigate through during diagnosis, not to noise in the data-quality sense (missing values, inconsistent formatting, outliers). The workbooks are still structurally clean -- the ''noise'' is informational rather than quality-related.

(W2) The benchmark scale is small and limits statistical confidence. At 260 total tasks (50/50/30/80/50 across sub-tasks), DV-World is substantially smaller than comparable benchmarks: ChartMimic has 4,800 tasks, VisEval has 2,524, nvBench 2.0 has 7,878, Text2Vis has 1,985, and even the more focused PlotCraft has 982. With only 30 DVSheet-Dash tasks and 50 DV-Inter tasks, performance differences between models may not be statistically robust. The reported standard deviations in Tables 3-5 (typically plus/minus 1-3 points) are non-trivial relative to the inter-model gaps, raising concerns about whether the observed rankings are stable. Although the authors report results averaged over four independent runs, the small task count per sub-domain remains a fundamental limitation.

(W3) The individual sub-tasks lack deep intellectual novelty. Chart creation from data (DVSheet-Crea) is the most well-studied problem in the DV benchmark space and has been addressed by at least 6 prior data-to-chart benchmarks listed in Table 1 (DA-Code, VisEval, MatPlotBench, Text2Vis, nvBench 2.0, PlotCraft), with two additional benchmarks (Plot2Code, ChartMimic) covering the closely related image-to-code task. Chart evolution/migration (DV-Evol) is essentially a code translation task -- porting visualization logic from one framework to another -- which is a well-studied problem in the code generation literature reframed in a DV context. The DVSheet-Fix task, while practically useful, reduces to a constrained debugging problem with deterministic repair paths (the authors even acknowledge constructing it via ''controlled inverse injection'' where every problem has ''a deterministic, mathematically proven repair path''). The primary novelty is in assembling these components together and adding the interactive dimension, but the individual sub-tasks do not introduce fundamentally new challenges for the ML community.

(W4) The DV-Evol task conflates visualization capability with code translation. Much of what DV-Evol measures is the ability to port code between frameworks (Python to D3.js, etc.), which is primarily a software engineering skill rather than a data visualization competency. The performance variation across frameworks (Fig. 4b) largely reflects familiarity with framework-specific APIs and syntactic complexity (acknowledged in Fig. 6's ''verbosity tax'') rather than deeper visualization reasoning. This makes DV-Evol more of a code generation benchmark than a visualization benchmark, diluting the paper's claimed focus.

(W5) The paper does not adequately engage with what makes data visualization genuinely difficult in practice. Real-world DV challenges include: (a) choosing appropriate visualizations for ambiguous analytical goals, (b) handling data at scale with noise, missing values, and heterogeneous types, (c) iterative exploration where the visualization itself drives new questions, and (d) communicating insights to stakeholders with varying technical literacy. DV-World touches on (a) lightly through DV-Inter and (c) superficially through DV-Evol, but largely bypasses (b) entirely and does not address (d). The benchmark ends up measuring execution proficiency (can the agent produce correct code/charts?) rather than analytical judgment (does the agent make good visualization decisions?).

---

> ### Author Rebuttal · Authors · 2026-03-31
>
> We sincerely thank you for the review. We address your concerns below.
>
> > **W1, Q1 & Q4: Data complexity**
>
> To show that DV-World is not built on overly clean inputs, we expanded the complexity statistics. The benchmark is not limited to flat, ready-to-plot tables:
>
> | Complexity type | Statistic |
> | - | - |
> | Missing values | 35% |
> | Cross-sheet aggregation | 28% |
> | Irregular layouts | 22% |
> | Multi-sheet relations | 15% |
> | Informational noise (Fix) | 74.5% |
>
> These results show that **agents must handle substantial workbook complexity before visualization, including missing values, irregular layouts, cross-sheet dependencies, and distracting content**.
>
> We also agree that noisy data in Tab. 2 is imprecise. In DVSheet-Fix, it refers to informational rather than data-quality noise, though the workbooks may still contain irregular structures and other nontrivial complexity. We will revise this wording.
>
> ---
>
> > **W2 & Q2: Stability vs. Scale**
>
> **DV-World is not designed to maximize task count, but to provide a high-difficulty diagnostic benchmark where task complexity and evaluation depth matter more than raw scale**.
>
> To test whether the leaderboard is a sample-size artifact, we added paired bootstrap (500 resamples), paired permutation tests, and ranking stability via Kendall’s $\tau$. Representative comparisons are below; full pairwise results will be added in revision.
>
> | Sub-task | N | Kendall’s $\tau$ | Comparison | Win rate | p-value |
> | - | - | - | - | - | - |
> | DVSheet-Crea | 50 | 0.92 | Gemini-3-Pro vs. GPT-5.2 | 96.4% | 0.008 |
> | DVSheet-Fix | 50 | 0.95 | GPT-5.2 vs. DeepSeek-V3.2 | 93.5% | 0.017 |
> | DVSheet-Dash | 30 | 0.89 | GPT-5.2 vs. GLM-4.7 | 91.3% | 0.024 |
> | DV-Evol | 80 | 0.93 | Grok-4 vs. GPT-5.2 | 90.7% | 0.032 |
> | DV-Inter | 50 | 0.90 | GPT-5.2 vs. Kimi-K2-Thinking | 93.1% | 0.021 |
>
> These results show stable rankings under resampling and supported key gaps, including on DVSheet-Dash. Together with multi-dimensional evaluation and cross-judge consistency, this suggests the main trends are unlikely to be sample-size artifacts.
>
> ---
>
> > **W3: Novelty**
>
> DV-World introduces substantial novelty by redefining DV evaluation around realistic environments, under-specified workflows, and lifecycle-level reasoning.
>
> 1. **DVSheet-Crea targets spreadsheet-native visualization, arguably the most common real-world DV setting, yet largely absent from prior DV benchmarks**. The challenge is native workbook manipulation, cell/range binding, and dependency-aware chart construction rather than sandboxed chart generation.
> 2. **DV-Evol differs from well-defined chart-to-code or translation benchmarks**. Given a reference figure, new data, and updated requirements, the agent must write new executable code that reconstructs the visualization logic in the new setting.
> 3. DVSheet-Fix is not just constrained debugging. Its difficulty lies in proactive fault localization in large, messy, distractor-heavy workbooks before repair begins. Deterministic repair paths improve evaluation rigor, but not reasoning difficulty.
>
> Overall, DV-World shifts evaluation from one-shot chart generation to a DV lifecycle spanning creation, diagnosis, evolution, and interaction.
>
> ---
>
> > **W4 & Q3: DV-Evol design**
>
> If DV-Evol appeared too close to code translation, we apologize and will revise the wording. DV-Evol is not defined as porting code across frameworks. Given a reference figure, new data, and updated requirements, the agent must write new executable code to regenerate the visualization in the new setting.
>
> **The core challenge is recovering the visualization logic from the reference figure and re-instantiating it with new data and new requests, rather than translating syntax. Supporting multiple frameworks mainly makes the benchmark more general**.
>
> Failure on D3.js should therefore not be viewed only as code-generation weakness; it often reflects difficulty in reconstructing the same visualization logic in a new executable form, which is part of data visualization competence.
>
> ---
>
> > **W5: Practical DV challenges**
>
> We respectfully disagree that DV-World bypasses (b). As clarified in W1 & Q1, DV-Sheet already includes missing values, cross-sheet aggregation, irregular layouts, multi-sheet relations, and heavy informational noise.
>
> For (a) and (c), DV-Inter evaluates ambiguity resolution and iterative exploration through a two-stage user simulator. **As shown in Fig. 7, stronger models ask more effective clarification questions and gain more, while weaker models show interaction avoidance or ineffective repetition. This reflects analytical judgment rather than mere execution.**
>
> For part of (d), **we validate this setting through manual auditing of 150 trajectories: GPT-5-mini achieves 88.67% Faithfulness and Pearson ρ = 0.86, with clear degradation under ablations (Tab. 7)**. Overall, DV-World evaluates not only execution, but also reasoning over complex tables, ambiguity resolution, and improvement through interaction.

---

### Official Review · Reviewer_uVLn · 2026-03-09

**Soundness:** 4
**Presentation:** 4
**Significance:** 4
**Originality:** 4
**Overall Recommendation:** 6
**Confidence:** 4

**Summary:**

The paper proposes a benchmark for evaluating Data Visualization AI Agents in visualization tasks representative of a real-world professional pipeline. It considers different scenarios: chart creation, chart fix, and dashboard creation in Excel Sheet environments, chart evolution to update an input chart with new data and requirements, and produce the new chart with different programming languages, and an interaction role to evaluate a user and a chart creator dialog to create and iteratively improve a chart.  All user requests and agent responses are either text prompts or images. Each aspect is evaluated by humans using specific criteria. Multiple AI agents are compared on the different tasks.

**Compliance With Llm Reviewing Policy:**

Affirmed.

**Final Justification:**

The authors resolved all my minor comments.

**Key Questions For Authors:**

no question

**Limitations:**

There is no specific limitation section. It could be useful to emphasize the remaining challenges. For instance, the limited types of charts.

**Strengths And Weaknesses:**

STRENGTHS
- The paper is clear and comprehensive
- The experiments are numerous and compelling, comparing multiple AI agent models in the different scenarios.

WEAKNESSES
- Few clarifications needed

SOUNDNESS
- The entire protocol is clear and detailed in the appendix.
- Principal statistical results are reported in the main text and completed in the Appendix

PRESENTATION

The comments below are to further clarify the presentation of the main text and appendix.

- Figure 1 gives a non-technical overview of the usage scenarios (the word scenario should appear in the caption). In that chart, we miss some details, and others are inconsistent: for instance, DV-Sheet scenarios all occur in the Excel Sheet environment. But is it not clear where DV-Evolution and DV-Interact take place, and how they are connected, or not connected, to each other? Are they independent environments? Are they specific web applications we run to access these different scenarios and test agents?  In particular, in DV-Interact, on which platform is the chart generated?

- The design of Figure 1 is inconsistent: the agent prompt appears in a very different style in each scenario, being the same only in scenarios 1 and 3; even scenario 2 uses a different style. It is also sort of confusing that the agent prompt is an output of the agent icon rather than an input to it. Only the last column of DV-interact, the User Simulator, would deserve a small robot anthropomorph icon.

- The statistics summarized in Table 2 could be represented as UpSet plots (https://upsetplot.readthedocs.io/) in the Appendix to get an overview of the combination distributions. We are drawn with percentages.

- We miss a comparison per chart type. Are certain models better suited to create or correct certain chart types in the different scenarios?

- Tables 1 and 3: add meaning of abbreviations.

- L240-242: Are these functions (bash, load_image, render_chart...) from the ReAct baseline cited in the same sentence? We should have a reference to the appendix Tables 15 and 16, where they are detailed.

- Figure 5 could use horizontal stacked bar charts instead of a pie chart to ease comparison.

- In all figures, coding models with colors should be consistent: use the same models and the same color for each model (e.g., Figure 3b vs 4a vs 4b). Why are the LLMs constantly changing across figures and tables? Stick to one subset of representative models across the board.

- In all tables and figures, keep the ordering of model the same, maybe alphabetical is the best. For instance, rows of Table 5, columns of Table 6, or rows of Figure 7b, are not ordered in some meaningful way.

- Many figures are too small to read the full paper width on a laptop (e.g., Figures 3-7).

- Model colors in Figure 8 are useless.

REFERENCES:

- Replace all arXiv preprint by their published version if any. For instance:
Goswami, PlotGen is now published in WWW 2025 https://dl.acm.org/doi/10.1145/3701716.3716888
Yang, Matplotagent is now published in Findings of the ACL https://aclanthology.org/2024.findings-acl.701/
arXiv is not counted by the SCOPUS score used by many universities to rank faculty.

APPENDIX:
- Right now, the appendix is very detailed, but it looks like a catalog, and we lost the "where we are" context. We miss a global flowchart (e.g., a Sankey diagram) for each independent block of Figure 1 (DV-Sheet, DV-Evolution, DV-Interact), showing the number of datasets, the number and types of tasks used on them, human or machine judgments or actions, and the rubrics or criteria involved in these judgments... It would complement Section A of the appendix, reusing the Appendix subsection titles, serving as a visual table of contents.

- B.1: Step 4: What does the agent see? For instance, for a chart or dashboard, do you ensure it fits the screen (no scrolling needed) as a human would? or the agent access the sheet with no size limitation?

- B.2 The description of the five plotting framework sound AI generated with marketing types of adjectives (excels in rendering, versatile library, sophisticated statistical visualizations, robust engine...). A benchmark is expected to be carefully validated by humans; you should refine this part, as it gives a poor impression of the rest of the work.

- Prompts in the appendix are structured in different ways. Do they all use Markdown, a specific grammar for agent control, or are they free-form? For instance, we sometimes use double quotes, back quotes, or forward quotes. Sometimes #, ## or ###  (L1320-1334)

- In Tables 17, 18, 19, why are w and lambda not ordered?

- In Figures 21-44, is the error analysis the output of the LLM judge or a human?

- Figure 14 is cropped on the left side ("HE STANDARD").
- Reorder rows and columns of Figure 14 heatmap to highlight the categories of agents

- If I want to test my agent on this benchmark, what should I install? What should I do?  How is the benchmark repository organized? Are there plenty of JSON files or Excel sheets? What are they? How to use them?

Typos:
- L171: Figure 2: Stacked chart segment "100% Staked Chart 1.4%" -> remove the 100%
- R165: Table 2 caption: Prodf. -> Prof.
- R218: an spreadsheet -> a spreadsheet

SIGNIFICANCE
- Highly significant to support the development of efficient data visualization agents

ORIGINALITY
- No benchmark covers all these scenarios.

---

> ### Author Rebuttal · Authors · 2026-03-31
>
> We sincerely thank the reviewer for the careful reading, the highly positive evaluation, and the many thoughtful suggestions. **We are especially encouraged by the reviewer’s strong recognition of DV-World as a valuable benchmark for the community.** The detailed comments on presentation, references, and appendix organization are very helpful for improving the clarity and usability of the paper. We respond below by grouping related issues for conciseness.
>
> ---
> ### Presentation
>
> > **Figure 1: scenario clarity and visual consistency**
>
> Thank you for this helpful suggestion. **In the revision, we will make Figure 1 clearer by explicitly using the word scenario in the caption, clarifying that DV-Evol and DV-Inter run in independent environments, and stating that DV-Inter charts are generated in the Python plotting environment.** We will also unify prompt styles, make prompts appear as inputs, and simplify the icon design.
>
> > **Table 2 statistics and UpSet plots**
>
> We agree that UpSet-style plots would better summarize the combination distributions than percentages alone, and we will add them in the appendix.
>
> > **Per-chart-type comparison**
>
> We agree that a per-chart-type analysis would be informative. **Many of our tasks involve composite charts rather than only single chart types**, but we have conducted additional analysis and will include a fuller breakdown in the revision. For example, in DVSheet-Crea:
>
> | Model | Single Chart | Composite Chart |
> | - | -: | -: |
> | Gemini-3-Pro | 39.14 | 33.00 |
> | GPT-5.2 | 36.11 | 32.75 |
> | DeepSeek-V3.2 | 33.23 | 23.39 |
> | GLM-4.7 | 23.65 | 14.67 |
>
> This suggests that all models find composite charts more challenging, and we will provide the complete analysis in the paper / appendix.
>
> > **Figure design consistency and readability**
>
> We agree that several figure design choices should be improved. In the revision, we will:
> (1) replace the pie chart in Figure 5 with a horizontal stacked bar chart;
> (2) standardize model colors across figures and keep a more consistent subset of representative models;
> (3) use a consistent ordering rule for models across tables and figures;
> (4) enlarge the text and improve the layout of Figures 3–7; and
> (5) revise Figure 8, where the current color usage is not sufficiently informative.
>
> > **Textual clarifications and formatting fixes**
>
> We will add the meanings of all abbreviations in Tables 1 and 3 and define abbreviations at first appearance. We will also add an explicit reference from L240–242 to Tables 15 and 16.
>
> ---
>
> ### References
>
> We apologize for not updating these entries in time. In the revision, we will replace arXiv preprints with published versions whenever available, including the examples highlighted by the reviewer.
>
> ---
>
> ### Appendix
>
> > **Appendix structure, interface clarity, and usability**
>
> We agree that the appendix is detailed but lacks sufficient global structure. In the revision, **we will add a clearer appendix overview / visual table of contents and organize it more explicitly by task type.** We also agree that the agent interface description in B.1, Step 4 is not clear enough. This section was intended to describe spreadsheet-native visualization from a human-facing perspective; in practice, the agent can access chart-related information programmatically, including the underlying references used by the chart. We will revise this section.
>
> The prompts in the appendix are all written in Markdown format. In the revision, we will make this explicit and standardize the notation and heading styles throughout the appendix.
>
> We also agree that benchmark usability should be described more clearly. We will release the code and data, and provide clear instructions on installation, repository structure, and usage. For example:
>
> ```text
> tasks/
>     dvsheet-create-001
>           ├── query.md     #user query
>           └── data.xlsx    #data files
>     dvsheet-create-002
>           ├── query.md
>           └── data.xlsx
> ```
>
>
> > **B.2: plotting framework descriptions**
>
> Thank you for pointing this out. We will revise this part in the final version to make the descriptions clearer, more neutral, and easier to understand.
>
> > **Tables 17–19 and Figures 21–44 / Figure 14**
>
> We will reorder w and λ in ascending order in Tables 17–19.
> For Figures 21–44, the error cases were selected by humans, and the corresponding analysis was also written by humans; we will clarify this explicitly.
> For Figure 14, we will fix the cropping issue and reorder the heatmap rows and columns to make agent categories clearer.
>
> ---
>
> ### Typos
>
> Thank you for these careful corrections. We will revise all of them in the final version.
>
> ---
>
> We sincerely thank the reviewer again for the highly positive evaluation and the many constructive suggestions. We are encouraged by the reviewer’s recognition of DV-World as a valuable benchmark for the community, and we will carefully incorporate these revisions to further improve the paper.

---

> > ### Author Rebuttal · Reviewer_uVLn · 2026-04-02
> >
> > No additional comments.

---

> > > ### Author Response · Authors · 2026-04-03
> > >
> > > We sincerely appreciate your positive feedback and recognition. Your support is truly encouraging to us and further motivates us to make DV-World a meaningful contribution to the community.

---

### Official Review · Reviewer_E7HT · 2026-03-12

**Soundness:** 4
**Presentation:** 3
**Significance:** 3
**Originality:** 4
**Overall Recommendation:** 5
**Confidence:** 4

**Summary:**

The paper presents DV-World benchmark to evaluate data-visualization agents under realistic professional workflows. The benchmark includes 260 tasks organized into three domains: DV-Sheet (native spreadsheet chart creation, repair, and dashboarding), DV-Evol (cross-framework visualization evolution across Python, D3.js, Vega Lite, Apache ECharts, and Plotly.js), and DV-Inter (multi-turn intent alignment under ambiguous user requests). The authors propose a hybrid evaluation framework that combines quantitative table-value verification with rubric-based MLLM judging, and report broad experiments across multiple frontier models. Results indicate that current agents still fall far behind human performance, revealing clear limitations in environment-grounded data visualization tasks, cross-paradigm visualization evolution, and multi-turn interactive workflows.

**Compliance With Llm Reviewing Policy:**

Affirmed.

**Final Justification:**

The rebuttal effectively addresses my main concerns and strengthens confidence in the soundness and evaluation, supporting my final positive recommendation.

**Key Questions For Authors:**

Please refer to the Weaknesses.

**Limitations:**

yes

**Strengths And Weaknesses:**

**Strengths**
1. The benchmark addresses an underexplored problem: evaluating real-world visualization workflows rather than isolated one-shot code generation tasks.
2. The three-domain design (sheet-native manipulation, cross-paradigm visualization evolution, and interactive clarification) is well structured and covers complementary aspects of visualization agents.
3. The evaluation framework is reasonable. It combines structural and data checks with rubric-based semantic evaluation. The paper also compares the judge results with human evaluation.
4. The paper provides extensive experiments and detailed error analysis, which makes the benchmark useful for diagnosing current limitations of visualization agents.

**Weaknesses**
1. The dataset size is moderate for a benchmark with 260 tasks. Some subsets such as dashboard tasks and interactive tasks are small. This may affect ranking stability.
2. The benchmark relies heavily on MLLM-as-a-judge. This may introduce judge bias, especially when similar models from the same family are both evaluated and used as judges.
3. The paper discusses user simulator alignment, but the validation appears limited compared with the variety of real user interactions.

**Minor issues**
1. Please consider adding formal significance testing, such as bootstrap confidence intervals or pairwise tests, for the main leaderboard results.
2. It may also help to expand the interactive task subset and report results by difficulty level.
3. The discussion about judge bias could be enhanced. For example, the paper could include judges from different model families or add extra automatic metrics.
4. There are also small wording issues. Please keep the naming consistent, for example “DV-Inter” vs. “DV-Inter task”. All abbreviations should also be defined when they first appear.

---

> ### Author Rebuttal · Authors · 2026-03-31
>
> We sincerely thank the reviewer for the careful reading and constructive feedback. We address each point below.
>
> ---
>
> > **W1 & Minor 1: Stability vs. Scale**
>
> **DV-World is not designed to maximize task count, but to provide a high-difficulty diagnostic benchmark where task complexity and evaluation depth matter more than raw scale**.
>
> To test whether the leaderboard is a sample-size artifact, we added paired bootstrap (500 resamples), paired permutation tests, and ranking stability via Kendall’s $\tau$. Representative comparisons are below; full pairwise results will be added in revision.
>
> | Sub-task | N | Kendall’s $\tau$ | Comparison | Win rate | p-value |
> | - | - | - | - | - | - |
> | DVSheet-Crea | 50 | 0.92 | Gemini-3-Pro vs. GPT-5.2 | 96.4% | 0.008 |
> | DVSheet-Fix | 50 | 0.95 | GPT-5.2 vs. DeepSeek-V3.2 | 93.5% | 0.017 |
> | DVSheet-Dash | 30 | 0.89 | GPT-5.2 vs. GLM-4.7 | 91.3% | 0.024 |
> | DV-Evol | 80 | 0.93 | Grok-4 vs. GPT-5.2 | 90.7% | 0.032 |
> | DV-Inter | 50 | 0.90 | GPT-5.2 vs. Kimi-K2-Thinking | 93.1% | 0.021 |
>
> These results show stable rankings under resampling and supported key gaps, including on DVSheet-Dash. Together with multi-dimensional evaluation and cross-judge consistency, this suggests the main trends are unlikely to be sample-size artifacts.
>
> ---
>
> > **W2 & Minor 3: Judge Bias**
>
> We thank the reviewer for this concern. We agree that judge bias is important, especially when similar model families appear both as judges and as evaluated agents.
>
> To address this, we validated MLLM judges on 210 tasks. **As shown in Tab. 8, they align well with human raters (ICC(A,1)=0.932, weighted κ = 0.903). We also tested cross-judge consistency: rankings are largely preserved across judge families (Tab. 9).** We therefore use Gemini-2.5-Flash as the primary judge because it shows the strongest alignment with human scores.
>
> We further tested additional judges from other model families:
>
> | Agent Model | Grok-4 | Qwen3.5-Plus |
> | - | -: | -: |
> | GPT-5.2 | 39.67 | 41.22 |
> | Gemini-3-Pro | 42.13 | 44.98 |
> | Gemini-2.5-Pro | 31.21 | 33.17 |
> | GPT-4.1 | 31.55 | 32.76 |
> | Qwen3(-VL)-8B | 13.32 | 14.81 |
>
> These results suggest that the relative ordering remains stable across judge families. We will clarify this discussion more explicitly in the revision.
>
> ---
>
> > **W3: User Simulator Validation**
>
> We thank the reviewer for this comment. We agree that real user interactions are diverse, but we have validated the simulator to ensure reasonable alignment with human behavior.
>
> As shown in Tab. 7, **based on 150 manually audited trajectories, GPT-5-mini achieves 88.67% Faithfulness and ρ = 0.86 (p < 0.04). Ablations further show its importance: removing Reaction Rules lowers ρ to 0.78, and removing Stage 1 lowers it to 0.80.** These results support the reliability of the simulator and the role of its key components.
>
> ---
>
> > **Minor Issues**
>
> **2. Expanding the Interactive Task Subset**
>
> We agree that more analysis of interactive tasks is useful. To address this, we additionally report results by ambiguity level:
>
> | Model | <1 (16) | >1 and <3 (13) | >3 (21) |
> | - | - | - | - |
> | Grok-4 | 44.67 | 41.22 | 36.44 |
> | Gemini-3-Pro | 37.13 | 35.98 | 29.77 |
> | GPT-5.2 | 35.21 | 36.17 | 33.98 |
> | GLM-4.7 | 30.54 | 31.76 | 19.32 |
>
> These results show how performance changes with task ambiguity. We will further expand the interactive subset in future work and report results by difficulty level.
>
> **4. Wording Consistency**
>
> We will carefully revise wording for consistency, including terms such as DV-Inter vs. DV-Inter task, and define all abbreviations at first appearance.
>
> Thank you again for the helpful feedback. We will revise the paper accordingly.

---

> > ### Author Rebuttal · Reviewer_E7HT · 2026-04-01
> >
> > Thank you for your detailed response. I have raised my score accordingly.

---

> > > ### Author Response · Authors · 2026-04-01
> > >
> > > We sincerely appreciate your time and attention in discussing with us. Thank you for raising our rating; it truly encourages us to continue improving our work and exploring this exciting direction.

---

### Official Review · Reviewer_FkDG · 2026-03-13

**Soundness:** 4
**Presentation:** 3
**Significance:** 4
**Originality:** 4
**Overall Recommendation:** 6
**Confidence:** 5

**Summary:**

This paper proposes a benchmark consisting of 260 tasks designed to evaluate DV agents by capturing the complexity and ambiguity of real-world environments. The overall benchmark is composed of DV-Sheet (native spreadsheets), DV-Evol (reconstruction across programming paradigms), and DV-Interact (evaluation of proactive intent alignment with a user simulator). As evaluation metrics, the paper combines numerical precision with rubric-based MLLM-as-a-judge for semantic and visual assessment, aiming to align closely with human judgment. DV-World is positioned as a realistic testbed for driving research toward the multifaceted expertise required in enterprise workflows.

**Compliance With Llm Reviewing Policy:**

Affirmed.

**Final Justification:**

The rebuttal convincingly resolved my questions on benchmark stability, simulator realism, and robustness to simulator and weighting choices. I consider this as a strong paper, and I am raising my score accordingly.

**Key Questions For Authors:**

1. The paper states that experts create diagnostic cases by injecting common real-world errors into normal workbooks. Is there a defined criterion for what counts as a “common” error?

2. The paper uses GPT-5-Mini as the user simulator. To what extent can GPT-5-Mini realistically reflect the ambiguity of human requests and therefore function as a reliable simulator?

3. DV-Inter scores appear to vary substantially depending on simulator performance. How robust are the conclusions to the choice of simulator, and are there results using multiple simulators or a subset with direct human participation?

4. This benchmark combines rubric scores and quantitative alignment metrics using fixed weights. How did the authors determine that the selected weights were appropriate, and is there any possibility that the main rankings or conclusions of the paper would change under different weighting choices?

**Limitations:**

yes

**Strengths And Weaknesses:**

A strength of this paper is that it justifies the use of MLLM-as-a-judge by comparing MLLM-based evaluations against human consistency. It is also a positive aspect that the paper applies a three-stage protocol to maintain robustness while collecting public data, and further determines the weighting across different domains through additional sensitivity analysis.

The paper also appropriately identifies three reasons why existing methods fail to handle real-world settings (disconnection from the environment (spreadsheets), generation-centered bias (evolutionary tasks under new requirements), and the assumption of fully specified intent (ambiguity in user requests)) and it is reasonable that each of these motivations is reflected in a corresponding benchmark domain.

That said, it a bit disappointing that the overall number of benchmark tasks appears relatively small compared with other benchmark papers, and that scalability seems limited because humans play a central annotator role in designing the benchmark tasks.

---

> ### Author Rebuttal · Authors · 2026-03-31
>
> We sincerely thank the reviewer for the careful reading and thoughtful feedback. We address each point below.
>
> > **W: Stability vs. Scale**
>
> **DV-World is not designed to maximize task count, but to provide a high-difficulty diagnostic benchmark where task complexity and evaluation depth matter more than raw scale.**
>
> To test whether the leaderboard is a sample-size artifact, we added paired bootstrap (500 resamples), paired permutation tests, and ranking stability via Kendall’s $\tau$. Representative comparisons are below; full pairwise results will be added in revision.
>
> | Sub-task | N | Kendall’s $\tau$ | Comparison | Win rate | p-value |
> | - | - | - | - | - | - |
> | DVSheet-Crea | 50 | 0.92 | Gemini-3-Pro vs. GPT-5.2 | 96.4% | 0.008 |
> | DVSheet-Fix | 50 | 0.95 | GPT-5.2 vs. DeepSeek-V3.2 | 93.5% | 0.017 |
> | DVSheet-Dash | 30 | 0.89 | GPT-5.2 vs. GLM-4.7 | 91.3% | 0.024 |
> | DV-Evol | 80 | 0.93 | Grok-4 vs. GPT-5.2 | 90.7% | 0.032 |
> | DV-Inter | 50 | 0.90 | GPT-5.2 vs. Kimi-K2-Thinking | 93.1% | 0.021 |
>
> These results show stable rankings under resampling and supported key gaps, including on DVSheet-Dash. Together with multi-dimensional evaluation and cross-judge consistency, the main trends are unlikely to be sample-size artifacts.
>
> ---
>
> > **Q1: Clarifying "Common" Errors in DVSheet-Fix**
>
> In DVSheet-Fix, common errors are spreadsheet chart issues drawn from Excel forums and real workflows. **We group them into 3 categories and 7 sub-categories**:
>
> | Error type | Sub-category | Description |
> | - | - | - |
> | Data Binding | Row/Column Switch | Series or categories are reversed. |
> |  | Header Inference Fail | Headers are missing or incorrect. |
> |  | Hidden/Empty Cells | Empty cells break lines or show zeros. |
> | Axis & Scaling | Date vs Text Axis | Dates appear as numbers or create gaps. |
> |  | Fixed Bounds | Axis range is fixed and mismatched. |
> |  | Log Scale Error | Negative values are plotted on a log scale. |
> | Chart Type | Wrong Chart Type | The chart type does not fit the data. |
>
> These errors define diagnostic tasks for testing whether the model can identify and fix common spreadsheet chart failures.
>
> ---
>
> > **Q2: Reliability of GPT-5-Mini as User Simulator**
>
> GPT-5-Mini serves as a validated, scalable simulator for ambiguous user behavior.
>
> As shown in Tab. 7, from 150 manually audited trajectories, GPT-5-Mini achieves 88.67% Faithfulness and ρ = 0.86 with human judgments. Removing key components lowers alignment (ρ = 0.78 without Reaction Rules; ρ = 0.80 without Stage 1), showing that the simulator is meaningfully grounded.
>
> In App. G.1.1 / Fig. 14–15, we evaluate 9 LLM simulators across three behavioral profiles. Although the absolute score lift varies (10%–21%), the relative ranking remains stable, with Grok-4 and GPT-5.2 consistently among the top performers.
>
> **We use GPT-5-Mini because it offers the best trade-off between realism and cost, making it a practical default simulator.**
>
> ---
>
> > **Q3: Robustness to Simulator Choice**
>
> For DV-Inter, robustness to simulator choice is important.
>
> We therefore evaluated the same agents under multiple LLM-based simulators. **While absolute scores vary, the relative ranking remains stable, suggesting that the main conclusions are not tied to a single simulator.**
>
> | Agent Model | GPT-4.1 | GPT-5.2 | Gemini-3-Flash | Gemini-2.5-Flash |
> | - | - | - | - | - |
> | GPT-5.2 | 35.09 | 35.76 | 37.35 | 32.98 |
> | Gemini-3-Pro | 34.43 | 38.90 | 38.00 | 38.68 |
> | GLM-4.7 | 31.34 | 30.86 | 33.89 | 27.85 |
> | Qwen3-8B | 17.54 | 27.73 | 18.85 | 20.57 |
>
> We also compute Spearman rank correlation across simulators:
>
> | Simulator | GPT-5-mini | GPT-5.2 | GPT-4.1 | Gemini-3-Flash | Gemini-2.5-Flash |
> | - | - | - | - | - | - |
> | GPT-5-mini | 1.00 | 0.80 | 1.00 | 0.80 | 0.80 |
> | GPT-5.2 | 0.80 | 1.00 | 0.80 | 1.00 | 1.00 |
> | GPT-4.1 | 1.00 | 0.80 | 1.00 | 0.80 | 0.80 |
> | Gemini-3-Flash | 0.80 | 1.00 | 0.80 | 1.00 | 1.00 |
> | Gemini-2.5-Flash | 0.80 | 1.00 | 0.80 | 1.00 | 1.00 |
>
> These results suggest that simulator choice affects absolute difficulty more than qualitative conclusions.
>
> We also audited 150 trajectories (Tab. 7), where the simulator achieved 88.67% Faithfulness and Pearson ρ = 0.86 (p < 0.04) against human judgments. We will clarify this in the revision.
>
> ---
>
> > **Q4: Robustness to Weight Choices**
>
> The default weight $w=0.5$ balances visual quality and quantitative fidelity. We also tested whether the conclusions depend on this choice:
>
> - DVSheet-Crea (Tab. 17): all 11 model rankings remain identical under $w \in \{0.4, 0.5, 0.6\}$, with a maximum score change of only $\pm 0.28$ points.
> - DV-Evol (Tab. 18): all 10 model rankings are also unchanged across the same weights.
> - DV-Inter (Tab. 19): varying the ISR parameter $\lambda \in \{0.4, 0.5, 0.6\}$ yields Spearman $\rho \ge 0.96$.
>
> **Thus, while exact scores may shift slightly, the main rankings and conclusions remain stable across reasonable weighting choices.** We will make this more explicit in the revision.

---

> > ### Author Rebuttal · Reviewer_FkDG · 2026-04-03
> >
> > Thank you for the thorough rebuttal. The added statistical stability analysis, the clarification of what constitutes common spreadsheet errors, the multi-simulator robustness study for DV-Inter, and the sensitivity analysis over weighting choices directly address my main questions.
> >
> > In particular, the new evidence makes it much more convincing that the paper’s conclusions are not artifacts of the benchmark size, a single simulator, or one particular score aggregation scheme. I also appreciate the audit of simulator faithfulness against human judgments, which strengthens the case for GPT-5-Mini as a practical default simulator.
> >
> > Overall, my concerns have been adequately addressed, and I remain supportive of the paper.

---

> > > ### Author Response · Authors · 2026-04-03
> > >
> > > We sincerely appreciate your positive feedback and recognition. Your support is truly encouraging to us and motivates us to make DV-World a meaningful contribution to the community.

---

### Decision · Program_Chairs · 2026-04-30

**Decision:**

Accept (regular)

**Comment:**

This paper introduces DV-World, a benchmark of 260 tasks evaluating data visualization agents across three domains: spreadsheet-native manipulation (DV-Sheet), cross-framework visualization evolution (DV-Evol), and multi-turn interactive intent alignment (DV-Inter). The benchmark is accompanied by a well-validated evaluation framework combining LLM-based judging with human alignment checks.

Three reviewers strongly support acceptance. Reviewer FkDG (5) raised their score to 6 after the rebuttal, confirming that the added statistical stability analyses (paired bootstrap, permutation tests, Kendall's tau) and expanded data complexity statistics fully addressed their concerns. Reviewer E7HT (5) found all concerns resolved and raised their score. Reviewer uVLn (6) confirmed their concerns were addressed and maintained their score. Reviewer Sud8 (2) raised concerns about the "real-world" framing given sanitized data, data complexity, and novelty of individual sub-tasks; S/he did not respond to the rebuttal or participate in the discussion despite a reminder from the AC, and his or her review has been weighted accordingly in the final decision. Two residual concerns remain but neither is considered blocking. First, the "real-world" framing may overstate the benchmark's scope, as full data cleaning and preprocessing challenges are not the primary focus. However, the authors demonstrated non-trivial workbook complexity (35% missing values, 28% cross-sheet aggregation, 22% irregular layouts) and committed to adjusting the framing. Second, whether DV-Evol measures visualization reasoning versus code translation remains a point of conceptual debate, though the authors clarified it as reference-guided regeneration rather than syntax porting, which is a reasonable characterization.

Overall, this is a well-constructed and practically relevant benchmark that fills a clear gap in evaluating data visualization agents. I recommend weak acceptance.